# Improving air quality forecasting with the assimilation of GOCI AOD retrievals during the KORUS-AQ period

Soyoung Ha[1], Zhiquan Liu[1], Wei Sun[1], Yonghee Lee[2], and Limseok Chang[2]

[1]National Center for Atmospheric Research, Boulder, Colorado, USA
[2]National Institute of Environmental Research, Incheon, South Korea

**Correspondence:** Soyoung Ha (syha@ucar.edu)

**Abstract.** The Korean Geostationary Ocean Color Imager (GOCI) satellite has monitored the East Asian region in high temporal (e.g. hourly) and spatial resolution (e.g. 6 km) every day for the last decade, providing unprecedented information on air pollutants over the upstream region of the Korean peninsula. In this study, the GOCI aerosol optical depth (AOD), retrieved at 550 nm wavelength, is assimilated to ameliorate the quality of the aerosol analysis, thereby making systematic improvements on air quality forecasting over South Korea. For successful data assimilation, GOCI retrievals are carefully investigated and processed based on data characteristics such as temporal and spatial distribution. The preprocessed data are then assimilated in the three-dimensional variational data assimilation (3DVAR) technique for the Weather Research and Forecasting model coupled with Chemistry (WRF-Chem). During the Korea-United States Air Quality (KORUS-AQ) period (May 2016), the impact of GOCI AOD on the accuracy of surface $PM_{2.5}$ prediction is examined by comparing with effects of other observations including Moderate Resolution Imaging Spectroradiometer (MODIS) sensors and surface $PM_{2.5}$ observations. Consistent with previous studies, the assimilation of surface $PM_{2.5}$ measurements alone still underestimates surface $PM_{2.5}$ concentrations in the following forecasts and the forecast improvements last only for about 6 h. When GOCI AOD retrievals are assimilated with surface $PM_{2.5}$ observations, however, the negative bias is diminished and forecast skills are improved up to 24 h, with the most significant contributions to the prediction of heavy pollution events over South Korea.

## 1 Introduction

With the recent increase of chemical and aerosol observations in the troposphere, chemical data assimilation is expected to play an essential role in improving air quality forecasting, particularly in the real-time environment. Although various data assimilation (or analysis) techniques have been developed for many decades, they were predominantly applied in the context of numerical weather prediction (NWP) (Kalnay, 2003) and have not been extensively exploited for the prediction of air pollution.

Uncertainties in aerosol chemistry, as well as its multiscale interactions with daily changing weather conditions, make it challenging to predict air pollutants accurately (Grell and Baklanov, 2011; Baklanov et al., 2014; Kong and coauthors, 2015; Baklanov et al., 2017). Surface concentrations are directly affected by transport and dispersion of chemical species through advection, convection, vertical diffusion and surface fluxes. In general, they are strongly driven by external forcing such as anthropogenic and natural emissions. The latter heavily relies on temperature, humidity, and wind speed in the boundary layer

as well as solar radiation and soil moisture. Aerosols in turn affect local meteorology via aerosol-meteorology interaction (by directly scattering and absorbing solar radiation and also as sources of cloud condensation nuclei) at short time scales. For the operational air quality forecasting in South Korea, the Korean National Institute of Environmental Research (NIER) performs chemical simulations on 3-km resolution at present (Chang et al., 2016). For such a high-resolution application and for situations with very high aerosol concentrations, these fast-varying complex mechanisms might be better represented through online coupling between chemical and meteorological components. The online coupled forecasting system is particularly suitable for air quality forecasting associated with strong synoptic forcing or long-range transport of air pollutants. Also, finer scale features may require more frequent coupling of the atmospheric system and only the online coupled system can provide the framework for such applications.

With large uncertainties in chemical modeling and emission data, particularly associated with meteorological components, one of the most effective ways of utilizing aerosol observations is to assimilate them into the forecast model and improve the initialization of aerosol simulations. However, due to the scarcity of three-dimensional chemical observations and the complexity of how to project the observed information (usually in the optical properties) onto the parameterized schemes in the chemical model, aerosol/chemical data assimilation in the coupled chemistry and meteorology models has been limited to date (Bocquet et al., 2015). Improving the quality of chemical assimilation will not only improve the prediction of air pollution, but also advance numerical weather prediction (NWP) for precipitation, visibility, and high impact weather.

An international cooperative air quality field study conducted in Korea, named as the Korea-United States Air Quality (KORUS-AQ), was a field campaign jointly developed by air quality researchers in the United States and South Korea to improve our understanding of major contributors to poor air quality in Korea for May 1-June 12, 2016. During this early summer time when it is mostly warm and humid, numerous measurements of pollutants were made at multiple platforms in an effort to identify local and transboundary pollution sources contributing to the formation of ozone and fine particulate matter ($PM_{2.5}$). Although local emissions played a nontrivial role throughout the period, the highest pollution event occurred by long-range transport from the upwind area on May 25-27, 2016 (Miyazaki et al., 2019). As the transboundary transport cannot be fully measured by surface stations over land, a proper use of satellite data that have a wide spatial coverage would have great potential to improve air quality forecasting for such events.

The Korean Geostationary Ocean Color Imager (GOCI) onboard the Communication, Ocean, and Meteorology Satellite (COMS) provides hourly AOD retrievals at multiple spectral bands monitoring the East Asian region centered on the Korean peninsula during daytime (Kim et al., 2017). Since its launch in 2010, the GOCI satellite has been producing AOD retrievals at high spatial and temporal resolution. It has long been demonstrated that the GOCI data were in high accuracy, comparable to the low-orbiting Moderate Resolution Imaging Spectroradiometer (MODIS) and Visible Infrared Imaging Radiometer Suite (VIIRS) products (Lee et al. (2010); Wang et al. (2013); Xiao et al. (2016); Choi et al. (2018)).

Liu et al. (2011) first implemented the capability of assimilating Aerosol Optical Depth (AOD) retrieved from MODIS satellite sensors (Remer et al., 2005) into the National Centers for Environmental Prediction (NCEP) Gridpoint Statistical Interpolation (GSI; Wu et al. (2002); Kleist et al. (2009)) system. Since they confirmed that the AOD assimilation improved aerosol forecasts in a dust storm event that occurred in East Asia, the GSI three-dimensional variational data assimilation

(3DVAR) system has been widely used for air quality forecasting and extended for additional aerosol observations such as surface particulate matter - all particles with aerodynamic diameter less than 2.5 $\mu$m (PM$_{2.5}$) or up to 10 $\mu$m (PM$_{10}$) (Schwartz et al. (2012) and Jiang et al. (2013), respectively).

GOCI AOD retrievals have been assimilated in several studies to assess their impact on short-term air pollution forecasts in the online coupled forecasting system. Saide et al. (2014) performed the Observing System Experiment (OSE) using the eight bin MOdel for Simulating Aerosol Interactions and Chemistry aerosol model (MOSAIC) (Zaveri et al., 2008) in the WRF-Chem/GSI 3DVAR system. Pang et al. (2018) assimilated AOD retrievals from GOCI and Visible Infrared Imaging Radiometer Suite (VIIRS; Jackson et al. (2013)) to predict surface PM$_{2.5}$ concentrations over Eastern China and found that the assimilation of AOD retrievals improved the forecast accuracy but still underestimated heavy pollution events.

This work further extends the assimilation capabilities in the GSI 3DVAR system to best use GOCI AOD retrievals during the KORUS-AQ period with careful investigation of data characteristics. Aiming at improving the operational air quality forecasting in Korea, which is currently lacking the state-of-the-art analysis system, we are discussing how to effectively assimilate satellite-derived aerosol data and examine its impact on surface PM$_{2.5}$ predictions compared to that of other observations. In the categorical forecasts for different air pollution events, we focus on severe pollution cases describing how air pollutants evolve, coupled with the synoptic weather systems.

A brief overview of the analysis and forecasting systems used in this study is presented in Section 2, followed by cycling experiments with details on observation processing for GOCI retrievals described in Section 3. Results are summarized in Section 4 discussing the observation impact during the cycles and extended forecasts separately. Forecast performances in heavy pollution events are briefly described as well. Finally, conclusions are made in Section 5, along with a discussion on the limitations of this study and suggestions for the future research.

## 2 The WRF-Chem forecast model and the GSI 3DVAR analysis system

### 2.1 WRF-Chem forecast model

The model used in this study is an online-coupled meteorology and chemistry model, WRF-Chem version 3.9.1 (Grell et al., 2005). The physics options used in WRF-Chem include the rapid and accurate radiative transfer model for GCM (RRTMG) for long-wave radiation (Iacono et al., 2008), new Goddard shortwave radiation (Chou and Suarez, 1994), the Yonsei University (YSU) planetary boundary layer (PBL) scheme (Hong et al., 2006), the Lin microphysics scheme (Lin et al., 1983), as well as a new Grell 3D cumulus parameterization scheme. These options are chosen based on the operational configuration currently used in the Korean National Institute of Environmental Research (NIER) for their daily air quality forecasting in South Korea. The Goddard Chemistry Aerosol Radiation and Transport (GOCART; Chin et al. (2002)), developed by the National Aeronautics and Space Administration (NASA), is used as an aerosol scheme. Aerosol direct effects are allowed through the interaction between GOCART and the Goddard shortwave radiation scheme (Fast et al. (2006); Barnard et al. (2010)).

The Model for Ozone and Related Chemical Tracers (MOZART) gas phase chemistry (Emmons et al., 2010) is generated with the kinetic preprocessor (KPP) (Damian et al. (2002); Sandu and Sander (2006)), and is used together with the simple

GOCART aerosol scheme, known as the MOZCART mechanism (Pfister et al., 2011). The MOZART chemistry in WRF-Chem is designed to run with the Madronich FTUV scheme for photolysis processes (Tie et al., 2003), reading in climatological O3 and O2 overhead columns. It also utilizes the standard WRF-Chem implementation of the Wesley dry deposition scheme (based on Wesely (1989)) allowing for seasonal changes in the dry deposition. The resolved scale wet scavenging is inactivated but convective wet scavenging is applied in the Grell cumulus parameterization. Also, GOCART sea salt emissions and dust emissions with AFWA modifications (LeGrand et al., 2019) are included in this study.

Anthropogenic emissions are estimated offline based on the global EDGAR-Hemispheric Transport of Air Pollutants (HTAP) emission inventory (http://edgar.jrc.ec.europa.eu/htap_v2/) that consisted of 0.1° x 0.1° gridmaps of CH4, CO, SO2, NOx, NMVOC, NH3, PM10, PM2.5, BC and OC from the year of 2010. The emission data mapped to our model grids has a single level with no vertical variations and is generated from the annual mean with no diurnal variations (e.g. time-invariant). In terms of data range, the maximum (average) value of $PM_{2.5}$ in the data, for example, is 3.56 (0.032) $\mu$g m$^{-2}$ s$^{-1}$ and 2.84 (0.026) $\mu$g m$^{-2}$ s$^{-1}$ in domain 1 and 2, respectively.

Biogenic emissions are built up using the Model of Emission of Gases and Aerosol from Nature (MEGAN; Version 2) (Guenther et al., 2006) and for biomass burning emissions, daily fire estimates provided by Fire Inventory from NCAR (FINN; Wiedinmyer et al. (2011)) are used with tracer transport allowed. All the WRF files including biomass and biomass burning emissions are processed using the MODIS landuse datasets (Friedl et al., 2002).

### 2.1.1 The GSI 3DVAR analysis system

To assimilate AOD retrievals and surface $PM_{2.5}$ observations in the Weather Research and Forecasting-Chemistry (WRF-Chem) model, the NCEP GSI 3DVAR Version 3.5 system is used. As Liu et al. (2011) and Schwartz et al. (2012) described the details of the system for aerosol data assimilation, only a brief explanation follows. Incorporating observations into the three-dimensional model state space, a 3DVAR system produces the best estimate to the true state by minimizing the differences between observations and background forecasts (e.g. innovations; (o-b)'s), which is called the "analysis". (The analysis is then used to initialize aerosol variables in the forecast model (e.g. WRF-Chem) so that the quality of aerosol forecasts can be largely dependent on the quality of the aerosol analysis produced in the 3DVAR system.) Given the model state vector ($\mathbf{x}$), the penalty function (or cost function) $J(\mathbf{x})$ is defined as

$$J(\mathbf{x}) = \frac{1}{2}(\mathbf{x} - \mathbf{x_b})^{\mathbf{T}}\mathbf{B}^{-1}(\mathbf{x} - \mathbf{x_b}) + \frac{1}{2}(H(\mathbf{x}) - \mathbf{y})^{\mathbf{T}}\mathbf{R}^{-1}(H(\mathbf{x}) - \mathbf{y}),  \tag{1}$$

where $\mathbf{x_b}$ stands for the background state vector (e.g. forecasts from the previous cycle), $\mathbf{y}$ an observation vector, and $H$ is an observation operator that projects the model states onto the observation space linearly or nonlinearly to compute the model correspondent to each observation. Background and observation error covariance matrices $\mathbf{B}$ and $\mathbf{R}$, respectively, indicate how reliable the background forecast ($\mathbf{B}$ in the first term) and the observed information ($\mathbf{R}$ in the second term) might be to determine how to properly weight the two disparate resources. By minimizing the cost function ($J(\mathbf{x})$) with respect to the model state vector $\mathbf{x}$ at the analysis time, the variational analysis algorithm produces the analysis that fits best to all the observations assimilated within the assimilation time window.

To characterize the forecast error magnitude and its spatial structure, background error covariance $\mathbf{B}$ is estimated for each aerosol species using the National Meteorological Center (NMC) method (Parrish and Derber, 1992) based on the differences between 48- and 24-h WRF-Chem forecasts valid at the same time for 30 samples ending at 0000 UTC in May 2016. The current GSI/3DVAR system does not allow cross-correlation between aerosol species or between aerosol and meteorological variables. As this is a 3DVAR analysis with no time information, B only characterizes the spatial correlations in each analysis variable, which determines how to propagate the observed information across the model grids.

Following Liu et al. (2011) and Schwartz et al. (2012), this study also takes the speciated approach where the analysis vectors are comprised of 15 WRF-Chem/GOCART aerosol variables - sulfate, organic carbon (O) and black carbon (B), mineral dust (D) in five particle-size bins (with effective radii of 0.5, 1.4, 2.4, 4.5, and 8.0 $\mu$m), and sea salt (S) in four particle-size bins (with effective radii of 0.3, 1.0, 3.25, and 7.5 $\mu$m for dry air), and $P$ as unspeciated aerosol contributions to PM$_{2.5}$ -, as opposed to using total aerosol mass of PM$_{2.5}$ as the analysis variable in Pagowski et al. (2010). For organic and black carbon, hydrophobic and hydrophilic components are considered (e.g. $O_1$, $O_2$, $B_1$, and $B_2$).

The observation operator $H(\mathbf{x})$ for surface PM$_{2.5}$ requires 10 GOCART aerosol variables as

$$H(\mathbf{x}) = \rho_d[P + D_1 + 0.286D_2 + 1.8(O_1 + O_2) + B_1 + B_2 + S_1 + 0.942S_2 + 1.375U], \tag{2}$$

where P represents unspeciated aerosol contributions to PM$_{2.5}$; U denotes sulfate; $O_1$ and $O_2$ ($B_1$ and $B_2$) are hydrophobic and hydrophilic organic (black) carbon, respectively; and $D_1$ and $D_2$ ($S_1$ and $S_2$) are dust (sea salt) aerosols in the smallest and 2nd smallest size bins. This formula originated from the WRF-Chem diagnostics of PM$_{2.5}$ for the GOCART aerosol scheme. PM observations are mass concentrations in $\mu$g/m$^3$ while all the model variables listed within the bracket in the right-hand side are aerosol mixing ratios ($\mu$g/kg), dry density $\rho_d$ is thus required for the unit conversion in equation 2.

In this study, we assimilate AOD retrievals at 550 nm from both MODIS and GOCI sensors using the same observation operator based on the community radiative transfer model (CRTM; Han et al. (2006); Liu and Weng (2006)) as described in Liu et al. (2011). Although the GOCART aerosol scheme is well known to underestimate surface PM concentrations due to the lack of secondary organic aerosol (SOA) formation, nitrate, and ammonium (Liu et al. (2011); Volkamer et al. (2006); McKeen et al. (2009); Pang et al. (2018)), it is widely used in the analysis study because it is the only scheme publicly available for assimilating AOD retrievals from satellite data in the GSI system. Aerosol optical depth (AOD) measures the amount of light extinction by aerosol scattering and absorption in the atmospheric column which depend on the refractive indices and the size distribution of aerosol. In GSI, the CRTM computes the effective radii and the refractive indices of the 14 speciated WRF-Chem/GOCART aerosol species, assuming spherical aerosol particles and lognormal size distributions. Applying single-scattering properties of spheres by Mie theory, the mass extinction coefficient is computed as a function of the effective radius for each aerosol species at a certain wavelength (here, 550 nm) at each model level. The mass extinction coefficient (m$^2$/g) for each aerosol species multiplied by the aerosol layer mass (g/m$^2$) produces dimensionless AOD for the species at that level. To represent the entire atmospheric column, model-simulated AOD is then computed as the column integration of AOD for all aerosol species. Using the CRTM as a forward operator, AOD retrievals are assimilated separately or simultaneously with PM$_{2.5}$ observations from the surface network over East Asia, as described in the following section.

## 3 Cycling Experiments

During the month of May 2016, observations are assimilated in the GSI 3DVAR system to produce the analysis that is used as an initial condition for the following WRF-Chem simulations. WRF-Chem forecasts valid at the next analysis time are then used as first-guess (or background) for the next GSI analysis. In this study, the whole process is repeated every 6 h (called "cycled") for the month-long period. Here we describe the analysis and the forecast systems used in the cycling.

### 3.1 Model configurations and cycling

All the analyses and the following forecasts are conducted over two one-way nested domains centered on the Korean peninsula, as shown in Fig. 1. Domain 1 uses 175 x 127 horizontal grids at 27-km resolution and domain 2 has 97 x 136 grids at 9-km resolution. Both domains have total of 31 vertical levels up to 50 hPa. The initial and boundary meteorological conditions for domain 1 are provided by the U.K. Met Office Unified Model (UM-MET) global forecasts operated by the Korean Meteorological Administration (KMA) with a horizontal resolution of $\sim$25 km ($0.3515°$x$0.234375°$) at 26 isobaric levels every 6 h. This configuration was chosen in the limitation of computational resources, but the use of higher resolutions both in time and space might be desirable to further improve forecast skills in the future. The chemical initial and boundary conditions for domain 1 are taken from the output of the global Model for Ozone and Related Chemical Tracers (MOZART-4) (Emmons et al., 2010) that are converted to WRF-Chem species by using the "mozbc" utility (downloaded from https://www2.acom.ucar.edu/wrf-chem/wrf-chem-tools-community/). Meteorological and chemical fields in domain 1 are reinitialized from global forecasts every cycle while initial and boundary conditions for domain 2 are nested down from domain 1 in a one-way nesting. Aerosol and chemical initial conditions are then overwritten by WRF-Chem forecasts from the previous cycle in each domain. The GSI analysis is consecutively performed in the two domains using the same observations within each domain. During the cycles, 24 h forecasts are initialized from the 00Z analysis every day.

### 3.2 Observations

#### 3.2.1 Surface PM$_{2.5}$

Hourly surface PM concentrations are provided by the NIER which collects real-time pollutant observations at 361 South Korean stations from AirKorea (http://www.airkorea.or.kr) and those at $\sim$900 Chinese sites from China National Environmental Monitoring Centre (CNEMC; http://www.cnemc.cn). Figure 1 shows the entire surface observing network that was used to assimilate surface PM$_{2.5}$. Observation sites are concentrated in the urban area where many sites are close enough to be overlapped with each other. The Seoul Metropolitan Area (SMA; centered around 37.5°N, 127°E), for example, has hourly reports from total of 41 stations.

As part of data quality control (QC), surface PM$_{2.5}$ concentrations higher than 100 $\mu$g/m$^3$ are not assimilated and observations producing innovations ((o-b)'s) that exceed 100 $\mu$g/m$^3$ were also discarded during the analysis step. To accommodate most measurements in China during heavy pollution events, a much higher threshold of 500 $\mu$g/m$^3$ was once applied as the

maximum observed value in our test experiment for the same month-long cycles, but it did not lead to any meaningful changes in the forecast performance over South Korea (not shown). Presumably this is because such high values were observed only over China where air pollutants were already overestimated by the emission data based on the 2010 inventory such that the forecast skills over Korea became insensitive to the assimilation of those additional surface observations in China. Therefore,
we applied the original threshold of 100 $\mu$g/m$^3$ to all our experiments presented here.

Observation error is composed of measurement error ($\epsilon_o$) and the representative error ($\epsilon_r$) caused by the discrete model grid spacing (e.g. $\epsilon_{pm_{2.5}} = \sqrt{\epsilon_o^2 + \epsilon_r^2}$). Following Elbern et al. (2007) and Schwartz et al. (2012), observation error for surface PM$_{2.5}$ increases with the observed value ($x_o$) as $\epsilon_o = 1.5 + 0.0075 * x_o$. The representative error is formulated as $\epsilon_r = \gamma \epsilon_o \sqrt{\frac{\Delta x}{L}}$ where $\gamma$ is 0.5, $\Delta x$ is grid spacing (here, 27 km for domain 1 and 9 km for domain 2) and the scaling factor L is defined as
3 km. Based on this formula, observation error ($\epsilon_{pm_{2.5}}$) ranges from 2.0 to 3.2 $\mu$g/m$^3$ in domain 2, assigining the error of 2.48 $\mu$g/m$^3$ to the PM$_{2.5}$ observation of 50 $\mu$g/m$^3$, for example. In this 3DVAR analysis, observation errors are considered to be uncorrelated so that the observation error covariance matrix **R** becomes diagonal. During the 6-h cycling, all the surface observations within $\pm 1$ h window at each analysis time were assimilated without further adjustment of observation error.

### 3.2.2 AOD retrievals and observation preprocessing

Total AOD retrievals at 550 nm from MODIS sensors onboard Terra and Aqua satellites have been widely used in aerosol studies (Zhang and Reid, 2006, 2010; Lee et al., 2011). But the polar-orbiting satellites produce a very limited dataset temporally (mostly around 06 UTC only) and spatially (with a sparse coverage) over Korea during the KORUS-AQ period. The MODIS AOD level 2 products over both land and ocean "dark" area are available at 10 km x 10 km resolution and thinned over 60-km resolution during the GSI analysis in this study. Following Remer et al. (2005), observation errors are specified as the retrieval
errors: (0.03 + 0.05 * AOD) over ocean and (0.05 + 0.15 * AOD) over land. They do not include the representativeness error and are slightly smaller than those for GOCI AOD, as described below.

The GOCI satellite monitors the East Asian region centered on the Korean peninsula (36°N, 130°E) covering about 2500 km × 2500 km. GOCI level II data has eight spectral bands from the visible to near-infrared range (412 to 865 nm) with hourly measurements during daytime from 9:00 (00 UTC) to 17:00 local time (08 UTC) at 6 km resolution. As summarized in Choi
et al. (2018), a recently updated GOCI Yonsei aerosol retrieval (YAER) Version 2 algorithm targets cloud- and snow-free pixels over land and cloud- and ice-free pixels over ocean in producing the level II data. By adopting the MODIS and VIIRS aerosol retrieval and cloud-masking algorithms, cloud pixels are filtered to avoid cloud contamination, and high reflectance or highly heterogeneous reflectance pixels are also masked to further increase data accuracy and consistency during the retrieval process.

Unlike MODIS retrievals, GOCI AOD has not been extensively used in the data assimilation community. The GSI system
takes most observation types in prepbufr format, which has already gone through some processing to be prepared for data assimilation, but the preprocessing algorithms are not publicly available. This means that, when a new dataset is assimilated in GSI, users need to investigate the characteristics of the data (such as temporal and spatial distribution) and thereby make the data suitable for assimilation, which is of crucial importance for the analysis quality.

In terms of temporal distribution, most of GOCI level II data are retrieved on 30 min passed each hour in the hourly report. For example, the actual time for most of the data reported at 00 UTC is centralized around 00:30:00 UTC (hh:mm:ss). In the 3DVAR algorithm, there is no time dimension and all observations are considered to be available at the analysis time. To account for temporal distribution, different weights are often given to observations based on the relative distance between the actual report time and the analysis time during the analysis step. However, taking possible latency in data transfer and retrieval processing into consideration, it is not legitimate to assign weights to the retrievals based on their final report time, without further information. Therefore, considering high temporal and spatial variability of aerosols, the assimilation window is set to $\pm1$ h in order to avoid inconsistent observed information within the window in this study.

Satellite data are known to have a large positive impact on the analysis quality thanks to the high data volume both in time and space, but such high density violates the assumption of uncorrelated observation errors in the analysis algorithm and increases the computation time for the analysis step excessively. Hence, a large volume of satellite retrievals are typically sampled on a regularly spaced grid through the horizontal thinning procedure. In GSI, satellite radiance data can be thinned such that retrievals are randomly sampled at a predefined spacing for each instrument type before getting ingested into the observation operator during the analysis (Rienecker and Coauthors, 2008). This thinning procedure, however, can pick up inconsistent data (near the cloud boundaries, for instance) and is reported as suboptimal (Ochotta et al., 2005; Reale et al., 2018). Therefore, we decided to preprocess GOCI AOD retrievals with superobing where all the data points are averaged within a certain radius. In this study, we superobed GOCI retrievals over each grid box in domain 1 (at 27-km resolution). Figure 2 shows the sample horizontal distribution of GOCI AOD retrievals valid at 06 UTC May 1 2016 before (a) and after (b) preprocessing them, comparing with those thinned over 60 km (c) and 27 km meshes (d) during the GSI analysis, respectively. Some high AOD values in the original dataset (as shown in a), especially on cloud edges, cannot be fully resolved by our 27-km model grids. By averaging all data points over each grid box at 27-km resolution, the superobed data in b) have a better quality control throughout the domain reducing the data volume effectively. A total number of observations marked in the upper right corner of each panel indicates that thinning over the 60 km mesh in c) reduces the number of assimilated observations to 2.5% of that in the original level II data while superobing and thinning over 27-km mesh utilize 8-10 % of the original data representing the whole data coverage fairly well.

It might be noteworthy to make two more points related to data processing here. First, superobing was applied as part of preprocessing before the GSI analysis gets started while the thinning was conducted during the analysis step so that the preprocessing could speed up the GSI analysis up to 25 times (by injecting less than 10 % of the original data and turning off the thinning process). This can facilitate the use of satellite retrievals in the operational air quality forecasting. Next, the thinning algorithm in GSI V3.5 resulted in erroneous values in some places, as indicated by the maximum values in c) and d). For the month of May 2016, multiple cases with such extreme fake values were found after the thinning process. This bug may need to be fixed in the GSI or avoided by bounding the values exceeding the original data.

To examine the effect of data processing on the performance of the analysis and the background during the cycles, we compare two cycling experiments - one with the assimilation of the original level II data thinned over 27-km mesh (named "GOCI_orig" in gray) and the other with the assimilation of GOCI retrievals preprocessed over 27-km grids in domain 1 (called

"GOCI" in black) - in Fig. 3. As GOCI data are reported from 00 to 08 UTC, only 00 and 06 UTC cycles are shown here in consecutive cycle numbers. The time series of (o-a)'s and (o-b)'s in each experiment show that the preprocessed data slightly fit better to the observations than the thinned data, assimilating more retrievals throughout the period. Because the differences between the two experiments are not significant, for the computational efficiency, we decided to preprocess all the GOCI retrievals and assimilate them turning off the thinning process in GSI for the rest of the experiments shown in this study.

Choi et al. (2018) described their improved retrieval algorithm (GOCI YAER V2) with updated cloud-masking and surface reflectance calculations, making a long-term evaluation against other ground- and satellite-based measurements. In their study, depending on the verifying objects - either ground-based Aerosol Robotic Network (AERONET) or satellite-based retrievals -, they specified uncertainties of GOCI AOD retrievals over land and ocean using two different linear regression formulae. We assign $\epsilon_1$ following their error specification with respect to AERONET and $\epsilon_2$ based on their expected error against retrieved satellite AOD in GOCI YAER V2.

$$\epsilon_1{}^{land} = 0.061 + 0.184\tau_A \tag{3}$$

$$\epsilon_1{}^{ocean} = 0.030 + 0.206\tau_A \tag{4}$$

$$\epsilon_2{}^{land} = 0.073 + 0.137\tau_A \tag{5}$$

$$\epsilon_2{}^{ocean} = 0.037 + 0.185\tau_A \tag{6}$$

where $\tau_A$ stands for GOCI AOD values. In an effort to account for representativeness error, we also tried with $\epsilon_2$ increased by 20% everywhere as the third error formula (e.g. $\epsilon_3 = 1.2 \times \epsilon_2$) and compared all three types of errors in Fig. 4. When these different observation errors were applied to GOCI retrievals in the assimilation, the smallest error ($\epsilon_2$) produced slightly better fits to observations specially for the high values (AOD > 2) during the cycles, as expected, but not in a statistically meaningful way (not shown). In fact, it is not straightforward to estimate the representativeness error which is subject to the model resolution (both in horizontal and vertical) and data processing in use. Therefore, in many cases, observation error is specified based on the resulting forecast performance (Ha and Snyder, 2014). But because our forecast skills were not very sensitive to three different error formulae tried here, for the rest of the experiments, $\epsilon_2$ is used as observation error for GOCI retrievals.

The goal of this study is to examine the relative impact of the GOCI assimilation on the prediction of surface $PM_{2.5}$ and ultimately to improve the forecasts for pollution events. Although it is rather easy to render the analysis close to GOCI observations by reducing the observation error, it is not guaranteed that the analysis in a good agreement with AOD retrievals would actually lead to better forecasts in surface $PM_{2.5}$. This is partly because AOD, a column integrated quantity, is not directly associated with surface $PM_{2.5}$ and partly because large uncertainties in the forecast model and the emission forcing can dominate over the analysis error during the model integration. Even if the efficiency of assimilating AOD on improving surface $PM_{2.5}$ forecasts can be largely affected by the quality of the forecast model and the emission data in use, the effectiveness of the AOD assimilation is based on the relationship between the column-integrated AOD and $PM_{2.5}$ on the ground. Therefore, it might be

worth checking the correlation between GOCI AOD retrievals and surface $PM_{2.5}$ observations for the cycling period. Figure 5 depicts a scatter diagram of GOCI AOD retrievals at 550 nm and surface $PM_{2.5}$ observations that are co-located in each grid box in domain 1 for the month of May 2016. As shown with the linear regression coefficient of 0.33, the two observation types have low correlations during this period, which is consistent with previous studies (Saide et al., 2014; Pang et al., 2018). Such an indirect relationship between the two observations makes the analysis challenging because it can induce a large error in the observation operator and heavily depend on the model's ability of deriving $PM_{2.5}$ from AOD based on the vertical structure of aerosol variables and the conversion from aerosol mass to optical properties.

## 4 Results

With a careful design of model configuration and observation processing, the overall impact of assimilating all the available observations ("DA") is illustrated, compared to the baseline experiment without data assimilation ("NODA") in Fig. 6. Here, the 0-23 h hourly forecasts from all the 00Z analyses in domain 2 are concatenated for the entire month. Surface $PM_{2.5}$ observations marked as black dots show that the air quality gets distinctively aggravated for the last 7 days, related to long-range transport of air pollutants. With data assimilation ("DA"), the analyses at 00Z and the following forecasts (red) make a better agreement with corresponding observations than those without assimilation (gray), especially from day 15 (e.g. after a full spin-up for two weeks). In particular, on May 25-27, forecast error grows quickly even from the good analysis at 00Z, possibly associated with large uncertainties in lateral boundary conditions and the forecast model in use. However, averaged over the entire period, the mean absolute error (mae) indicates that the performance of 0-23-h forecasts at 9-km resolution gets improved by $\sim$30% through data assimilation.

### 4.1 Observation impact during the cycles

Given that the aerosol assimilation has a positive impact on air quality forecasting, it might be worth isolating the contribution of each observation type to the improvement of the analysis and the following forecasts. We first assimilate individual observation types separately, naming the experiment following each observation type, then we assimilate them all together (called "ALL"). Figure 7 illustrates the vertical profile of 10 three-dimensional GOCART aerosol variables that are used in diagnosing $PM_{2.5}$ in the GOCART scheme, in the analysis (solid) and background (e.g. 6-h forecast; dashed) averaged over domain 2. Assuming that cycles may need to spin up meteorology and chemistry at least for three days in the regional simulations, all the statistics are computed from day 4 in the rest of the figures. Although the analysis variables only at the lowest model level are used in the observation operator for surface $PM_{2.5}$, the observation impact is detected throughout the atmosphere due to the spatial correlations specified in the background error covariance. Contributions of different observations to each analysis variable vary, with the largest variability in the analysis increments (analysis-minus-background) displayed in sulfate. Interestingly, a large impact of AOD retrievals is noticed in hydrophilic organic carbon ($O_2$) aloft (e.g. between 12 and 25 levels) and unspeciated aerosol (P) in the boundary layer. The assimilation of all the observations ("ALL") tends to reduce $O_2$, dust in both size bins ($D_1$ and $D_2$) and unspeciated aerosol (P) in the lower atmosphere.

Figure 8 summarizes the effect of different observations on $PM_{2.5}$ in both domains. The assimilation of surface $PM_{2.5}$ observations (green) results in the smallest $PM_{2.5}$ while the GOCI assimilation (blue) produces the largest $PM_{2.5}$ throughout the atmosphere in both domains. When the analysis (solid line) is compared to background (dashed), it is revealed that $PM_{2.5}$ is predominantly increased over domain 1 with the assimilation of GOCI retrievals. Overall, the aerosol assimilation affects the entire profile of $PM_{2.5}$ with the largest impact at the surface. It is noted that the vertical distribution of model aerosol species is associated with the vertical stratification of the model as well as the vertical distribution of the species in the background error covariance. It might be worth evaluating the vertical structure of individual species simulated in the model with respect to the vertical profiles observed during the KORUS-AQ field campaign (such as NASA DC-8 aircraft) in the future, although all the flight tracks were limited to the vicinity of the Korean Peninsula (Peterson et al., 2019).

To understand the observation impact in the horizontal distribution, Fig. 9 shows the analysis increments (analysis-minus-background) averaged over the period of May 4-31, 2016. Generally, the assimilation of surface $PM_{2.5}$ observations ("PM") reduces surface $PM_{2.5}$ over most regions in China while the GOCI assimilation largely increases surface $PM_{2.5}$ almost everywhere, consistent with Fig. 8. As MODIS retrievals have a relatively low coverage of the East Asian region for the entire period, they have the smallest impact among all the observation types. When all the observations are assimilated together (in "ALL"), it combines the effect of surface $PM_{2.5}$ and GOCI retrievals, changing the vertical distribution of aerosol species to match with the AOD column values and pulling the surface states towards surface $PM_{2.5}$ concentrations. While the observing network of surface $PM_{2.5}$ is widely distributed over China, the impact of GOCI data is more centralized over Korea, making unequivocal contributions to air quality forecasting in the Korean peninsula.

Note that we employ the 2010 inventory for our emission data, which does not reflect the emission control started from 2013 in China (Zheng et al., 2018). Given that air pollutants in the emission data constitute the majority of the precursors of $PM_{2.5}$ pollution, surface $PM_{2.5}$ concentrations could strongly depend on emissions which might have led to the overestimation in the background (e.g. first-guess). Therefore, the assimilation of surface $PM_{2.5}$ tends to counteract the overestimation driven by the emission data over China. On the other hand, over South Korea, the emission data does not seem to be overestimated and the assimilation of surface $PM_{2.5}$ leads to increasing surface $PM_{2.5}$ most effectively during the cycles.

Different from surface particulate matter, AOD in the background is contingent upon the optical properties described in the observation operator (e.g. CRTM) and the vertical structure of aerosols simulated in the column. The influence of GOCI assimilation may indicate the model deficiencies in the two aspects because the model states are pulled toward the observed information during the analysis step, as depicted in the analysis increment.

## 4.2 Observation impact on 24-h forecasts

Since the real effect of data assimilation is manifested in the subsequent forecasts, we now examine forecast improvements when initialized by our own analyses. A good analysis is expected to slow down the forecast error growth, leading to better forecasts. In this subsection, forecast errors at the lowest model level are compared between experiments for 24 h with respect to surface observations from various sites in South Korea. As we focus on 9-km simulations over the Korean peninsula, it is hard to anticipate the direct effect of the assimilation beyond 24 h, specially in such a small domain where the weather

systems dramatically change from day to day. As shown in Fig. 10, the forecast error is the largest in the baseline experiment ("NODA"), followed by the assimilation of MODIS retrievals alone ("MODIS") in terms of mean absolute error (mae). Note that the analysis in the "PM" experiment is verified against the same surface $PM_{2.5}$ observations used in the assimilation. Therefore, the analysis error is smaller than those in other experiments, but the forecast error grows quickly over the next 24

h. The assimilation of surface $PM_{2.5}$ alone generally underestimates the prediction of surface $PM_{2.5}$ with the fastest growth of forecast error. On the other hand, the assimilation of AOD retrievals (either GOCI or MODIS) alone does not improve the surface analysis and mostly overestimates surface $PM_{2.5}$ for 24 h. This might be ascribed to an imperfection of the forward operator of AOD and the model deficiency in the representation of three-dimensional aerosol species that comprised AOD and $PM_{2.5}$. When assimilated with surface $PM_{2.5}$ observations (in "ALL"), however, AOD retrievals effectively reduce the forecast

error and suppress the error growth throughout 24 h forecasts.

     Recently, heavy pollution events have often taken place over Korea and considerable attention was drawn to the accuracy of the operational air quality forecasting in the country, particularly in surface $PM_{2.5}$. As it has a great social impact to accurately predict exceedance and non-exceedance events in categorical predictions, it is necessary to evaluate the forecast accuracy for different categorical events. While Miyazaki et al. (2019) classified the entire KORUS-AQ campaign period into four different

phases based on dominant atmospheric circulation patterns, we categorize events for the month of May 2016 based on hourly surface $PM_{2.5}$ concentrations, as summarized in Table 2 and 3. Figure 11 summarizes the evaluation of 24 h forecasts based on the formulae described below.

$$Overall\_Accuracy(\%) = \frac{a1 + b2 + c3 + d4}{N} \times 100 \tag{7}$$

$$High\_Pollution\_Accuracy(\%) = \frac{c3 + d4}{III + IV} \times 100 \tag{8}$$

$$Overestimation(\%) = \frac{b1 + c1 + c2 + d1 + d2 + d3}{N} \times 100 \tag{9}$$

$$Underestimation(\%) = \frac{a2 + a3 + a4 + b3 + b4 + c4}{N} \times 100 \tag{10}$$

$$False\_Alarm(\%) = \frac{II}{II + IV} \times 100 \tag{11}$$

$$Detection\_Rate(\%) = \frac{IV}{III + IV} \times 100 \tag{12}$$

where $I = a1 + a2 + b1 + b2$, $II = c1 + c2 + d1 + d2$, $III = a3 + a4 + b3 + b4$, and $IV = c3 + c4 + d3 + d4$. The air quality forecasting operated by the Korean NIER is currently evaluated in the same way on a daily basis, except for daily mean values.

In all events, the overall accuracy of 0-24-h forecasts is the highest in "ALL" ($\sim$70 %) and the lowest in "NODA" ($\sim$60 %), making about 10 % improvement by assimilation during this KORUS-AQ period. It is noted that the forecast error illustrated in Fig. 10 is dominated by days with clear sky or moderate air quality conditions (about two thirds of the month-long period, as shown in Fig. 6) while the forecast accuracy summarized in Fig. 11 is determined by equally weighting different categorical forecasts with different sample sizes. This implies that the categorical forecast evaluation tends to emphasize the forecast accuracy for pollution events (which has a smaller sample size). As such, Fig. 11a highlights the effect of data assimilation on improving air pollution forecasts. Differences between experiments are much larger in high pollution events (Fig. 11b) and the detection rate (Fig. 11f) where AOD retrievals (both GOCI and MODIS) make the biggest positive contributions. While "NODA" produces poor forecasts consistently in most metrics shown in Fig. 11, the forecast accuracy in "PM" (green) drops very quickly for the first 12 h for all events (a) and pollution events (b), indicating that the assimilation of surface $PM_{2.5}$ alone may not be enough to maintain the forecast skills beyond the cycling frequency (e.g. 6 h). It also increasingly underestimates surface $PM_{2.5}$ with time, especially after 20 h, and produces more false alarms even though its overestimation rate is the lowest among all experiments. Overall, the AOD assimilation tends to overestimate the prediction of surface $PM_{2.5}$ with a relatively large false alarm, but clearly helps enhance the forecast accuracy up to 24 h when assimilated with surface $PM_{2.5}$ observations. Even with low correlations with surface $PM_{2.5}$ (as illustrated in Fig. 5), AOD retrievals keep the surface air pollution forecasts from drifting away from the true state, compensating for model deficiencies. This demonstrates that it could be substantially beneficial to monitor a wide range of the surrounding area using the geostationary satellite in the enhancement of air quality forecasts.

In order to verify our forecasts against independent observations, we processed total AOD at 500 nm from the Aerosol Robotic Network (AERONET; https://aeronet.gsfc.nasa.gov/) sites and surface $PM_{2.5}$ concentrations measured at three more stations operated by NIER during the KORUS-AQ field campaign (Fig. 12). The level 2 quality level data are used for AERONET AOD observations as cloud-free and quality-assured data. Figure 13 illustrates the time series of hourly AOD from our experiments compared to hourly averages of AOD observations from 8 AERONET sites (black dots). At all sites, GOCI (blue) produces the largest AOD at most of high peaks while PM (green) and NODA (gray) simulate the smallest AOD throughout the period. Regardless of relative AOD values between the experiments, model forecasts are well matched with observations at low AOD values, but mostly miss high AOD observations, especially during the high pollution events for May 24-27. This leads to the negative mean bias (as (f-o)'s) in all experiments (shown in the legend), implying that our forecasts produce AOD slightly lower than the observed one as a whole. The rms error and mean bias at total of 16 AERONET sites are summarized in Table 4, indicating that GOCI has the smallest forecast error in AOD nation-wide.

Surface $PM_{2.5}$ measurements from three NIER sites were downloaded from https://www-air.larc.nasa.gov/cgi-bin/ArcView/korusaq as raw data with no quality control. They are provided as hourly averages starting from May 9 and compared to our hourly model output for May 9 - 31 (Fig. 14). These observations look somewhat noisy, but our forecasts broadly follow them through-out the period. Similar to the AOD verification shown in Fig. 13, forecasts from GOCI produce the smallest forecast mean bias

among all the experiments in Olympic Park (in a) and Daejeon (in b), predicting the high surface PM$_{2.5}$ concentrations between May 24 and 26. But GOCI was worse than other experiments in Ulsan (in c), overestimating surface PM$_{2.5}$, especially during the high pollution days. In the assimilation system, raw data are not considered to be reliable, but this verification is included for the completeness because there was no other instrument that reported surface PM$_{2.5}$ concentrations or all the precursors of

PM$_{2.5}$ concentrations to validate PM$_{2.5}$ forecasts on the ground level.

### 4.3   A heavy pollution case

The effect of assimilating different observations is most distinguishable in high pollution events, as demonstrated in Fig. 11. During the KORUS-AQ period, there were about 5 heavy pollution cases (when surface PM$_{2.5}$ > 50 $\mu$g/m$^3$, as defined in Table 2) over South Korea. The longest and the most severe pollution events have occurred on May 25-26, 2016. Figure 15 illustrates

how air pollutants have been transported from China, associated with the strong synoptic weather systems in the region for a few days. As the analysis of our best experiment "ALL" showed, the Korean peninsula was positioned in the downstream region of the upper-level trough at 500 hPa (in the left panel). In the low troposphere, the center of the North Pacific High was situated in the east of Japan bringing lots of moisture to Korea at 00 UTC 24 May 2016 and blocking the eastward movement of the surface low pressure system located north of Korea (centered around 46°N, 125°E), as shown in Fig. 15d. With the slowly

approaching upper-level westerlies, these warm and moist conditions in the low troposphere provided a favorable environment for increasing air pollution in the Korean peninsula for the next few days. At 00 UTC 25 May, the Shangdong area in China (shown as the largest polluted area to the west of Korea) exceeded 150 $\mu$g/m$^3$ in surface PM$_{2.5}$ (Fig. 15b). This area is in high topography with elevations higher than 3.5 km (in height above ground level; AGL) while most regions in South Korea, especially the Seoul Metropolitan Area (SMA), are elevated near sea level. Therefore, when slow and deep baroclinic systems

are approaching the Korean peninsula like these events, a deep pool of highly polluted air can be advected from China as a whole to substantially degrade the air quality in South Korea at least for a day or two. This long-range transport case produced an hourly maximum surface PM$_{2.5}$ observation of 117 $\mu$g/m$^3$ over the SMA in Korea at 00 UTC May 26, 2016, as shown in Fig. 16a.

One notable difference between observations (a) and all the model simulations (b-f) in Fig. 16 is that 9-km forecasts driven

by 0.1° x 0.1° anthropogenic emissions cannot simulate such a high spatial variability across stations. During this heavy pollution event, there were dozens of missing observations to have a less number of stations in a) than all the experiments (b-f). With only 145 stations reporting high concentrations (e.g. surface PM$_{2.5}$ > 50 $\mu$g/m$^3$), the observed distribution still shows a sharp gradient between the stations, specially in SMA. Consistent with all the previous figures, the assimilation of surface PM$_{2.5}$ alone (in "PM") underpredicts surface PM$_{2.5}$ (even more than "NODA") while GOCI overpredicts surface PM$_{2.5}$ most

among all observation types almost everywhere except for SMA. MODIS retrievals slightly increase the concentrations from NODA (by ~10 $\mu$g/m$^3$), with the spatial distribution almost the same as that of NODA. In the concurrent assimilation of all the observations (in "ALL"), a moderate overestimation is presented everywhere, but higher levels of pollution in SMA are not simulated either. To resolve such a large variability between urban and rural area and to increase the sharpness of the

forecast accuracy, the use of higher grid resolutions (such as 3 km), more accurate emission data and more sophisticated aerosol chemistry mechanisms might be indispensable.

## 5   Conclusions and discussion

GOCI AOD retrievals provide reliable and consistent aerosol information, monitoring air pollutants over the Korean peninsula
at high resolution every day. One of the best ways of utilizing such invaluable observations is to inject them into the forecast system through data assimilation and better initialize numerical forecasts. For the successful assimilation of real observations, specially retrievals from satellites, extra attention should be paid to processing the data properly, based on the characteristics. The spatial and temporal representativeness of GOCI retrievals was carefully examined and the corresponding data processing was conducted before assimilation in this study. We averaged all the pixels over each grid box at 27-km resolution (e.g.
superobing) instead of thinning them randomly, for instance.

It is worth noting several challenges in the assimilation of AOD retrievals for improving the prediction of surface $PM_{2.5}$ concentrations: i) AOD is not directly associated with $PM_{2.5}$ concentration on the ground. Although the two datasets can be highly correlated in specific conditions such as cloud-free, low boundary layer heights and low relative humidity, the overall correlation is low ($\sim$0.3) in the present study and it is hard to expect the direct impact on each other. ii) an observation operator
for AOD has errors due to the simplification and the limited aerosol specifications in the community radiative transfer model (CRTM). iii) significant model error, which is presumably one of the most critical issues. In the 3DVAR assimilation, in particular, the model estimates of AOD, a column-integrated quantity, are strongly constrained by the model error structure of each aerosol species both horizontally and vertically.

Even with these challenges, however, satellite-based AOD, especially from geostationary satellites like GOCI, can be ex-
tremely useful for improving the prediction of air pollution on a daily basis. In the situation where the air quality can be largely affected by long-range transport of air pollutants, such consistent information on the wide upstream area is essential but hard to be obtained otherwise.

Using the GSI 3DVAR system coupled with the WRF-Chem forecast model, we assimilated the satellite AOD retrievals as well as surface $PM_{2.5}$ observations for the month of May 2016 during the KORUS-AQ period. Compared to the baseline
experiment ("NODA"), the simultaneous assimilation of various observations consistently improved the prediction of ground $PM_{2.5}$ for 24-h forecasts, reducing systematic error and false alarms. The assimilation of ground $PM_{2.5}$ alone improved the analysis during the cycles, reducing the analysis error to almost half the size compared to the experiment without assimilation. However, the forecast error grew very quickly over the next 12 hours, underestimating $PM_{2.5}$ at the surface, especially in the heavy pollution events where the forecast accuracy dropped from over 70% to $\sim$30% only in four hours. Meanwhile, the GOCI
AOD retrievals alone tended to overestimate surface $PM_{2.5}$ but significantly contributed to improving air quality forecasts up to 24 h when assimilated with surface $PM_{2.5}$ observations. The effect of data assimilation is most distinguishable and remarkable for high pollution events. During the month of May 2016, most heavy pollution events were associated with long-

range transport from China. In such cases, it was particularly beneficial to monitor the wide upstream region using geostationary instruments such as GOCI.

To assess the effect of data assimilation with respect to independent observations, 0-23 h forecasts from different experiments are verified against AOD from AERONET sites and ground $PM_{2.5}$ measurements from the sites operated during the KORUS-AQ field campaign. In this verification, the assimilation of GOCI retrievals is the most effective in improving the forecast performance at most sites, especially for high pollution events.

Even with the successful data assimilation, there are several limitations in this study. First, the simple GOCART aerosol scheme is well known for the underestimation of air pollutants due to the lack of the aerosol size distribution and the secondary organic aerosol (SOA) formation. We had to use the scheme for the assimilation of AOD retrievals since the observation operator for AOD was only built for the GOCART scheme in the GSI system. Next, as there is no cross-covariance between aerosol and meteorological variables considered in the background error covariance estimates, the influence of aerosols on meteorological variables was not fully simulated in this study. Without the assimilation of meteorological observations, it was not possible to make an optimal estimate that is fully coupled between chemistry and meteorology although the meteorological information was provided through the first guess and lateral boundary conditions. Finally, the emission inventory used in this study was based on the annual mean of 2010, which did not reflect the actual emissions for the year of 2016, especially over China. The large bias and uncertainties in the emission data was particularly detrimental to the assimilation of surface $PM_{2.5}$ alone.

To overcome the systematic underestimation of the GOCART aerosol scheme in the assimilation context, there is an ongoing effort for a new development of an interface for more sophisticated aerosol schemes such as MOSAIC and/or the Modal for Aerosol Dynamics in Europe and the Volatility Basis Set (MADE/VBS; Ackermann et al. (1998), Ahmadov et al. (2012)) in the WRFDA system (Barker et al., 2012). This would be advantageous for more realistic forecast behavior in high resolution applications.

The positive impact of data assimilation is generally limited to 24-h forecast because of three major reasons: First, most air pollutants have a short lifetime due to dry and wet deposition and transformations through interactions with solar radiation and clouds. Secondly, pollutant transport and transformations in chemical transport models are strongly driven by external forcing, such as emissions, boundary conditions, and meteorological fields. Lastly, there are large uncertainties in aerosol and gas-phase chemistry parameterized in chemical transport models. Therefore, to extend the period of forecast improvements, emission data needs to be improved and large uncertainties in chemical and meteorological boundary conditions should be minimized. It has been shown that the estimation of emission inventories as part of the DA procedure can help extend the impact of data assimilation in longer forecasts (Elbern et al., 2007; Kumar et al., 2019). Also, more sophisticated aerosol and chemical mechanisms might be able to improve air quality forecasting by reducing model deficiencies (Chen et al., 2019). A simultaneous assimilation of meteorological observations and measurements of individual chemical species as well as particulate matter would be certainly beneficial in both NWP and air quality forecasting. To better account for high nonlinearities and uncertainties of aerosol forecasting on small scales, more advanced analysis techniques such as ensemble or hybrid data assimilation would be more desirable.

*Code and data availability.* The WRF-Chem v3.9.1 and the GSI v3.5 codes used in this paper are publicly available in the website https://www2.mmm.uca and https://dtcenter.org/com-GSI/users/downloads/index.php, respectively. Input observations and boundary conditions for a sample test period can be also provided upon request.

*Author contributions.* ZL helped formulating the study and WS performed initial test runs. YL and LC provided input datasets, partially
5   funding this study. SH designed and ran the experiments, analyzed the results and wrote the manuscript.

*Competing interests.* The authors declare that they have no conflict of interest.

*Acknowledgements.* All the experiments presented here were performed on the Cheyenne supercomputer at the National Center for Atmospheric Research (NCAR). This work was jointly supported by the National Science Foundation under Grant No. M0856145 and the grant from the National Institute of Environment Research (NIER), funded by the Ministry of Environment (MOE) of the Republic of Korea
10   (NIER-SP2018-252). We acknowledge use of the WRF-Chem preprocessor tool (mozbc, fire_emiss, megan_bio_emiss, and anthro_emiss) provided by the Atmospheric Chemistry Observations and Modeling Lab (ACOM) of NCAR. Authors are also thankful for Seunghee Lee, Ganghan Kim and Myong-In Lee at UNIST in South Korea for their help transferring the input data for our experiments. Dave Gill and Wei Wang at MMM/NCAR helped us processing data in WPS and tuning the WRF configuration, respectively. Finally, special thanks should go to Gabriele Pfister at ACOM/NCAR and Dan Chen at China Meteorological Administration for their internal review, which greatly improved
15   the manuscript.

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

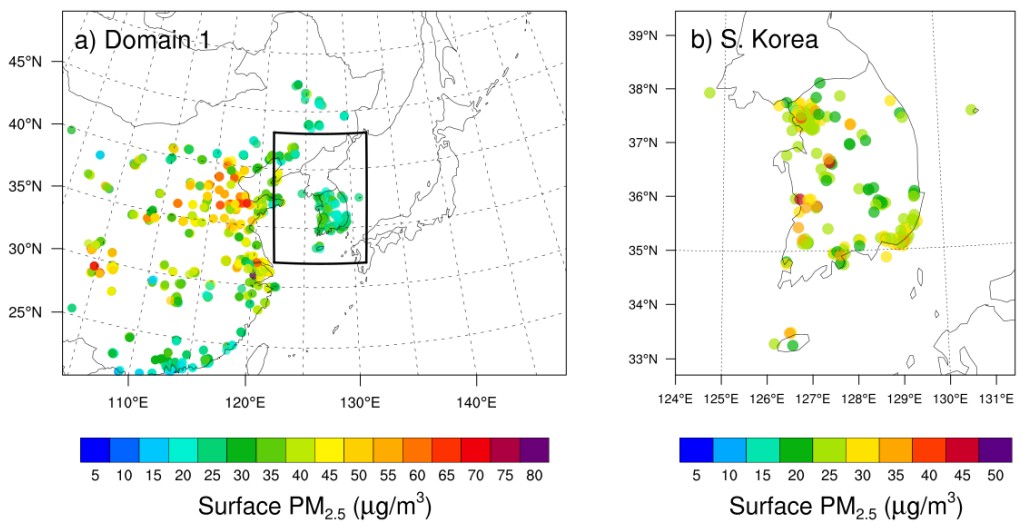

**Figure 1.** Surface observation network with 960 Chinese stations and 361 Korean stations in domain 1 (a) and zoomed in over South Korea in (b). A black box in a) indicates domain 2 over the Korean peninsula. Dots indicate surface $PM_{2.5}$ observations averaged over the month of May 2016.

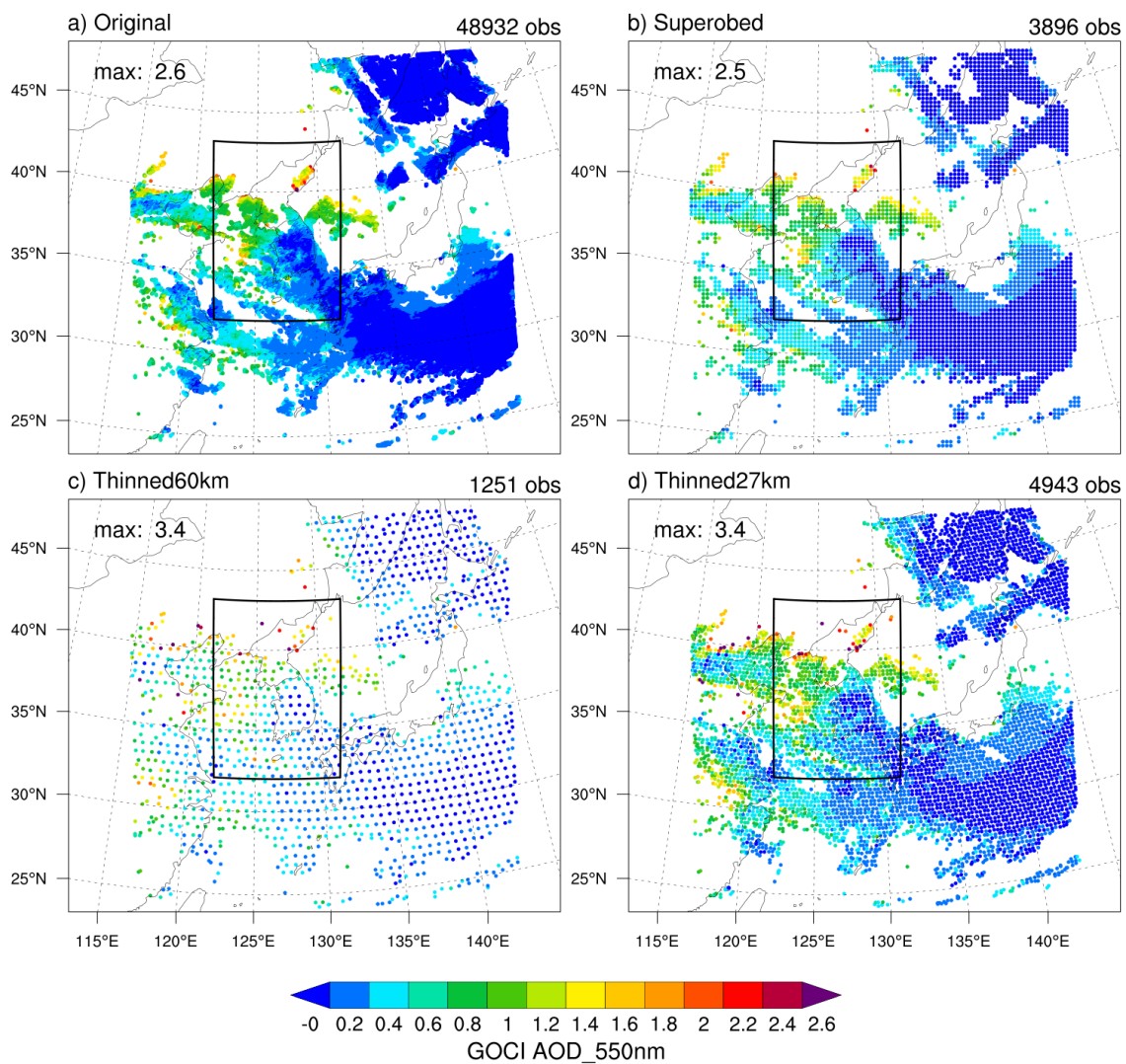

**Figure 2.** Horizontal distribution of GOCI AOD at 550 nm retrieved at 2016-05-01_06:00:00 UTC in a) the original Level II data at 6 km resolution b) the preprocessed at 27 km resolution before GSI, and the data thinned over c) 60 km and d) 27 km resolution during the GSI analysis, respectively. A total number of observations available for the GSI analysis is shown in the upper right corner of each panel and the maximum value in the upper left corner of each map. Domain 2 is marked as a black box in each panel.

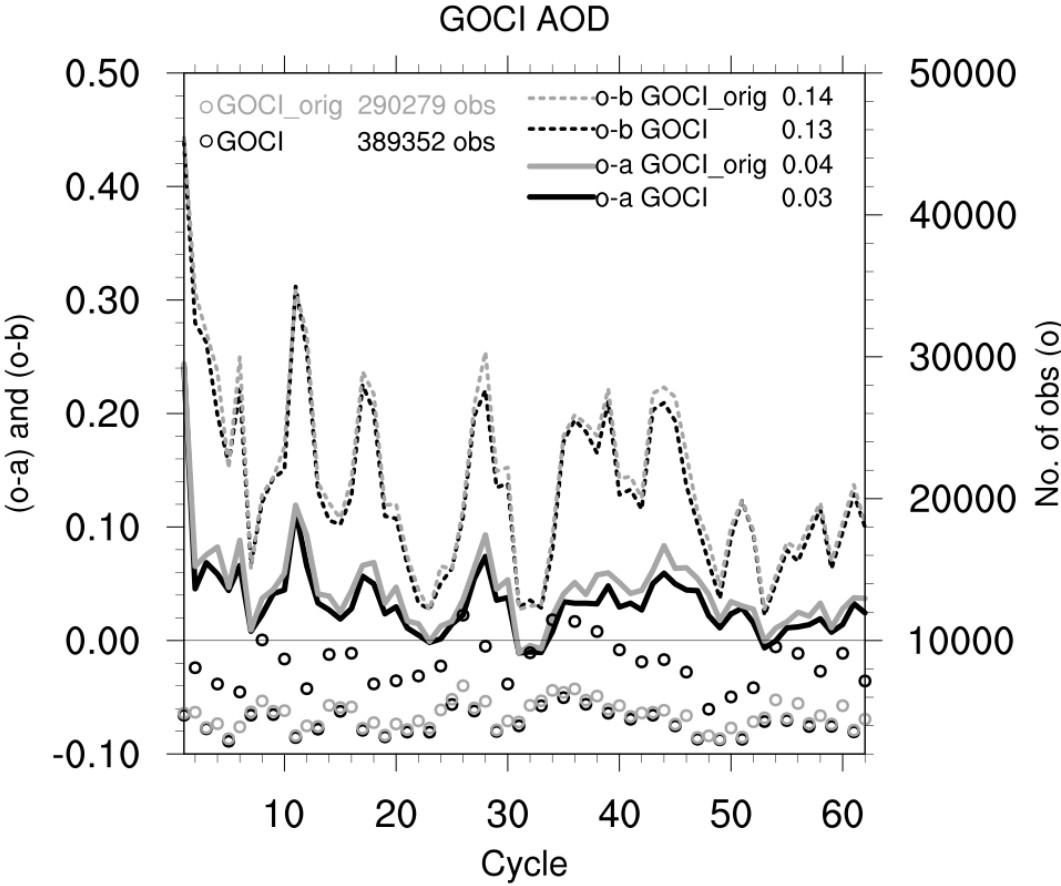

**Figure 3.** Time series of observation-minus-analysis (o-a; solid lines) and observation-minus-background (o-b; dotted) with respect to GOCI AOD retrievals at 550 nm for two cycling experiments over domain 1. The "GOCI_orig" experiment assimilates the original data thinned over 27-km mesh (in gray) while the "GOCI" experiment assimilates GOCI retrievals averaged over 27-km grids in domain 1 (black). Cycle-mean values are displayed next to each component. Total number of observations assimilated in each experiment at each cycle is also plotted as "o" sign on the right y-axis ranging from 2,000 to 12,000.

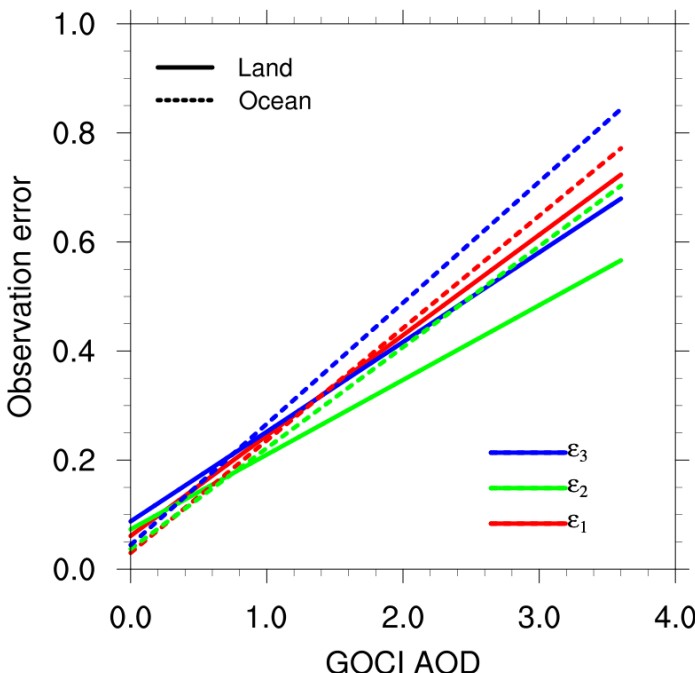

**Figure 4.** Three different types of observation errors ($\epsilon$) applied to GOCI AOD retrievals over land (solid line) and ocean (dashed line), respectively. The first two errors ($\epsilon_1$ and $\epsilon_2$) are described in equations (3) - (6) and the third error ($\epsilon_3$) increases $\epsilon_2$ by 20% everywhere.

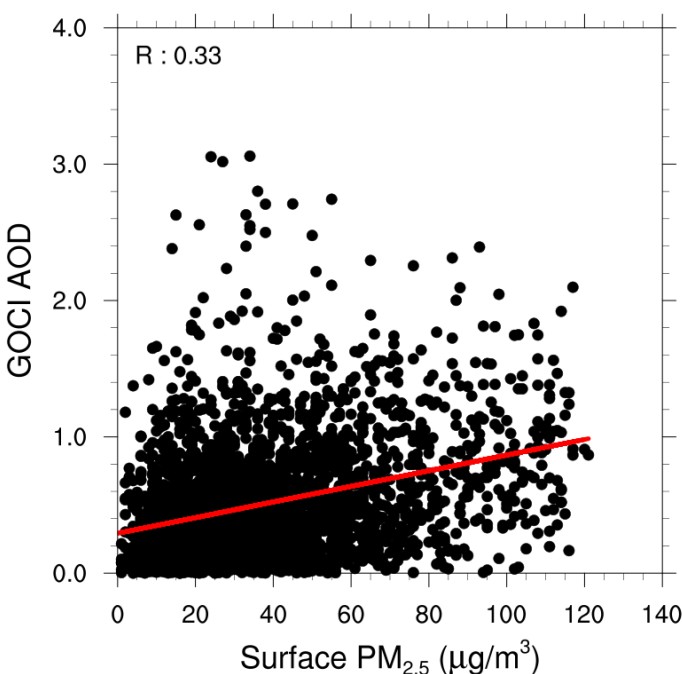

**Figure 5.** Scatter plots of GOCI AOD retrievals versus ground PM$_{2.5}$ observations collocated in domain 1 for the month of May 2016. The value of R is the correlation coefficient between the two observation types based on the linear regression shown as the red line.

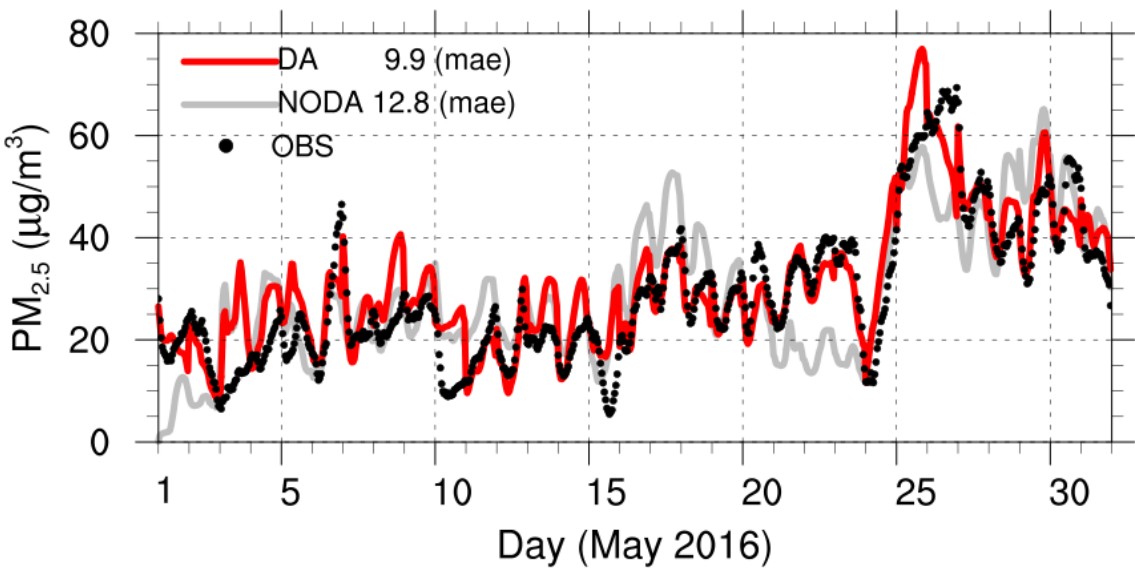

**Figure 6.** Time series of surface PM$_{2.5}$ simulated with (DA; red) and without assimilation (NODA; gray) in domain 2, representing hourly 0-23-h forecasts from 00Z every day, as averages over 361 stations over South Korea. Corresponding observations are marked as black dots. The mean absolute error (mae; $|o - f|$) averaged over the entire period is shown for each experiment. Here, "DA" refers to the "ALL" experiment.

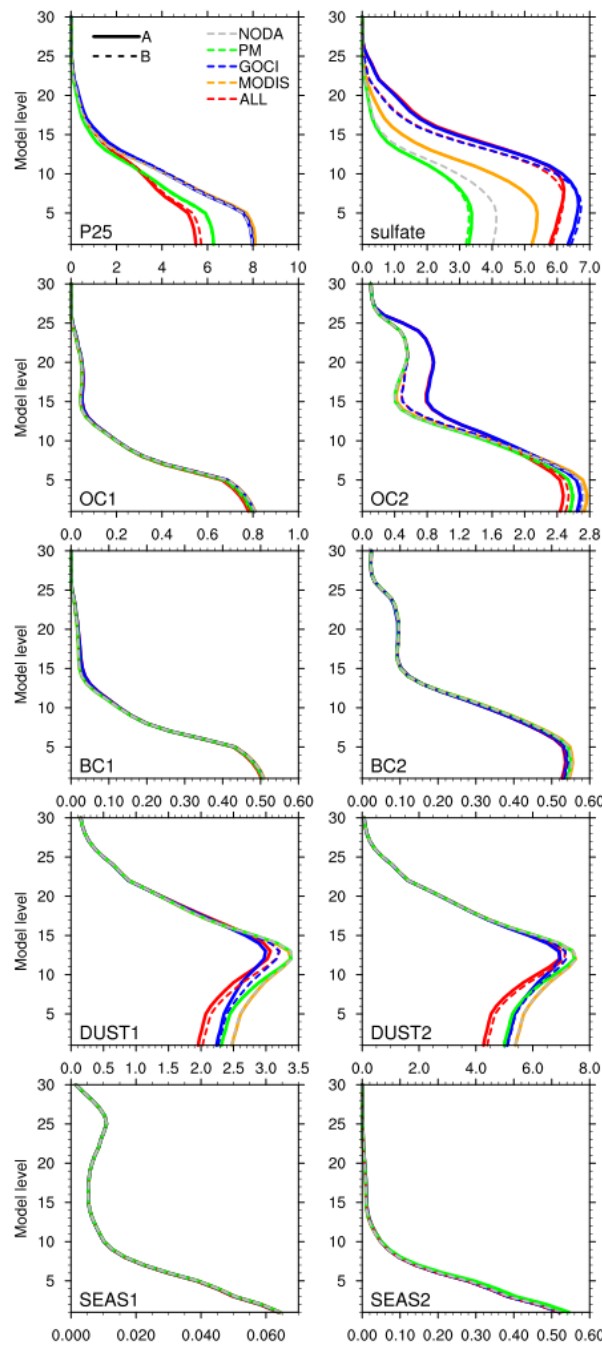

**Figure 7.** Vertical profile of 10 GOCART aerosol variables composed of PM$_{2.5}$ - unspeciated aerosol contributions to PM$_{2.5}$ (P25), sulfate, OC1 and OC2 (BC1 and BC2) as hydrophobic and hydrophilic organic (black) carbon, respectively, DUST1 and DUST2 (SEAS1 and SEAS2) as dust (sea salt) aerosols in the smallest and 2nd smallest size bins. All the variables shown are mixing ratios in the unit of $\mu$g/kg. Different experiments are depicted in different colors, as averaged over domain 2 for the period of May 4 - 31, 2016. Analysis ("A") is drawn as solid line while background (e.g. 6-h forecast; "B") as dashed line.

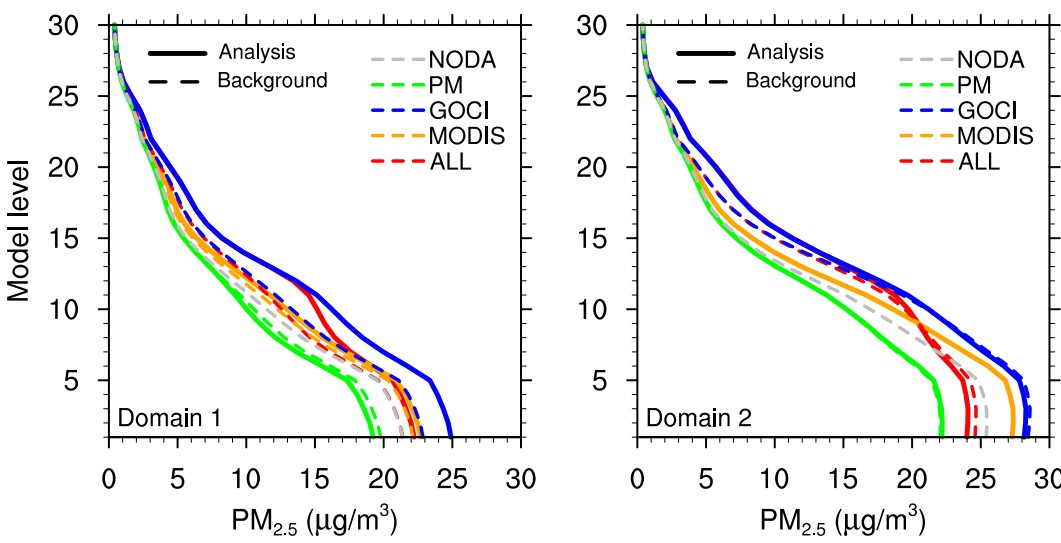

**Figure 8.** Same as Figure 7, except for PM$_{2.5}$ in both domains.     **30**

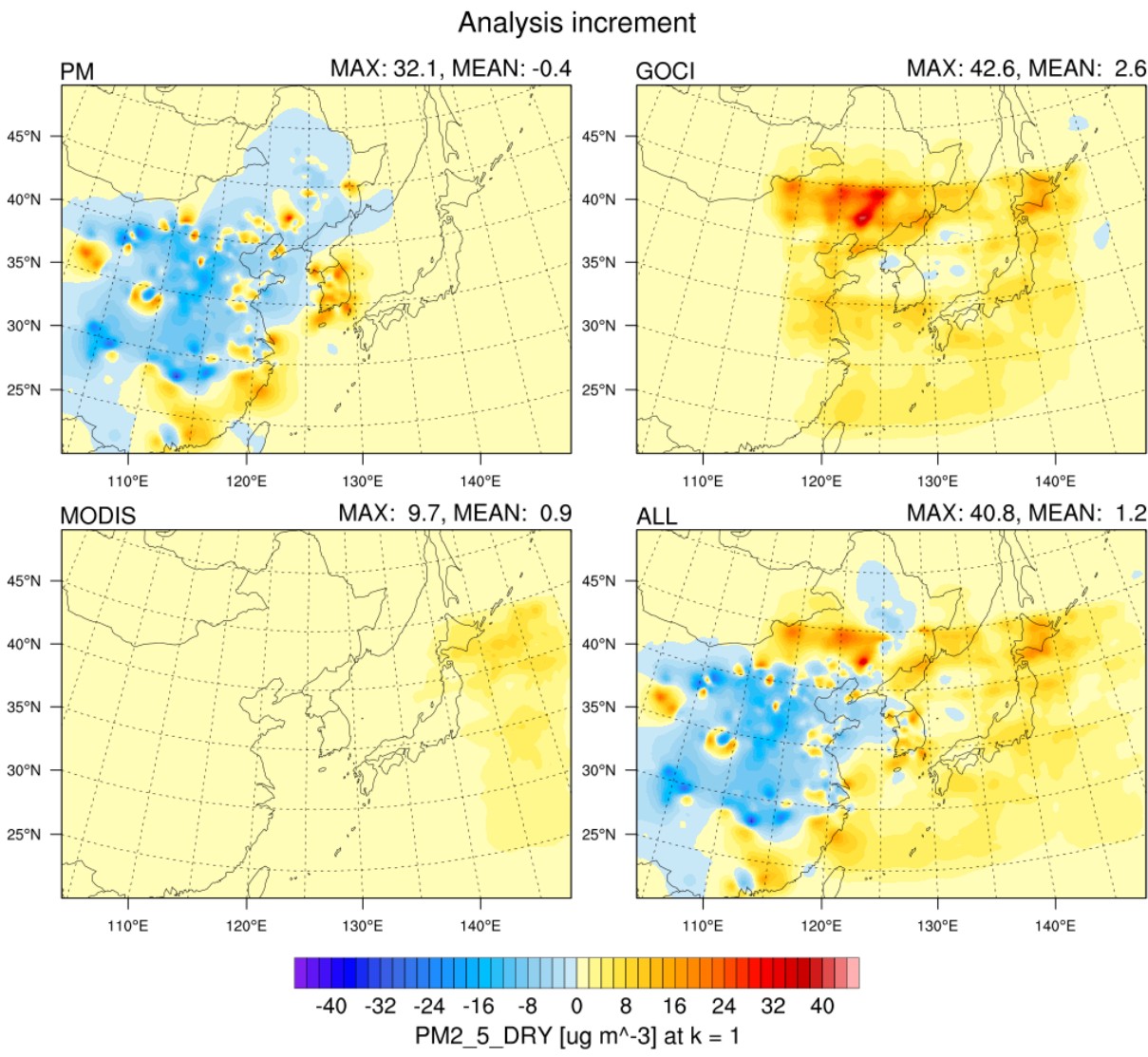

**Figure 9.** Horizontal distribution of analysis increments (analysis-minus-background) in PM2_5_DRY, the model variable corresponding to PM$_{2.5}$, at the lowest level in domain 1, averaged over the period of May 4 - 31, 2016. Maximum and mean values of the domain in each experiment are shown in the upper right corner of each panel.

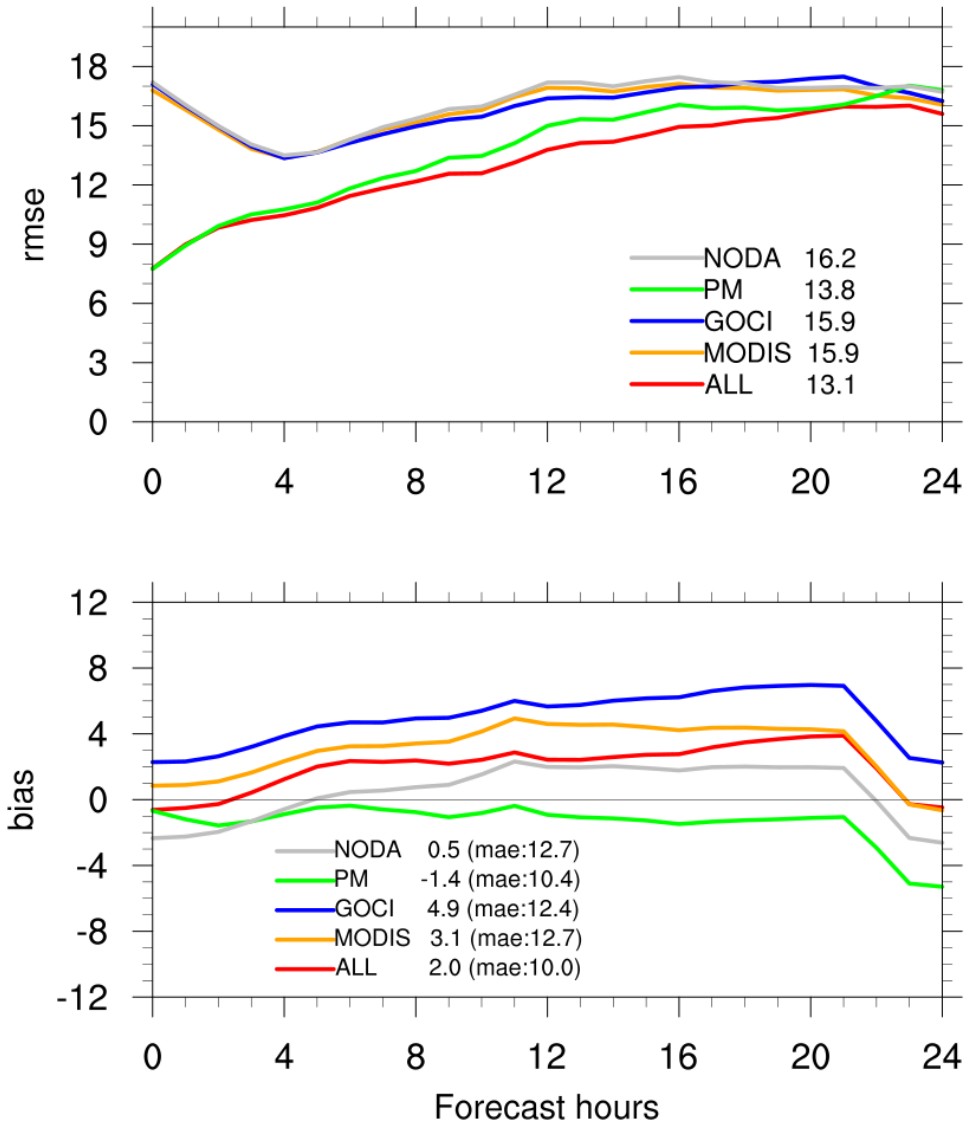

**Figure 10.** Time series of root-mean-square-error (rmse; upper panel) and bias (lower panel) of the hourly forecasts from the 00 Z initialization for May 4 - 31, 2016. Different experiments in domain 2 are verified against surface PM$_{2.5}$ observations from 361 stations in South Korea. An average of 0-24 h forecast errors is shown next to each experiment name. The mean absolute error (mae) over the 24-h forecasts is also shown in the lower panel.

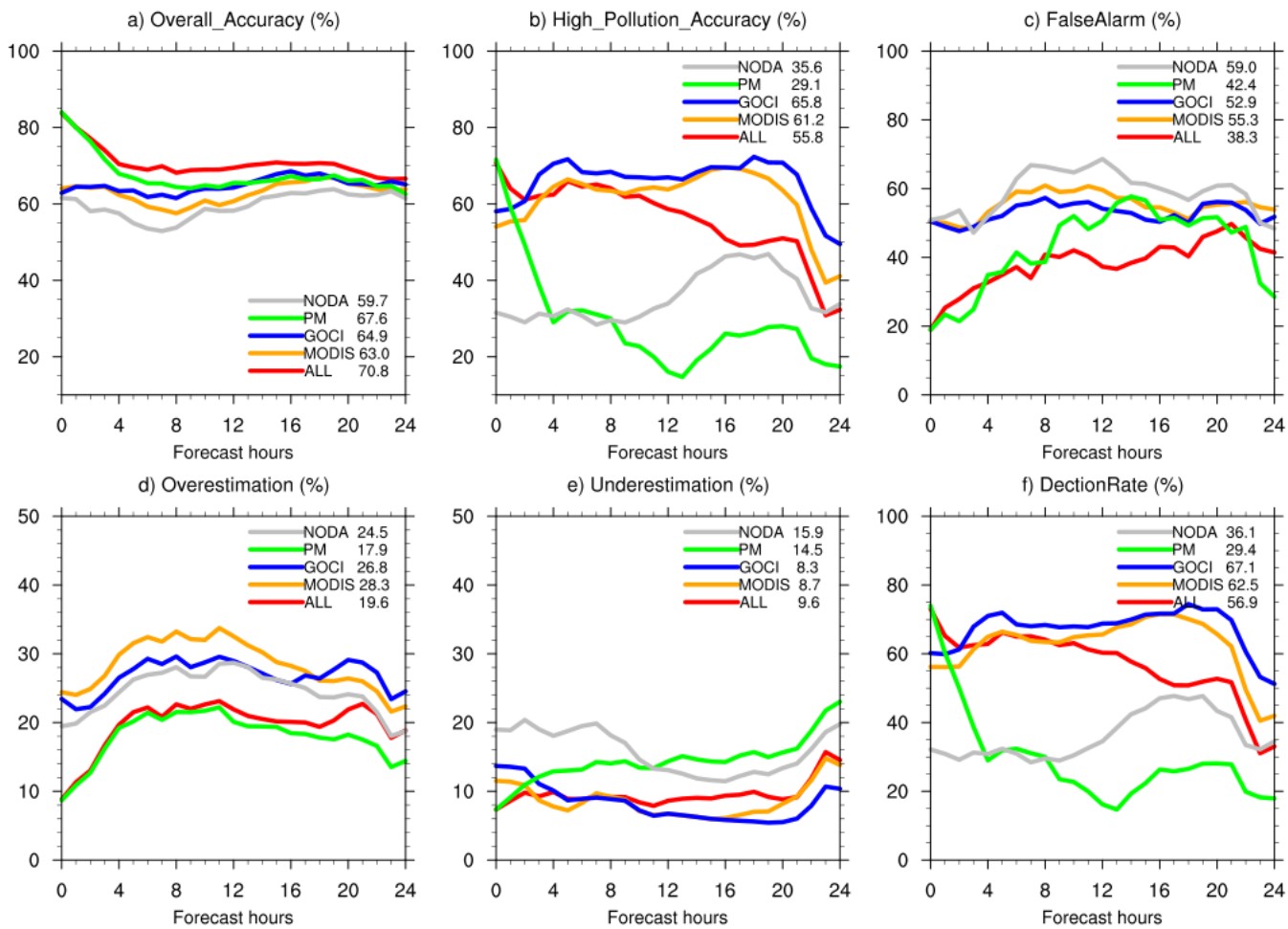

**Figure 11.** Time series of forecast accuracy (%) of the hourly forecasts from the 00 Z initialization for May 4 - 31, 2016 in domain 2 for categorized events based on hourly surface PM$_{2.5}$ concentrations, as defined in Tables 2 and 3.

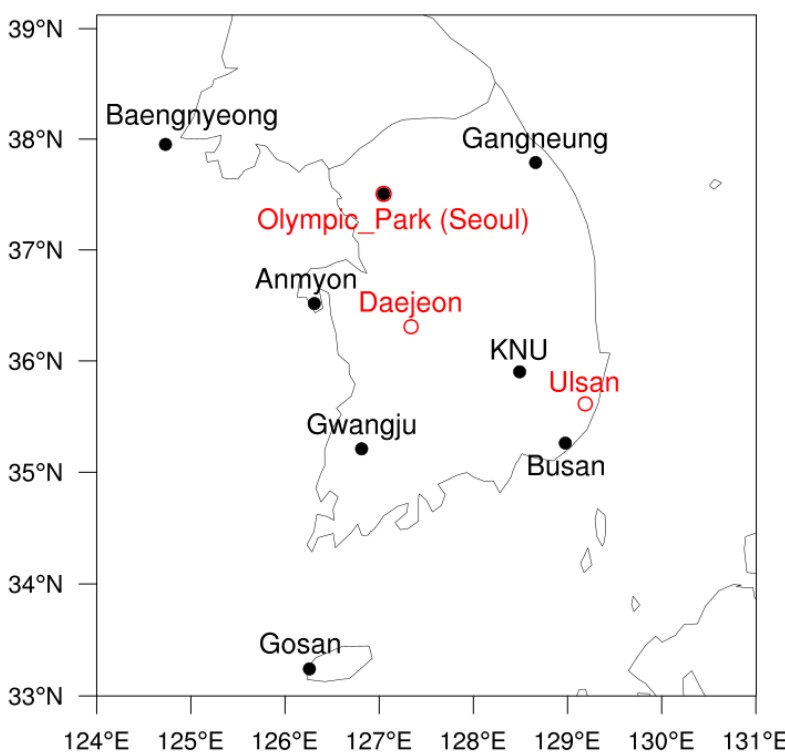

**Figure 12.** Map of AERONET sites (black dots) used for verification shown in Fig. 13. Three red open dots are the stations operated by NIER to measure surface PM$_{2.5}$ concentrations during the KORUS-AQ field campaign, which are used in the verification illustrated in Fig. 14.

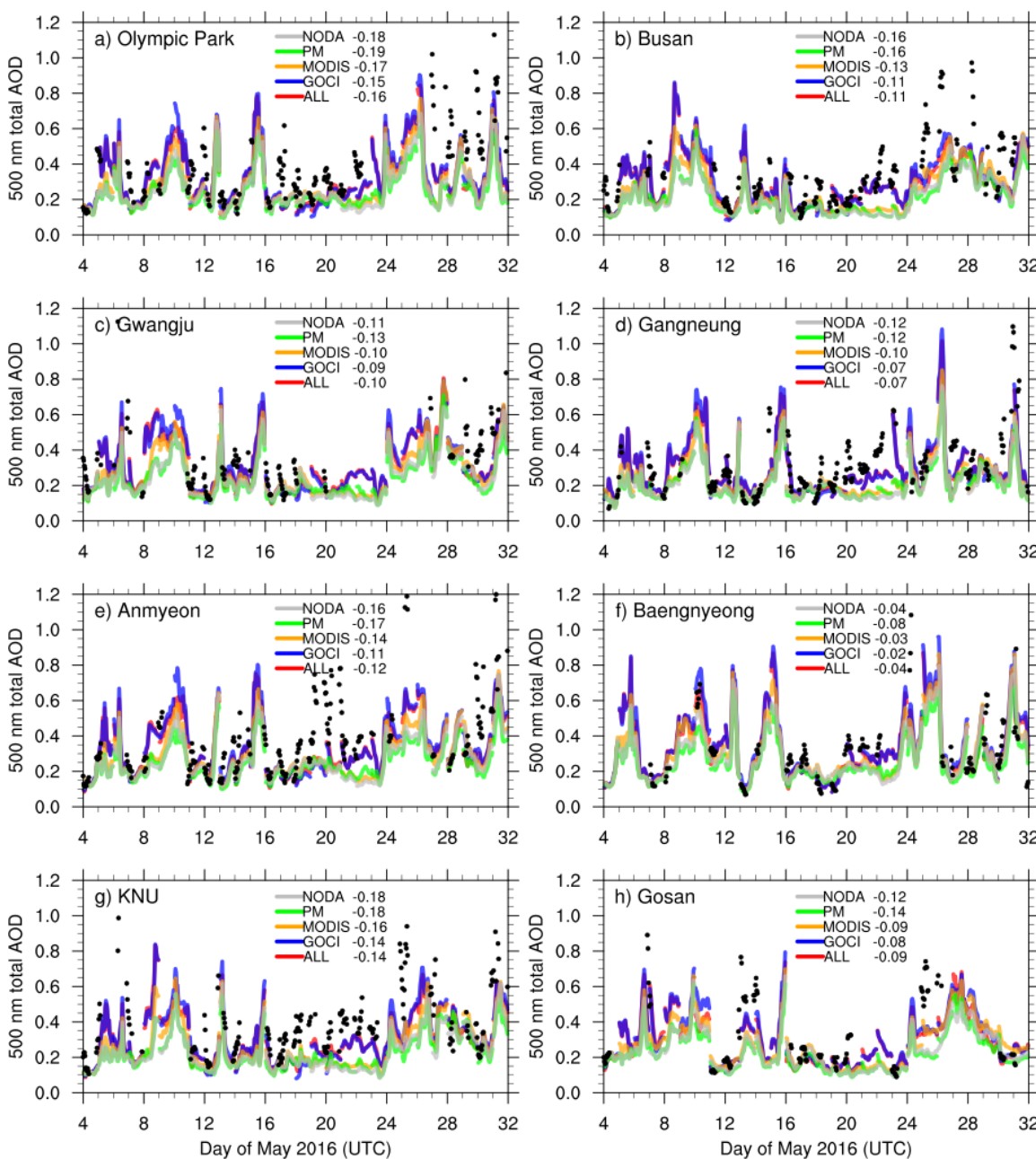

**Figure 13.** Hourly time series of total AOD at 500 nm from 0000 UTC 04 May to 2300 UTC 31 May at 8 different AERONET sites. Model values in different colors represent output every hour beginning at the initial time and ending at the 23rd hour of integration patched together for each 0000 UTC forecast. The bias (as (f-o)'s) averaged over the entire period is shown next to each experiment name. AERONET observations represent hourly averages as black dots.

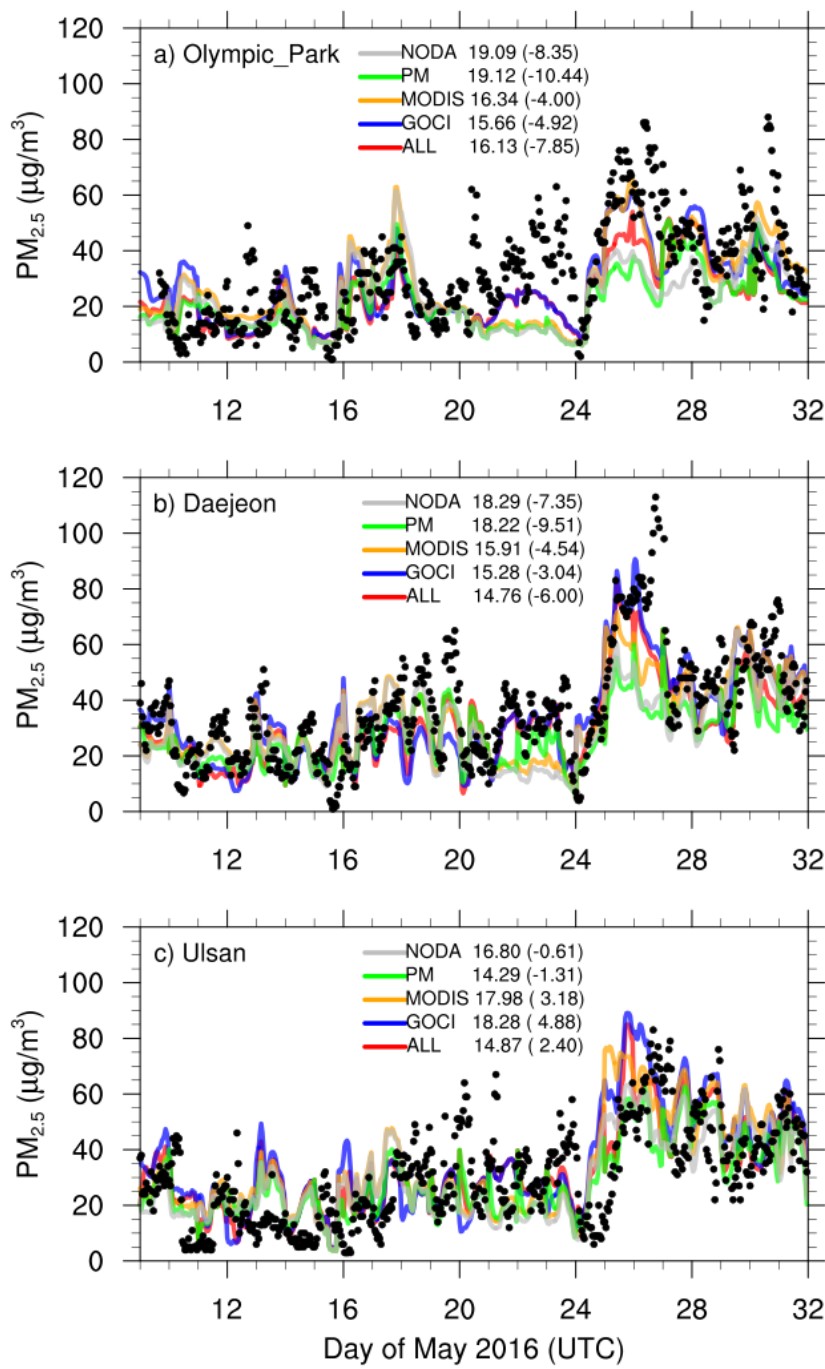

**Figure 14.** Same as Fig. 13, but for surface PM$_{2.5}$ concentrations from 0000 UTC 09 May to 2300 UTC 31 May at a) Olympic Park in Seoul b) Daejeon and c) Ulsan. The sites are marked as red open dots in Fig. 12. The rmse over the whole period is written next to each experiment name, along with the mean bias (as (f-o)'s) in the parenthesis.

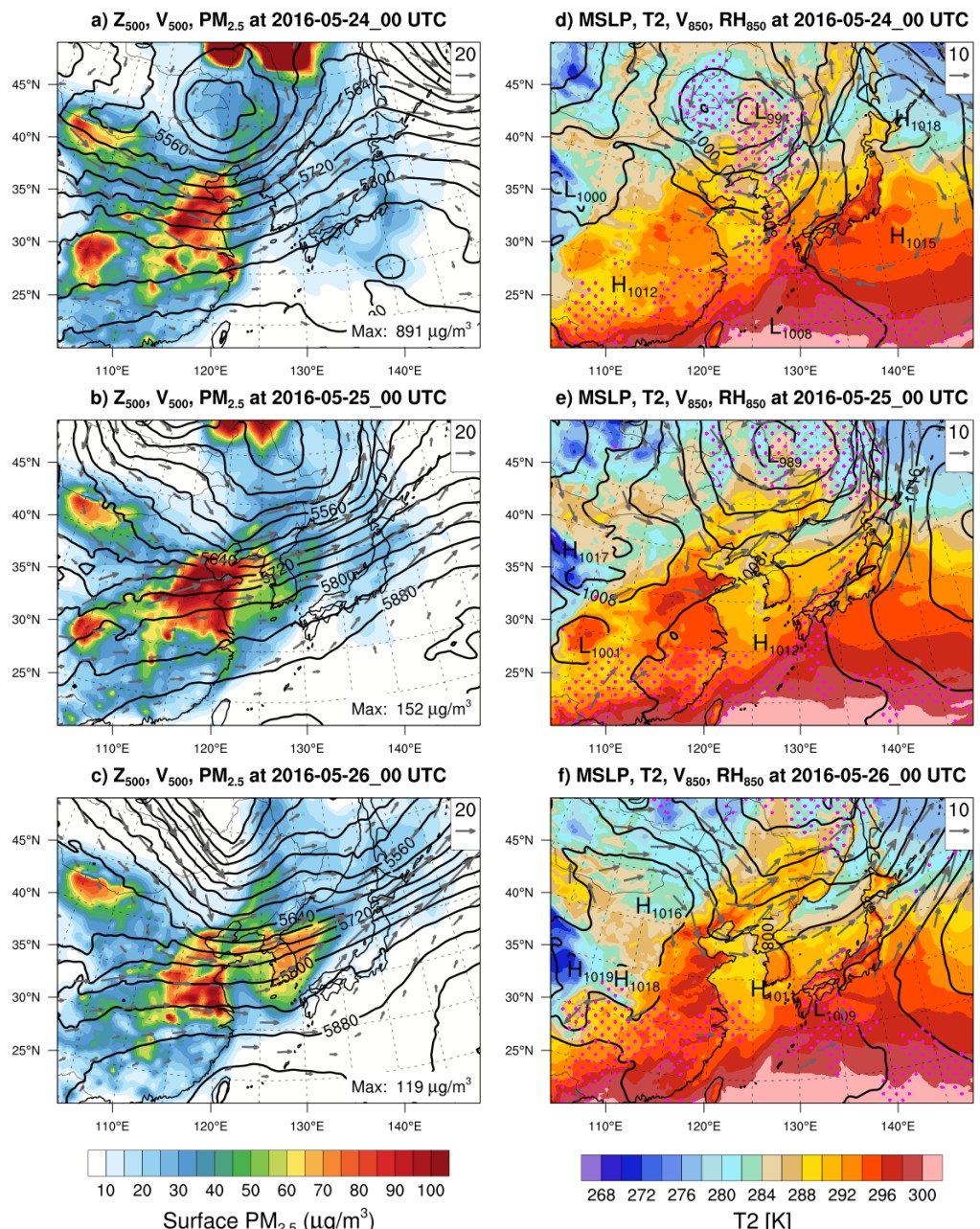

**Figure 15.** The GSI 3DVAR analyses at 27-km resolution in domain 1 in the "ALL" experiment for three days from 24 to 26 May 2016 at 00 UTC (top to bottom). In the left panel, the horizontal distribution of surface $PM_{2.5}$ ([$\mu g/m^3$], filled), geopotential height (contours every 40 m) and horizontal winds ([m/s] in gray vectors) at 500 hPa illustrates that the long-range transport of air pollution from China causes the heavy pollution over South Korea. In the right panel, mean sea level pressure (contours every 4 hPa), 2-m temperature ([K], filled), relative humidity (> 90% in pink dots) and horizontal winds ([m/s] in gray vectors) at 850 hPa represent the weather system in the low troposphere at the same time.

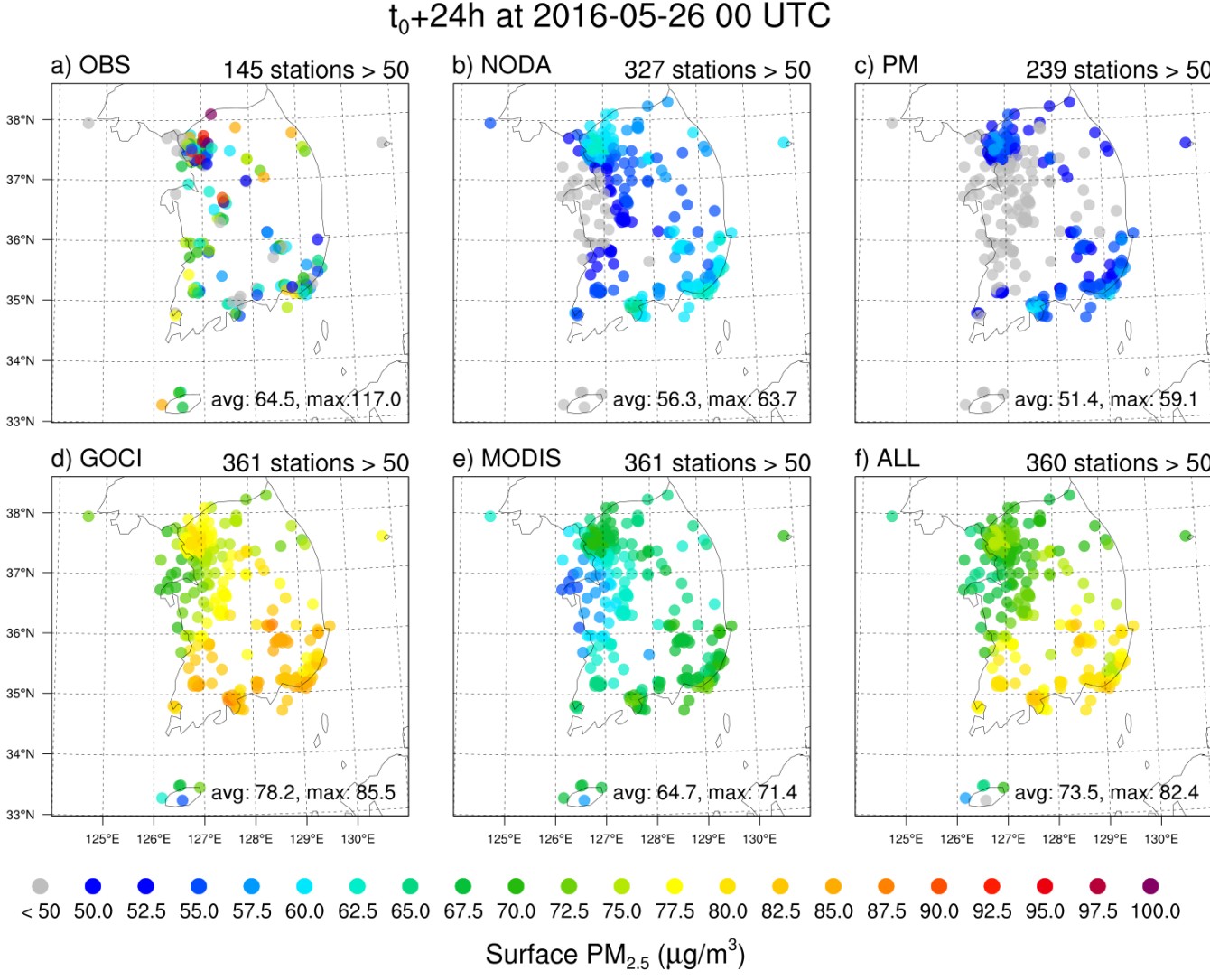

**Figure 16.** Horizontal distribution of 24-h forecast in 9-km simulations in $PM_{2.5}$ at the lowest level in each experiment compared to a) observations from 361 stations in South Korea valid at 00 UTC 26 May 2016.

**Table 1.** Physical and chemical parameterizations used in the experiments.

| Physical processes | Parameterization schemes |
|---|---|
| Aerosol chemistry | GOCART |
| Gas-phase chemistry | MOZART-4 |
| Photolysis | Fast-TUV |
| Cloud microphysics | Lin |
| Cumulus | Grell 3D ensemble |
| Longwave radiation | RRTMG |
| Shortwave radiation | Goddard |
| PBL | YSU |
| Surface layer | Monin-Obukhov |
| Land surface | Noah |

**Table 2.** Air quality index values.

| Concentration ($\mu$g/m$^3$, hourly) | Good | Moderate | Unhealthy | Very Unhealthy |
|:---:|:---:|:---:|:---:|:---:|
| PM$_{2.5}$ | 0-15 | 16-50 | 51-100 | > 100 |

**Table 3.** Categorical forecasts for different air pollution events.

| Category | | Forecast | | | |
|---|---|---|---|---|---|
| | | Good | Moderate | Unhealthy | Very Unhealthy |
| Observation | Good | a1 | b1 | c1 | d1 |
| | Moderate | a2 | b2 | c2 | d2 |
| | Unhealthy | a3 | b3 | c3 | d3 |
| | Very Unhealthy | a4 | b4 | c4 | d4 |

**Table 4.** Forecast error in total AOD at 500 nm verified against AERONET sites, computed over 0-23 h forecasts from 00Z analysis for May 4 - 31.

| | rmse | | | | | bias | | | | |
|---|---|---|---|---|---|---|---|---|---|---|
| | NODA | PM | MODIS | GOCI | ALL | NODA | PM | MODIS | GOCI | All |
| OlympicPark | 0.26 | 0.27 | 0.25 | 0.23 | 0.24 | -0.18 | -0.19 | -0.17 | -0.15 | -0.16 |
| Busan | 0.22 | 0.22 | 0.19 | 0.17 | 0.18 | -0.16 | -0.16 | -0.13 | -0.11 | -0.11 |
| Gwangju | 0.18 | 0.19 | 0.17 | 0.16 | 0.16 | -0.11 | -0.13 | -0.1 | -0.09 | -0.1 |
| Gangneung | 0.18 | 0.18 | 0.17 | 0.13 | 0.13 | -0.12 | -0.12 | -0.1 | -0.07 | -0.07 |
| Anmyeon | 0.26 | 0.27 | 0.24 | 0.22 | 0.22 | -0.16 | -0.17 | -0.14 | -0.11 | -0.12 |
| Baengnyeong | 0.15 | 0.15 | 0.14 | 0.13 | 0.13 | -0.04 | -0.08 | -0.03 | -0.02 | -0.04 |
| KNU | 0.24 | 0.25 | 0.22 | 0.2 | 0.21 | -0.18 | -0.18 | -0.16 | -0.14 | -0.14 |
| Gosan | 0.21 | 0.21 | 0.18 | 0.15 | 0.15 | -0.12 | -0.14 | -0.09 | -0.08 | -0.09 |
| Seoul_SNU | 0.22 | 0.23 | 0.21 | 0.2 | 0.2 | -0.14 | -0.16 | -0.13 | -0.12 | -0.12 |
| NIER | 0.21 | 0.21 | 0.2 | 0.19 | 0.19 | -0.13 | -0.15 | -0.12 | -0.11 | -0.12 |
| YSU | 0.22 | 0.23 | 0.21 | 0.2 | 0.2 | -0.15 | -0.17 | -0.14 | -0.13 | -0.13 |
| Daegwallyeong | 0.13 | 0.12 | 0.12 | 0.09 | 0.09 | -0.08 | -0.07 | -0.06 | -0.03 | -0.03 |
| Iksan | 0.35 | 0.35 | 0.33 | 0.28 | 0.29 | -0.26 | -0.26 | -0.24 | -0.2 | -0.2 |
| Ulsan | 0.21 | 0.22 | 0.19 | 0.17 | 0.17 | -0.17 | -0.17 | -0.14 | -0.12 | -0.13 |
| Mokpo | 0.21 | 0.22 | 0.19 | 0.18 | 0.18 | -0.13 | -0.14 | -0.11 | -0.1 | -0.1 |
| Taehwa | 0.27 | 0.28 | 0.25 | 0.23 | 0.24 | -0.17 | -0.19 | -0.16 | -0.14 | -0.15 |