# Peer review of "Improving air quality forecasting with the assimilation of GOCI AOD retrievals during the KORUS-AQ period"

_Atmospheric Chemistry and Physics, 2019_

## Referee Comment (RC1) · Anonymous Referee #1 · 7 Oct 2019

General comments

The focus in the present study is to implement AOD derived from satellite observation over East Asian region with the Korean Geostationary Ocean Color Imager in attempt to improve air quality forecast. The preprocessed data were assimilated with three-dimensional variational data assimilation technique for the Weather Research and Forecasting model coupled with Chemistry. The impact of GOCI AOD on the air quality forecasting is examined by comparing the obtained results with AOD derived from MODIS observation as well as against in-situ PM2.5 at the surface. In the present study, the assimilation of purely surface PM2.5 concentrations systematically underes-

timates surface PM2.5 and prediction hold only for 6 hours. When the present GOCI AOD retrievals are assimilated with surface PM2.5 observations the forecasts are improved up to 24 h, with the most significant contributions to the prediction of heavy pollution events over South Korea.

The present study is very interesting and it is based on a comprehensive method, which is also very well described in the manuscript. In addition, the discussion of the results hold very well and this is also the case when uncertainties and limitations in the present study are discussed, as well as possible future improvements that could results in more realistic forecasts. However, there are two important questions or major comments that are in dispute and must be settled before this study can be accepted for publication in ACP.

Major comments

1. There is an issue when introducing information of GOCI AODs in the approach to simulate forecasts of air quality when the improvement of the latter is caused by an overestimation in GOCI AOD. For me this is not a correct approach and uncertain to rely on. One reason for the latter is that it seems not to be robust, considering that you will have differences in the statistics, (differences in the weights), when performing forecasts of air quality. The main reason for that is the cloudiness, thus, diverse cloudy conditions means differences in the availability/statistics in GOCI AOD from case to case.

2. There are issues with the language, which need to be improved. In the section Technical corrections below suggestions are given in an attempt to improve the language and clearness of the manuscript. However, my review and corrections of the manuscript concerning the language has only been carried out for the abstract and Introduction to show that the clearness of the text need to be improved. Therefore, I recommend that the full text needs an English proof-check.

Specific comments

[Figure]

Page 5 Line 17. MODIS is not a satellite and which version is used here, 6.1? I guess version 6.1 is used in the present study, otherwise it is necessary to switch to this latest versions, since there are biases in the previous versions. Line 22. It is not correct to write that AOD measures something. This sentence need to be rewritten.

Page 7 Line 15. Which version of MODIS? Line 17. "Observation errors" are not the best name to use here and there is a later estimates from the MODIS aerosol team of the expected error in the MODIS retrievals of AOD. Suggestion: "The MODIS land and ocean retrievals give AOT at 550nm with expected error envelopes of ïĄĎAOT $= \pm 0.05 \pm$ 0.15 *AOT (Levy et al., 2010) and ïĄĎAOT = +0.04+0.1 *AOT, ïĄĎAOT = âŤĂ0.02âŤĂ0.1*AOT (Levy et al., 2013), respectively, which arise from combined errors in assumed boundary conditions (e.g. surface reflectance, instrument calibration) and type of aerosol model (such as in single scattering albedo)."

Page 9 Line 1. And the sentence beginning with "Because the difference….." If the difference is so small should you not then go for thinning? Lines 5 and 6. The word "validation" can be used when comparing satellite derived AOD against ground-based sun-photometer measurements. However, you cannot validate AOD obtained from passive remote sensing against AOD derived from observations with another satellite sensor used in passive remote sensing. Line 15. Concerning the sentence "When these different observation errors were applied to GOCI retrievals in the assimilation, the smallest error (ÆŘ2) produced slightly better fits to observations specially for the high values (AOD > 2)……." This statement seems not hold, since ÆŘ2 is not better than ÆŘ1 over land for the situations with lower AOD. Line 26. Concerning this statement "This is partly because AOD is not directly associated with surface PM2.5 and partly because large uncertainties in the forecast model and the emission forcing can dominate over the analysis error during the model integration." how about uncertainties may also be induced in the forecasts of PM2.5 since here we have to deal with ambient AOD while dry conditions for PM2.5?

Page 11 Line 1. Is data used both from MODIS Aqua and Terra?

[Figure]

Section 5 and page 15 Line 1 and first sentence. I think too strong positive words are used here when describing the GOCI AOD retrievals.

Line 15 and the sentence "However, the forecast error grew very quickly over the next 12 hours, underestimating PM2.5 at the surface, especially in the heavy pollution events where the forecast accuracy dropped from over 70% to ~30% only in four hours. Meanwhile, the GOCI AOD retrievals alone tended to overestimate surface PM2.5 but significantly contributed to improving air quality forecasts up to 24 h when assimilated with surface PM2.5 observations." Thus, the improvement increase for the wrong reason and one of the problems with that seems to be that the approach is not robust, considering that you will probably get variations in the amount of data/statistics you get from GOCI AOD when investigating different cases. This is because of cloudiness, which means no data when clouds are presented. Thus, the GOCI AOD statistics will varies between different forecasts investigated, thus the latter will be dependent on this. Why is not the results obtained with MODIS included in the discussion of Section 5?

Figures

Figure 1 It is not clear how the two domains are connected to the solid box in the figure. Neither the text in the beginning Section 3.1 is clear about this. The number of the in-situ stations at Korean peninsula that deliver PM2.4 data for the present study are so much more than is shown by the dots in this figure. This could be improved somewhat by reducing the size of the current Figure 1 (the right part/eastern part) and include instead at this place on the right an enlargement of Korean peninsula? The size of Korean

Figure 2 It is not correct to write that AOD is retrieved at this time, since it is the observations that is carried out at this time and it is a very long way to come up with an estimate of AOD, for example you have to introduce a model that describe radiation transfer in the atmosphere. Change "retrieved" to "corresponding" in the first sentence

[Figure]

of the figure caption to Figure 2. Describe in the figure caption the solid black box introduced.

Figure 3 What is it on the y-axes? Should it be AODo – AODa and AODo – AODb ?

Figure 4 In the figure caption you have to refer to the body text about the three different types of observation errors. Take the color blind persons in consideration and use the three colors in combination with solid, dashed and dotted lines and separate land and ocean with heavy and normal lines, respectively.

Figure 5 $R^2$ is squared correlation coefficient.

Figure 6 Keep the color for the lines but use solid and dashed lines. Why is not the results discussed more than for is included as phrase in the bracket? A suggestion, skip the figure and write "(not shown)".

Figure 7 It is a lot of space in the figure and therefore write the names of the species in all figures.

Figure 8 Write "Model levels" connected to the y-axes.

Figure 9 The title (above the figures) is problematic both considering the language and that it is actually not 100% monthly mean values that are presented. I suggest to remove it and change the figure caption to. "Horizontal distribution of analysis increments in PM2.5 (analysis-minus-background) at the lowest model level (k=1) in domain 1, averaged over the period 4 – 31 May 2016. Maximum and mean values corresponding to the domain in each experiment are shown in the upper right corner of each panel."

However, text describing the different figures are also needed in the figure caption. Since no alphabetic characters, a – d, have been included in these four figures then you have to include the more complicated "upper left, upper right" etc. In addition the x-label text should be "dry PM2.5 [ïA■m m-3]"

The results presented in Figure 9 are somewhat difficult to understand, since including

result of GOCI AOD means that the final scene (All) abrupt get higher PM2.5 values in the upper part of the figure (Figure 9d). Is this realistic? In addition, how could the MODIS AOD results that are so limited in available values/statistics for the investigation area contribute to an improved forecast? It seems also that the MODIS AODs are only available east of the Korean peninsula, while the aerosol sources are located in west, over China.

Figure 10 Writing "9-km simulations" is not clear, explain what it is. Suggestion for the figure caption "Root-mean-square-error (rmse, upper figure) and bias error (lower figure) obtained for forecasts, with respect to the investigation area of 9-km (domain 1), verified against surface in-situ PM2.5 from 361 stations in South Korea of the period 4 – 31 May 2016. Average values of the forecasts (24 hours with increment of 1 hour) is shown next to each experiment name, where also mean absolute error is presented." Should the latter be presented in the upper figure instead? Please adjust the suggested figure text above if needed.

Figure 11 Suggestion "Figure 11. Same as Fig. 10, while here the results of forecast accuracy (%) for categorical forecasts are presented, subdivided according to classification of air quality in Tables 2 and 3." Include "Model level" on the y-axes.

Technical corrections

Abstract Line 2. Suggestion "The Korean Geostationary Ocean Color Imager (GOCI) satellite provide, based on daily high temporal and spatial resolution data, unprecedented information on air pollutants over the upstream region of the Korean peninsula for the last decade." Line 3. "the GOCI aerosol" Line 6. "......assimilated with three-dimensional.....technique in the Weather....." Line 9. Sensors are not observations, probably better with "....(MODIS) AOD..", but better with "...... Moderate Moderate Resolution Imaging Spectroradiometer (MODIS) AOD and in-situ ground-based fine particle matter (PM2,5)." Line 11."....... underestimates predicted surface PM2..5........." Line 11. What is meant by "positive impact" Is it better with "and the

prediction hold only for about 6 h." Line 13. The last part beginning with "with the most. . ." of this sentence is not clear.

Introduction Page 1 Line 19. "Complicated aerosol chemistry and due to its multiscale interactions. . . . . . . . . ." Line 21. "Surface concentrations" of what ? Line 22. If line 21 is corrected then it hold to write "they" in line 22. Line 23. "The latter is highly dependent on. . . . . ." Line 24. Take away "via aerosol-meteorology" and write ". . .meteorology due to interactions directly with solar radiation in the cloud-free atmosphere (scattering and absorption) and indirectly by acting as cloud condensation nuclei."

Page 2 Line 3. ". . ..performs chemical simulations according to 3-km horizontal resolution at present day." Line 4. What is meant by "these fast-varying complex mechanisms" or what is it pointed to? Lines 9-10. This sentence need to be rewritten. Line 12. Not clear written: "(usually in the optical properties)" Line 12. Change "observed information" to "information" or "results" Line 15. Change "for" to "of" Line 16. ". . ..conducted in Korea between 1 and 12 June 2016. . . . .. . . . .. . ." Line 17. Remove "a field campaign" Line 18. "of the major factors that highly influenced the air quality in Korea in the period 1 – 12 June 2016. Line 21. "occurred due to long-range. . .. . ." Lines 25-27. Based on the Korean Geostationary Ocean Color Imager (GOCI) onboard the Communication, Ocean, and Meteorology Satellite (COMS) retrievals of hourly AOD scenes, for multiple spectral bands, are centred with respect to the Korean peninsula during daytime (Kim et al., 2017). AOD scenes with high spatial and temporal resolutions are available since 2010." Line 28. "It has been demonstrated" Line 31. "assimilating AOD derived from MODIS observations (Remer et al., 2005)" Line 34. "forecasts of a dust storm" Line 34. When did the dust storm occurred? Line 35. "..widely used for air quality forecasting. The system has been extended for. . . .."

Page 3 Line 5. "MOdel. . ..." Line 8. ". . ..assimilation of AOD improved. . ." Line 9. "In the present study, the assimilation............system has been extended to be better used in the GOCI AOD retrievals during the current investigation period. . ." Line 10. Not clear what is meant by this"careful investigation of data characteristics." Lines 10-13.

[Figure]

The last part in this sentence is not clear ". . .. . .compared to that of other observations."
Line 13. "data and examine" Line 18. "conclusions are presented"

---

## Referee Comment (RC2) · Anonymous Referee #2 · 24 Oct 2019

The manuscript presents a study assimilating ground-based observations and satellite based retrievals to improve PM2.5 forecasts. The topic is relevant and in the scope of the Journal. There have been several previous studies about assimilating geostationary satellite retrievals, especially those from GOCI. This study builds over them and, in my mind, has a few additional contributions: The first and most important is that assimilating GOCI by itself doesn't seem to improve the forecasts for the study period, sometimes even making it worse. The authors only find improvements when they assimilate both surface and satellite data. Other contributions include separating assimilation performance by pollution regimes, and showing sensitivity studies on how to represent the error on the observations and how to aggregate them to speed

up the assimilation algorithm without worsening performance. These represent good contributions to the field and would grant publishing in ACP.

However, I think the paper needs a lot more work before it's ready for publication. In terms of the science, I think they need to do more work on understanding why GOCI makes the assimilation worse and why does it get better when including the surface observation. The study period coincides with a major field campaign where additional airborne and ground-based observation were made to try to tackle these questions. The authors also talk about the operational forecasts, so it would be good if they could include the performance of this system to understand if the assimilation efforts can help improve the current system. Also, it needs major improvements in the English. I suggest the authors to find support by a native English speaker. A few major and minor comments below.

Comments by line (<page> <lines>):

2 3-8. Provide references to these statements

5 21-22. But you mention above that Saide et al. (2014) used MOSAIC within GSI. For for Page 15, lines 1-5.

6 6. By doing cycles every 6 hours you are not taking advantage of the ∼hourly time resolution of GOCI data

Figure 1. Why show observations for a given time? Why not show maybe an average of the period analized?

Eqns 3-6. Please explain why there are more than one error equation and when would you use which

Figure 5 does not contributes to much information so I would drop it along with the discussion about it

Figures 8 nd 7. You could model vertical distribution and impact after assimilation using

airborne data and surface lidars deployed as part of KORUS-AQ

14 4-9. Be more specific here, mention the approached that you used of smoothing observations instead of thinning

15 6-11. I don't agree with these statements. There were a couple of flights during the hazy period you study that could be useful. You are showing that GOCI only degrades performance while including surface improves, so you should be changing the vertical resolution through the data assimilation. KORUS-AQ had airborne and ground based lidars that you can evaluate against to assess this. You can also evaluate the model ability to represent aerosol composition, both from supersites observations and aircrafts. You could also be including the ground based (AERONET-DRAGON) and airborne (4STAR) AOD data. There were even PM monitors in different ships that you could also use for evaluation.

Minor Edits (<page> <lines>): A few corrections but in general the manuscript doesn't read well for English

1 2: ". . . every day for the last decade, providing . . ."

1 4: "assimilated to make systematic improvements on air quality forecasting in South"

1 19: I would change "complications" by "uncertainties"

8 2-4: This sentence is very confusion, I would just erase it and keep the last sentence of the paragraph

9 14: ". . . account for representativeness error, we also tested with . . ."

9 24: "particularly for pollution events"

9 32: "the two observations types. . .."

13 27. This sentence is not clear, why do you mean by power instability?

13 33-34: . . ., but higher levels of pollution in SMA are not simulated either.

---

## Author Comment (AC1) · 22 Dec 2019

xcolor, colortbl

We appreciate the reviewer's valuable comments. It is our belief that they greatly helped improving our manuscript. In response to your major concern, we now added three more figures (as new figures 12 - 14) and one more table (Table 4) for additional verification against independent observations during the KORUS-AQ field campaign. Please find our point-by-point response in blue below.

[Figure]

Anonymous Referee 2:

The manuscript presents a study assimilating ground-based observations and satellite based retrievals to improve PM2.5 forecasts. The topic is relevant and in the scope of the Journal. There have been several previous studies about assimilating geostationary satellite retrievals, especially those from GOCI. This study builds over them and, in my mind, has a few additional contributions: The first and most important is that assimilating GOCI by itself doesn't seem to improve the forecasts for the study period, sometimes even making it worse. The authors only find improvements when they assimilate both surface and satellite data. Other contributions include separating assimilation performance by pollution regimes, and showing sensitivity studies on how to represent the error on the observations and how to aggregate them to speed up the assimilation algorithm without worsening performance. These represent good contributions to the field and would grant publishing in ACP.

=> The most important result of this study is summarized in Fig. 11 where the assimilation of GOCI retrievals was shown to be particularly helpful in improving the forecast accuracy in high pollution events and keeping the positive impact for 24 h. Without the help of GOCI retrievals, the assimilation of surface PM alone was not effective after 6 h, especially in predicting high pollution events. Hence, we disagree with your first and the most important finding from our study that assimilating GOCI alone doesn't improve the forecasts and sometimes makes them worse.
In response to your comments and for clarification, however, we added new figures 12 -14 where 0-23h forecasts from each experiment are verified against independent observations to demonstrate the positive impact of GOCI without the help of surface PM observations. With respect to both total AOD at 500 nm from AERONET sites (Fig. 13) and to surface PM2.5 concentrations from the stations operated by NIER during

the field campaign (Fig. 14), the assimilation of GOCI alone mostly outperforms PM and MODIS experiments. Details on the new figures are now added as the last two paragraph in section 4.2.

However, I think the paper needs a lot more work before it's ready for publication. In terms of the science, I think they need to do more work on understanding why GOCI makes the assimilation worse and why does it get better when including the surface observation. The study period coincides with a major field campaign where additional airborne and ground-based observation were made to try to tackle these questions. The authors also talk about the operational forecasts, so it would be good if they could include the performance of this system to understand if the assimilation efforts can help improve the current system. Also, it needs major improvements in the English. I suggest the authors to find support by a native English speaker. A few major and minor comments below.

=> Again, GOCI did not make the assimilation worse. As noted in lines 27 – 28 in page 11 in the original version, the assimilation of AOD retrievals (either GOCI or MODIS) alone does not improve the surface analysis. The main reason for that is that AOD retrievals are not directly associated with surface PM2.5 concentrations. This is already discussed along with the imperfection of the observation operator and the numerical modeling system in the last paragraph of section 3.2.2 (Page 9: lines 25 - 29) and lines 12 - 15 in Page 11. Moreover, in the 3DVAR assimilation, the model estimates of AOD are strongly constrained by the model error structure of each aerosol species both horizontally and vertically. As such, the most challenging part with the real data assimilation is to improve subsequent forecasts *per se*. Please note that this is not an idealized study using a simple model or simulated observations. It is not practically feasible to isolate numerous factors in the study where real observations are assimilated using the real (e.g. non-idealized) system. However, based on your comments,

we decided to add one more paragraph in the last section. Please find the second paragraph in section 5 that summarizes our discussion on this matter.

As for the operational forecasts, it is correct that this study was motivated by an effort to improve the current operational forecasts, but because they use a different chemical transport model with no data assimilation, it is not relevant to directly compare to them. Also, the focus of this study is to examine the impact of GOCI retrievals (which are not included in the operational forecasts at NIER), so we did not include them here.

In regards to English, this manuscript was internally reviewed twice and already proofread by another native English speaker. Without exact lines or paragraphs specified, it is hard to figure out where and how we need to make major improvements. But we made some corrections while editing section 4.2 to add new figures and table. For instance, we changed page 11, line 28 as "the analysis error is smaller than those in other experiments" by as "those in". Also, in respect of reviewer's comment, we've gone through another round of proof-reading and made necessary corrections throughout the manuscript once again. Thanks for your suggestion.

Comments by line (<page> <lines>):

2 3-8. Provide references to these statements

=> This article is now added to our reference: Chang, L.-S., Cho, A., Park, H., Nam, K., Kim, D., Hong, J.-H., and Song, C.-K.: Human-model hybrid Korean air quality forecasting system, Journal of the Air and Waste Management Association, 66, 896–911, https://doi.org/10.1080/10962247.2016.1206995, 2016.

5 21-22. But you mention above that Saide et al. (2014) used MOSAIC within GSI. For for Page 15, lines 1-5.

=> We now added "publicly" before "available ∼".

6 6. By doing cycles every 6 hours you are not taking advantage of the hourly time resolution of GOCI data

=> Agree. We may need to increase the cycling frequency to 3 hourly or hourly in the near future. We now added one statement "This configuration was chosen in the limitation of computational resources, but the use of higher resolutions both in time and space might be desirable to further improve forecast skills in the future." in lines 12 - 14 in page 5 (Section 3.1).

Figure 1. Why show observations for a given time? Why not show maybe an average of the period analized?

=> Figure 1 simply shows the model domain with the observing network. No changes are made.

Eqns 3-6. Please explain why there are more than one error equation and when would you use which

=> Please check the paragraph for the equations (Page 9; lines 4-16) once again as we already explained that the retrieval error was estimated (by Choi et al. 2018) differently depending on which data was used for the verification of the retrievals. And due to representativeness error in the model, the observation error can be adjusted in the assimilation. The following statement (lines 17-22 in the same page 9) already described what we used ($\epsilon_2$) and why we used it.

Figure 5 does not contributes to much information so I would drop it along with the discussion about it

=> Figure 5 is answering part of your question on why the assimilation of GOCI alone does not seem to improve forecasts. We keep it.

Figures 8 nd 7. You could model vertical distribution and impact after assimilation using airborne data and surface lidars deployed as part of KORUS-AQ

=> Not clear on your point here. Figures 7 and 8 show how the model responded to the assimilation of observations in use. This analysis is needed to understand how our assimilation worked in the model space. It has nothing to do with verification.

14 4-9. Be more specific here, mention the approached that you used of smoothing observations instead of thinning

=> We now added at the end of line 9 "We averaged all the pixels over each grid box at 27-km resolution (e.g. superobing) instead of thinning them randomly, for instance."

15 6-11. I don't agree with these statements. There were a couple of flights during the hazy period you study that could be useful. You are showing that GOCI only degrades performance while including surface improves, so you should be changing the vertical resolution through the data assimilation. KORUS-AQ had airborne and ground based lidars that you can evaluate against to assess this. You can also evaluate the model ability to represent aerosol composition, both from supersites observations and aircrafts. You could also be including the ground based (AERONET-DRAGON) and airborne (4STAR) AOD data. There were even PM monitors in different ships that you could also use for evaluation.

=> We now added three new figures (figures 12 – 14) and one more table (Table 4) for more verification. Please find the corresponding statement in the last part of section 4.2.

Minor Edits (<page> <lines>): A few corrections but in general the manuscript doesn't read well for English

1 2: ". . . every day for the last decade, providing . . ."

=> Changed, as suggested.

1 4: "assimilated to make systematic improvements on air quality forecasting in South"

=> We prefer our original statement. No changes.

1 19: I would change "complications" by "uncertainties"

=> Changed.

8 2-4: This sentence is very confusion, I would just erase it and keep the last sentence of the paragraph

=> This statement is related to time windowing, which is an important aspect of 3DVAR since it counts all the observations within the window as available at the same time. And weighting the observations based on their report time can affect the analysis quality. We leave it as it is.

9 14: ". . . account for representativeness error, we also tested with . . ."

=> We do not see any difference by changing "tried" with "tested". No changes.

9 24: "particularly for pollution events"

=> "polluted" is now changed to "pollution", as suggested.

9 32: "the two observations types. . .."

=> We believe "two observation types" is correct, not "two observations types". No changes.

13 27. This sentence is not clear, why do you mean by power instability?

=> That's what we were told. But we omit "due to the power instability" now.

13 33-34: . . ., but higher levels of pollution in SMA are not simulated either.

=> Changed as suggested. Thank you.

---

## Author Comment (AC2) · 22 Dec 2019

[journal abbreviation, manuscript]copernicus hyperref xcolor, colortbl

We appreciate the reviewer's time and effort to improve the clarity of this manuscript. It is our belief that the comments helped make our draft clear. Please find our point-by-point response in blue below.

General comments

The focus in the present study is to implement AOD derived from satellite observation over East Asian region with the Korean Geostationary Ocean Color Imager in attempt to improve air quality forecast. The preprocessed data were assimilated with three-dimensional variational data assimilation technique for the Weather Research and Forecasting model coupled with Chemistry. The impact of GOCI AOD on the air quality forecasting is examined by comparing the obtained results with AOD derived from MODIS observation as well as against in-situ PM2.5 at the surface. In the present study, the assimilation of purely surface PM2.5 concentrations systematically underestimates surface PM2.5 and prediction hold only for 6 hours. When the present GOCI AOD retrievals are assimilated with surface PM2.5 observations the forecasts are improved up to 24 h, with the most significant contributions to the prediction of heavy pollution events over South Korea.

The present study is very interesting and it is based on a comprehensive method, which is also very well described in the manuscript. In addition, the discussion of the results hold very well and this is also the case when uncertainties and limitations in the present study are discussed, as well as possible future improvements that could results in more realistic forecasts. However, there are two important questions or major comments that are in dispute and must be settled before this study can be accepted for publication in ACP.

Major comments
1. There is an issue when introducing information of GOCI AODs in the approach to simulate forecasts of air quality when the improvement of the latter is caused by an overestimation in GOCI AOD. For me this is not a correct approach and uncertain to rely on. One reason for the latter is that it seems not to be robust, considering that you will have differences in the statistics, (differences in the weights), when performing forecasts of air quality. The main reason for that is the cloudiness, thus, diverse cloudy conditions means differences in the availability/statistics in GOCI AOD from case to

case.

=> This statement is not concise, but we assume that the reviewer is concerned about two factors.

First, for the small data availability in the cloudy conditions, the total number of GOCI data used in the assimilation was at least more than 2,000 each cycle, as shown in Fig. 3 (with the right y-axis). As the number fluctuates with cycle, the data impact certainly changes (based on the differences between (o-b)'s and (o-a)'s in black), but (o-a)'s are always smaller than (o-b)'s, showing how robust our assimilation system is. Also, our conclusion is made not from a single case, but based on the one-month statistics.

In regards to the overestimation due to the assimilation of GOCI AODs, here is our response: Model trajectories are always deviated from the observed states and data assimilation is trying to pull the model states toward the observed information. If the model states were severely underestimated, they are drawn to observations to the extent the model is uncertain and the observations are trusted (based on their error statistics). Here the GOCI data tends to slightly overestimate surface PM2.5, trying to compensate for the systematic underestimation (which is a long-standing issue of the bulk GOCART aerosol scheme). But when assimilated with surface PM2.5 observations, the overestimation mostly disappears as both GOCI and surface data are affecting the model states together. The effect of GOCI data can be further adjusted through the observation error variance, but the observation error should not be adjusted for different experiments with different datasets. Moreover, this study is not meant to optimize the system for the particular case. As always, there might be a room for further optimizing the assimilation algorithm (such as improving observation error statistics or the observation operator) or further constraining the model states through other components like emission data, but that is not the focus of this study. The goal of this work is to examine the relative impact of the GOCI data in the same analysis and forecast system using the same forcing (e.g. emissions and boundary

conditions). Please note that our results were reliable and consistent throughout the month-long period.

2. There are issues with the language, which need to be improved. In the section Technical corrections below suggestions are given in an attempt to improve the language and clearness of the manuscript. However, my review and corrections of the manuscript concerning the language has only been carried out for the abstract and Introduction to show that the clearness of the text need to be improved. Therefore, I recommend that the full text needs an English proof-check.

=> This manuscript has been internally reviewed twice in our lab and proofread by another native English speaker. As we replied to the reviewer's technical corrections at the very bottom, most of the corrections the reviewer suggested are incorrect in English or distorted our points, if not irrelevant, in the manuscript. The reviewer raised most of the comments or questions regarding MODIS retrievals, which is not the focus of our study but included only for completeness, we thus wanted to stay focused on our goal and highlights of our work. However, we appreciate different views and tried our best to accommodate the reviewer's comments and reflect them in our manuscript unless we have a specific reason not to.

Specific comments

Page 5, Line 17: MODIS is not a satellite and which version is used here, 6.1?

=> We modified "MODIS and GOCI satellites" to "MODIS and GOCI sensors". And yes, version 6.1 was used.

Line 22: It is not correct to write that AOD measures something. This sentence need to be rewritten.

=> This is simply another way of expressing the definition.

In $https://neo.sci.gsfc.nasa.gov/view.php?datasetId = MODAL2\_M\_AER\_OD$, for example, aerosol optical thickness (or depth) is described as "a measure of how much light the airborne particles prevent from traveling through the atmosphere", which is consistent with our sentence "Aerosol optical depth (AOD) measures the amount of light extinction by aerosol scattering and absorption in the atmospheric column". As nothing is wrong with the expression, it is unchanged.

Page 7 Line 15. Which version of MODIS? Line 17. => V6.1

"Observation errors" are not the best name to use here and there is a later estimates from the MODIS aerosol team of the expected error in the MODIS retrievals of AOD. Suggestion: "The MODIS land and ocean retrievals give AOT at 550nm with expected error envelopes of $AOT =\pm$0.05$\pm$ 0.15 \*AOT (Levy et al., 2010) and AOT = +0.04+0.1 AOT = ?0.02??0.1AOT$(Levy et al., 2013), respectively, which arise from combined errors in assumed boundary conditions (e.g. surfac$

=> In the data assimilation framework, errors are divided up into two major categories - background errors and observation errors. We appreciate the details of the reviewer's comments, but this study should read in the context of data assimilation, not for the retrieval of MODIS aerosol products. In other words, the errors should be defined either from the observation side or from the model side in the assimilation system, so we believe that it is more appropriate to describe the retrieval errors as observation errors. Also, the error estimate was first described in Remer et al. (2005), as stated in our draft, so we leave the reference as it is. But for better clarification, we modified the statement as below. "Following Remer et al. (2005), observation errors are specified as the retrieval errors: (0.03 + 0.05 \* AOD) over ocean and (0.05 + 0.15 \* AOD) over

land. They do not include the representativeness error and are slightly smaller than those for GOCI AOD, as described below."

Page 9 Line 1. And the sentence beginning with "Because the difference. . ..." If the difference is so small should you not then go for thinning?

=> As we stated in the second paragraph in page 8, it is common practice to go thinning satellite data to reduce the data volume in data assimilation. But based on our results, we decided to go for superobing instead of thinning, not because the forecast performance is much different (which is not the case), but mostly for the computational efficiency.

Lines 5 and 6. The word "validation" can be used when comparing satellite derived AOD against ground-based sun-photometer measurements. However, you cannot validate AOD obtained from passive remote sensing against AOD derived from observations with another satellite sensor used in passive remote sensing.

=> We do not want to argue about how others described their work. The term of "validation" was used in Choi et al. (2018) and we just adopted it here. Also, in a broad sense, the terminology of "validation" is commonly used when one data is evaluated against another independent observation in the data assimilation community, so we do not see it problematic. No changes.

Line 15. Concerning the sentence "When these different observation errors were applied to GOCI retrievals in the assimilation, the smallest error ($\epsilon_2$) produced slightly better fits to observations specially for the high values (AOD > 2)........" This statement seems not hold, since $\epsilon_2$ is not better than $\epsilon_1$ over land for the situations with lower AOD.

[Figure]

=> We do not understand why our statement doesn't hold due to the case of lower AOD, which we did not even discuss here. We mentioned that the smallest error ($\epsilon_2$) produced slightly better fits to observations for the high(!) AOD values. We also stated that such a result is not statistically significantly different, so we do not understand why the reviewer is arguing over the statement. No changes.

Line26. Concerning this statement "This is partly because AOD is not directly associated with surface PM2.5 and partly because large uncertainties in the forecast model and the emission forcing can dominate over the analysis error during the model integration." how about uncertainties may also be induced in the forecasts of PM2.5 since here we have to deal with ambient AOD while dry conditions for PM2.5?

=> The sentence "how about uncertainties may also be induced in the forecasts of PM2.5 since here we have to deal with ambient AOD while dry conditions for PM2.5?" doesn't make sense, so we are not sure what the reviewer's point is. But just for clarification, here is what we meant: Even if the high-volume AOD data are assimilated, if the information in AOD retrievals is not well matched with surface PM2.5, it may not contribute much to improving the forecast of surface PM2.5 concentrations. That's why it is important to examine the relationship between AOD and surface PM2.5.

Section 5 and page 15 Line 1 and first sentence. I think too strong positive words are used here when describing the GOCI AOD retrievals.

=> Which part is too strong? We concluded solely based on our results here. Please elaborate your comment or specify the statement that is considered to be unsuitable.

Line 15 and the sentence "However, the forecast error grew very quickly over the next 12 hours, underestimating PM2.5 at the surface, especially in the heavy pollution events where the forecast accuracy dropped from over 70% to ∼30% only in four

hours. Meanwhile, the GOCI AOD retrievals alone tended to overestimate surface PM2.5 but significantly contributed to improving air quality forecasts up to 24 h when assimilated with surface PM2.5 observations." Thus, the improvement increase for the wrong reason and one of the problems with that seems to be that the approach is not robust, considering that you will probably get variations in the amount of data/statistics you get from GOCI AOD when investigating different cases. This is because of cloudiness, which means no data when clouds are presented. Thus, the GOCI AOD statistics will varies between different forecasts investigated, thus the latter will be dependent on this.

=> We agree with the reviewer in that the results might vary between different cases to some extent. But we do not agree with the comment that the improvement was made for a wrong reason. During the one-month cycling, we could assimilate at least a couple of thousands of data points even in cloudy days, as shown in Fig. 3 where the right y-axis starts from 2,000. Also, if you compare the black dashed line (o-b GOCI) with the black solid line (o-a GOCI) in the figure, you can see the analysis produced much better fits to the observations, meaning that the analysis itself was done correctly. When the number of data in use gets reduced, the differences between (o-b)'s and (o-a)'s get reduced as well, implying that the effect of the data gets smaller, as expected. But (o-a)'s are consistently better than (o-b)'s throughout the cycles, showing how robust the system is. And even if the spatial coverage of GOCI data varies cycle to cycle, we still use more GOCI data than surface observations (which is typically around 1,000 at each cycle). Also note that we concluded about the data impact based on the month-long statistics, not on a single case.

Why is not the results obtained with MODIS included in the discussion of Section 5?

=> We chose to summarize our main points in the last section, rather than listing all the results. The impact of MODIS data has been examined in many previous studies, and we included MODIS data just for completeness, so nothing to emphasize on the

MODIS data here.

Figure 1 It is not clear how the two domains are connected to the solid box in the figure. Neither the text in the beginning Section 3.1 is clear about this.

=> We now modified the caption as "Two model domains - domain 1 (outer box) at 27-km resolution nested down to the inner box for domain 2 " to add "(outer box)".

The number of the in- situ stations at Korean peninsula that deliver PM2.4 data for the present study are so much more than is shown by the dots in this figure. This could be improved somewhat by reducing the size of the current Figure 1 (the right part/eastern part) and include instead at this place on the right an enlargement of Korean peninsula?

=> In fact, many Korean sites are tightly overlapped to each other. As such, zooming in the map does not help much to recognize individual sites. But Figure 13 (now as Figure 16) can give readers a sense of how they are distributed over South Korea. Since this study focused on the impact of GOCI data, not surface stations, we believe it is enough to show the entire model configuration in Fig.1. No changes.

Figure 2 It is not correct to write that AOD is retrieved at this time, since it is the observations that is carried out at this time and it is a very long way to come up with an estimate of AOD, for example you have to introduce a model that describe radiation transfer in the atmosphere. Change "retrieved" to "corresponding" in the first sentence of the figure caption to Figure 2. Describe in the figure caption the solid black box introduced.

=> This study is not meant for describing the retrieval process, but how the data is used in the assimilation cycle. The data was processed at the time and that's how they are described and presented in Figure 2. Even in-situ measurements such as

[Figure]

radiosonde do not report the values at the exact time (depending on the vertical levels as it goes up), but in the data assimilation context, that's how they are all described. Please note that we already illustrated the temporal distribution of the data in the last paragraph of page 7 ("In terms of temporal distribution, ). In response to your last comment, though, we added one paragraph "Domain 2 is marked as a black box in each panel." at the end of the caption.

Figure 3 What is it on the y-axes? Should it be AODo – AODa and AODo – AODb ?

=> We believe the reviewer already understood what we plotted based on our y-axis label (and our caption). Along with the main title "GOCI AOD", it must be clear that our y-axis label "(o-a) and (o-b)" means "AODo-AODa and AODo-AODb". No changes are made.

Figure 4 In the figure caption you have to refer to the body text about the three different types of observation errors. Take the color blind persons in consideration and use the three colors in combination with solid, dashed and dotted lines and separate land and ocean with heavy and normal lines, respectively.

=> This draft uses a lot of colors throughout the figures, and is not meant for color-blinded readers, unfortunately. But based on your comment, we added "The first two errors ($\epsilon_1$ and $\epsilon_2$) are described in equations (3) - (6) and the third error ($\epsilon_3$) increases $\epsilon_2$ by 20% everywhere." in the caption.

Figure 5 R2 is squared correlation coefficient. => Corrected in figure and caption.

Figure 6 Keep the color for the lines but use solid and dashed lines. Why is not the results discussed more than for is included as phrase in the bracket? A suggestion,

skip the figure and write "(not shown)".

=> Figure 6 is important in that it gives an overview of the entire month of interest, summarizing how much we can improve the forecasts with the assimilation of all the data considered for the whole period. Moreover, it nicely introduces the high pollution events we discuss later, so we should not omit it. And we decided to go with solid lines to better distinguish the forecasts from dots for observations. No changes are made.

Figure 7 It is a lot of space in the figure and therefore write the names of the species in all figures.

=> We decided to put the species name in the main title because the first panel does not have room for it due to the legend. As this figure has to take up the whole page (height-wise) anyway, we decided to keep the main title. No changes.

Figure 8 Write "Model levels" connected to the y-axes.

=> The caption already stated that it is the same as Figure 7. We tried to reserve more x-axis space to zoom in differences between the experiments here, dropping y-axis title intentionally. No changes made.

Figure 9 The title (above the figures) is problematic both considering the language and that it is actually not 100% monthly mean values that are presented. I suggest to remove it and change the figure caption to. "Horizontal distribution of analysis increments in PM2.5 (analysis-minus-background) at the lowest model level (k=1) in domain 1, averaged over the period 4 – 31 May 2016. Maximum and mean values corresponding to the domain in each experiment are shown in the upper right corner of each panel." However, text describing the different figures are also needed in the figure caption. Since no alphabetic characters, a – d, have been included in these four figures then

you have to include the more complicated "upper left, upper right" etc. In addition the x-label text should be "dry PM2.5 [um m-3]"

=> We changed the main title to "Analysis increment". As for the annotation of each panel, we believe the experiment name is the best way to describe each panel since this figure highlights differences between the experiments with different observations assimilated. We never use "upper left, upper right", etc. here and go by the experiment name to be clear on which data we compare. As for the label bar title, PM2_5_DRY is the exact name of the model variable we read from the model output. The caption is now changed accordingly, as follows. "Horizontal distribution of analysis increments (analysis-minus-background) in PM2_5_DRY, the model variable corresponding to PM2.5, at the lowest level in domain 1, averaged over the period of May 4 - 31, 2016. Maximum and mean values of the domain in each experiment are shown in the upper right corner of each panel."

The results presented in Figure 9 are somewhat difficult to understand, since including result of GOCI AOD means that the final scene (All) abrupt get higher PM2.5 values in the upper part of the figure (Figure 9d). Is this realistic? In addition, how could the MODIS AOD results that are so limited in available values/statistics for the investigation area contribute to an improved forecast? It seems also that the MODIS AODs are only available east of the Korean peninsula, while the aerosol sources are located in west, over China.

=> To determine if the analysis increment is realistic, we verified the analysis and the following forecasts with respect to observations, as shown in the following figures. In regards to the impact of MODIS AOD products, the data can affect 15 three-dimensional GOCART aerosol species in the model states and the impact of each data point can be extended to the neighboring area as specified in the background error covariance.

Figure 10 Writing "9-km simulations" is not clear, explain what it is. Suggestion for the figure caption "Root-mean-square-error (rmse, upper figure) and bias error (lower figure) obtained for forecasts, with respect to the investigation area of 9-km (domain 1), verified against surface in-situ PM2.5 from 361 stations in South Korea of the period 4 – 31 May 2016. Average values of the forecasts (24 hours with increment of 1 hour) is shown next to each experiment name, where also mean absolute error is presented." Should the latter be presented in the upper figure instead? Please adjust the suggested figure text above if needed.

=> Based on the reviewer's comment, the caption is changed as below. "A time series of root-mean-square-error (rmse; upper panel) and bias (lower panel) of the hourly forecasts from the 00 Z initialization for May 4 - 31, 2016. Different experiments in domain 2 are verified against the same surface PM2.5 observations from 361 stations in South Korea. An average of 0-24 h forecast errors is shown next to each experiment name. The mean absolute error (mae) over the 24-h forecasts is also shown in the lower panel."

Figure 11 Suggestion "Figure 11. Same as Fig. 10, while here the results of forecast accuracy (%) for categorical forecasts are presented, subdivided according to classification of air quality in Tables 2 and 3." Include "Model level" on the y-axes.

=> Figure 11 shows different statistics, not the model level, as shown in the main title. No changes made.

Technical corrections

It was very hard to read through all the comments because they are mostly incomplete and are not separated by lines. In many cases, the modifications that the reviewer suggested either do not flow well in our manuscript, or misrepresent our points, or are

simply wrong in grammar. The reviewer tries to change our manuscript line by line in his/her way, but we should ask for being respectful for the authors' work. But here are our responses to the questions:

What is meant by "positive impact"?

=> The impact is considered to be positive when the following forecasts are improved with the reduction of forecast error (in terms of rmse, bias, and the categorical forecast accuracy).

Line 13 in abstract: The last part beginning with "with the most. . ." of this sentence is not clear.

=> We clearly demonstrated the most significant contributions to the prediction of heavy pollution events in Figure 11 where the assimilation of GOCI data produced the biggest improvement in b) high pollution accuracy.

Introduction Page 1 Line 21: "Surface concentrations" of what?

=> of chemical species. We believe this should be clear as the previous paragraph is immediately followed by this one.

Page 2, Line 4: What is meant by "these fast-varying complex mechanisms" or what is it pointed to?

=> All the mechanisms described in the previous paragraph, particularly the aerosol-meteorology interaction at short time scales. Again, this sentence is also connected with the paragraph right ahead.

Page 3, Line 10: Not clear what is meant by this "careful investigation of data characteristics".

=> We meant by examining the data in various ways, particularly in preprocessing and the error characterization, as shown in figures 2-5.

The last part in this sentence is not clear ". . .. . .compared to that of other observations." Line 13.

=> As described, compared to the impact of other observations on surface PM2.5 forecasts, we meant.

---

## Referee Report (RR1)

*I have used the first review of the present original version of the manuscript as a ground here in my second review of the current study. Additional comments by me are denoted with Italic font style. For some of the issues below I have included the answers from the authors to my comments in the first review, which are marked with blue colors and an arrow at the begging of the text. The comments by me and answers by the authors below are pointing to updated page and line numbers that correspond to the revised version of the manuscript. I have only included issues from my first review of this manuscript that are still relevant to include here.*

**Major comment**

**2**. There are issues with the language, which need to be improved. In the section Technical corrections below suggestions are given in an attempt to improve the language and clearness of the manuscript. However, my review and corrections of the manuscript concerning the language has only been carried out for the abstract and Introduction to show that the clearness of the text need to be improved. Therefore, I recommend that the full text needs an English proof-check.
=> This manuscript has been internally reviewed twice in our lab and proofread by another native English speaker. As we replied to the reviewer's technical corrections at the very bottom, most of the corrections the reviewer suggested are incorrect in English or distorted our points, if not irrelevant, in the manuscript. The reviewer raised most of the comments or questions regarding MODIS retrievals, which is not the focus of our study but included only for completeness, we thus wanted to stay focused on our goal and highlights of our 10 work. However, we appreciate different views and tried our best to accommodate the reviewer's comments and reflect them in our manuscript unless we have a specific reason not to.
*It is not ok by the authors to ignore many of my suggestions concerning the language, thus, here they have not replied at all (see technical corrections below). My suggestion of an English proof-check remain. However, when now have reading more of particular Sections 4 and 5 I do not think it is much work needed to improve the text in the Sections 3, 4 and 5.*

**Specific comments**

Line 17: MODIS is not a satellite and which version is used here, 6.1?
=> We modified "MODIS and GOCI satellites" to "MODIS and GOCI sensors". And yes, version 6.1 was used.
*Since I do not find it in the revised manuscript, please include text that describe it is Version 6.1 that is used in the present study.*

Line 22. It is not correct to write that AOD measures something. This sentence need to be re-written.
*By writing "AOD measures the amount…." then the latter word is a verb and it somewhat odd that a parameter measuring something (an active action), although it might be correct in the English language. By using "AOD is a measure of the amount…." Instead then the latter*

*word is a substantive, which is a more suitable phrase. Itt is the same as writing "AOD is an estimate of the amount….".*

Lines 19-21. "Following Remer et al. (2005), observation errors are specified as the retrieval errors: (0.03 + 0.05 * AOD) over ocean and (0.05 + 0.15 * AOD) over land. They do not include the representativeness error and are slightly smaller than those for GOCI AOD, as described below."
*The former part of this sentence sounds better now. However, it is very strange that the authors refer to retrieval errors corresponding to the ocean retrievals, estimated by the MODIS aerosol team (including Lorraine Remer), that are not valid anymore. Please update the retrieval error and corresponding references according to my comment in the first review of the original version of the manuscript.*

Lines 7 and 8. The word "validation" can be used when comparing satellite derived AOD against ground-based sun-photometer measurements. However, you cannot validate AOD obtained from passive remote sensing against AOD derived from observations with another satellite sensor used in passive remote sensing.
=> We do not want to argue about how others described their work. The term of "validation" was used in Choi et al. (2018) and we just adopted it here. Also, in a broad sense, the terminology of "validation" is commonly used when one data is evaluated against another independent observation in the data assimilation community, so we do not see it problematic. No changes.
*You don't need to argue here and instead of just adopt this term and rely on what is commonly used it is better to go to a dictionary. The word validation is a too strong word to describe an inter-comparison of AOD obtained with passive remote sensing from two different satellite platforms (see also comment below corresponding to Figure 2). The word "evaluation", used in the text above by the authors or "comparison" are options that are more suitable.*

Lines 18-21. Concerning the sentence "When these different observation errors were applied to GOCI retrievals in the assimilation, the smallest error ($\epsilon_2$) produced slightly better fits to observations specially for the high values (AOD > 2)…….." This statement seems not hold, since $\epsilon_2$ is not better than $\epsilon_1$ over land for the situations with lower AOD.
=> We do not understand why our statement doesn't hold due to the case of lower AOD, which we did not even discuss here. We mentioned that the smallest error ($\epsilon_2$) produced slightly better fits to observations for the high (!) AOD values. We also stated that such a result is not statistically significantly different, so we do not understand why the reviewer is arguing over the statement. No changes.
*Remove then "specially" from the original sentence in the manuscript, then your argumentation and statement hold.*

Lines 29-31. *Suggestion, change the sentence "This is partly because AOD is not directly associated with surface PM2.5 ………" to "This is partly because AOD is not directly related to surface PM2.5 ………2, since "related" is a more correct word to use.*

Section 5 and page 14
Line 26 and first sentence in this section. I think it is too strong positive words are used here when describing the GOCI AOD retrievals.

*I am sorry about this comment since I refer to the sentence on page 15 instead of the correct page 14: "GOCI AOD retrievals provide reliable and consistent aerosol information, monitoring air pollutants flowing over to the Korean peninsula at high resolution every day." Thus, I think it is too strong positive words used in the first part of the sentence when describing the GOCI AOD satellite retrievals on the accuracy of air quality forecasting*
*This is what the authors write in the abstract "During the Korea-United States Air Quality (KORUS-AQ) period (May 2016), the impact of GOCI AOD on the accuracy of air quality forecasting is examined by comparing with other observations including Moderate Resolution Imaging Spectroradiometer (MODIS) sensors and fine 10 particulate matter (PM2:5) observations at the surface." Concerning the latter parameter the result presented in Figure 5 is not impressive. In addition, you cannot rely the statement on an inter-comparison with AOD derived from observations carried out from another satellite platform.*
*The first and second parts of the sentence is not synchronized and this is an odd phrase "air pollution flowing over…."*
*Therefore, here is a suggestion: "GOCI provide daily AODs with high spatial resolution and frequently detect air pollutants over to the Korean peninsula."*

*Figures*

Figure 2 It is not correct to write that AOD is retrieved at this time, since it is the observations that is carried out at this time and it is a very long way to come up with an estimate of AOD, for example you have to introduce a model that describe radiation transfer in the atmosphere. Change "retrieved" to "corresponding" in the first sentence of the figure caption to Figure 2. Describe in the figure caption the solid black box introduced.

=> This study is not meant for describing the retrieval process, but how the data is used in the assimilation cycle. The data was processed at the time and that's how they are described and presented in Figure 2. Even in-situ measurements such as radiosonde do not report the values at the exact time (depending on the vertical levels as it goes up), but in the data assimilation context, that's how they are all described. Please note that we already illustrated the temporal distribution of the data in the last paragraph of page 7 ("In terms of temporal distribution, ). In response to your last comment, though, we added one paragraph
25 "Domain 2 is marked as a black box in each panel." at the end of the caption.

*It has nothing to do with reporting the values at the exact time, instead you have to separate "retrieved" and "observed", thus, the TOA radiance was measured/observed at the current time. It is just simply to write it more correctly so you don't hide that AOD derived from satellite measurements is not a direct observation, instead it need to be related to measured radiance scattered only by aerosols. The latter means that the contributions of Rayleigh scattering and surface reflection have to be reduced for. In addition, to relate AOD with TOA radiance associated with purely aerosols you have to introduce radiative transfer calculations. I present here a new suggestion: "Field of GOCI AOD at 550 nm derived/retrieved from observations carried out at 2016-05-01_06:00:00 UTC…………"*

*Figure 3. You have to include information that the right y-axis goes down to 2000 (the latter value I picked up in the file with the author's response in the review process), since it is otherwise hard for the readers to understand part of the figure. You need to use different symbols for each cycle or at least each pair of cycles. The half-last part of the last sentence in the figure caption need to be re-written.*

Figure 4

In the figure caption you have to refer to the body text about the three different types of observation errors. Take the color blind persons in consideration and use the three colors in combination with solid, dashed and dotted lines and separate land and ocean with heavy and normal lines, respectively.

=> This draft uses a lot of colors throughout the figures, and is not meant for color-blinded readers, unfortunately. But based on your comment, we added "The first two errors (1 and _2)) are described in equations (3) - (6) and the third error (_3) increases by 20% everywhere." in the caption.

*I agree on "a lot of colors", but why not make it easier, particularly for the colorblind person, by improving those figures for which it possible to do that for? You do not use colors in Figure 3 and it seems to work relatively well (see comments concerning Figure 3 above).*

Figure 7

It is a lot of space in the figure and therefore write the names of the species in all figures.

=> We decided to put the species name in the main title because the first panel does not have room for it due to the legend. As this figure has to take up the whole page (height-wise) anyway, we decided to keep the main title. No changes.

*I guess the journal will not accept the presentation of these figure and then it could be worthwhile to wait with the changes. You could in any case already now try to improve the presentation of the figures by using the free spaces on the right and left sides of the figures. Thus, make three rows, 3, 3 and 4, instead and remove "Model levels" and values on the y-axis corresponding to the figures presented at the middle and right positions. This means that you can move the figures closer to each other. The results of it will be that you get somewhat larger figures, thus, it will be possible to remove the titles and put all species names in the figures. In addition, you have to explain the species name in the figure caption or at least write in the figure caption that this information can be found in Section 4.2.*

Figure 8

Write "Model levels" connected to the y-axis.

=> The caption already stated that it 5 is the same as Figure 7. We tried to reserve more x-axis space to zoom in differences between the experiments here, dropping y-axis title intentionally. No changes made.

*It is not acceptable to refer to the previous figure concerning text belong to the y-axis. You will neither reduce the size of the figures by including this text. Thus, you have free space available on the left side of the figures and you need only to include "Model levels" (once) at this place. You could also exclude the values on the y-axis corresponding to the right figure.*

Figure 11

Suggestion "Figure 11. Same as Fig. 10, while here the results of forecast accuracy (%) for categorical forecasts are presented, subdivided according to classification of air quality in Tables 2 and 3."

Include "Model level" on the y-axis.

=> Figure 11 shows different statistics, not the model level, as shown in the main title. No changes made.

*The authors are of course right in the comment above, but the figure caption is once again not clear. For example you have to write "with the exception for" instead of just "except" and*

*"Tables 2 and 3" instead of "Table 2 and 3". However, even with these changes suggested the sentence is not clear and complete.*

**Technical corrections**

It was very hard to read through all the comments because they are mostly incomplete and are not separated by lines. In many cases, the modifications that the reviewer suggested either do not flow well in our manuscript, or misrepresent our points, or are simply wrong in grammar. The reviewer 5 tries to change our manuscript line by line in his/her way, but we should ask for being respectful for the authors' work. But here are our responses to the questions:
*It seems that in the creation of the pdf-file, when my review was uploaded into the reviewing system, the space between two comments disappeared. It is pity that I did not notice that. Even so, all the comments are clearly separated in the way that every new comment start with a new line, for example "Line 2" and "Line 3" corresponding to the abstract. My comments are either incomplete. With an example I explain here how to handle the comments. This text "…….. underestimates predicted surface $PM_{2..5}$………" on Line 11 in the abstract means that it is only these words in the sentence that the authors need to put attention on. The following comments presented below remain for the authors to take in consideration, and for some of these I have included a motivation.*

Abstract

Line 2. Suggestion "The Korean Geostationary Ocean Color Imager (GOCI) provides, based on daily high temporal and spatial resolution data, unprecedented information on air pollutants over the upstream region of the Korean peninsula for the last decade."
Note that GOCI is not a satellite and the phrase "…..has monitored the East Asian region in high temporal and spatial resolution every day for the last decade……" need to be rewritten.

Line 3. "the GOCI aerosol optical depth (AOD)" instead of "the GOCI Aerosol optical depth (AOD)."

Line 6. "……assimilated with three-dimensional…..technique in the Weather….."

Line 9. "Sensors" are not observations and therefore change to "….(MODIS) AOD…….", but this is a better suggestion "…… (MODIS) AOD and in-situ ground-based fine particle matter $(PM_{2,5})$.", since then you don't need to include "observations" again in the sentence.

Line 11."……. underestimates predicted surface $PM_{2..5}$………" You have to be more clear what you mean with the second time you write "surface $PM_{2..5}$" in the sentence. Is it "predicted" or "estimated" or something else?

 Line 13. The last part beginning with "with the most…" of this sentence is not clear.
=>We clearly demonstrated the most significant contributions to the prediction of heavy pollution events in Figure 11 where the assimilation of GOCI data produced the biggest improvement in b) high pollution accuracy.

*Make then a new sentence of it "This resulted in the most significant contributions to the prediction of heavy pollution events over South Korea."*

Introduction

Line 21. "Surface concentrations" of what ?

=> of chemical species. We believe this should be clear as the previous paragraph is immediately followed by this one.

*You have to make the current sentence clear and not rely on the information in the previous paragraph, particularly not in this case when you also describe aerosols in the previous paragraph.*

Line 23. "The latter is highly dependent on……" The word "relies" don't suit here.

Line 3. "….performs chemical simulations according to 3-km horizontal resolution at present day." *Please accept this change or make the original text more clear.*

Line 4. What is meant by "these fast-varying complex mechanisms" or what is it pointed to?

=> All the mechanisms described in the previous paragraph, particularly the aerosol-meteorology interaction at short time scales. Again, this sentence is also connected with the paragraph right ahead.

*To improve the text the best solution is to include the information you touch on in the answer above. Another but not the best solution could be to write it like this : "For such a high-resolution application and for situations with very high aerosol concentrations, the fast-varying complex mechanisms described above might be better represented through online coupling between chemical and meteorological components."*

***The authors have not given any respond to the comments below and again this is not ok***.

Lines 9-10. This sentence need to be re-written.

*Suggestion: "Chemical modeling are associated with large uncertainties, particularly concerning emission data and to simulate meteorology process. One of the most effective ways of utilizing aerosols is instead to assimilate aerosol observations into the forecast model and improve the initialization of aerosol simulations."*

Line 12. Not clear written: "(usually in the optical properties)"

*Suggestion: "(usually the optical properties)"*

Line 12. Change "observed information" to "information" or "results"

Line 15. Change "for" to "of"

Line 16. "….conducted in Korea between 1 and 12 June 2016……………"

Line 17. Remove "a field campaign"
*The original sentence "An international cooperative air quality field study conducted in Korea between 1 and 12 June 2016, named as the Korea-United States Air Quality (KORUS-AQ), was a field campaign jointly developed by air quality researchers in the United States and South Korea to improve our understanding of major contributors to poor air quality in Korea for May 1-June 12, 2016." need to be improved. Note that this sentence include the word "field" twice and this is redundant. I think also that it is possible and more suitable to make the sentence shorter.*
*Suggestion: The Korea-United States Air Quality (KORUS-AQ) field campaign conducted in Korea between 1 and 12 June 2016 was developed by researchers in the United States and South Korea to improve our understanding of the major contributors to the poor air quality in Korea."*

Line 21. "….occurred due to long-range……"

Lines 25-27. Based on the Korean Geostationary Ocean Color Imager (GOCI) onboard the Communication, Ocean, and Meteorology Satellite (COMS) retrievals of hourly AOD scenes, for multiple spectral bands, are centred with respect to the Korean peninsula during daytime (Kim et al., 2017). AOD scenes with high spatial and temporal resolutions are available since 2010."
*The sentence in the original manuscript lose connection at "....spectral bands monitoring....". This is a new suggestion to uses two sentences instead of the original one:*
*"The Korean Geostationary Ocean Color Imager (GOCI) onboard the Communication, Ocean, and Meteorology Satellite (COMS) provides hourly AOD retrievals at multiple spectral bands. GOCI monitoring the East Asian region centered on the Korean peninsula during daytime (Kim et al., 2017). AOD scenes with high spatial and temporal resolutions are available since 2010."*

Line 28. "It has been demonstrated"
*The original sentence need to be re-written. Here is a new suggestion:*
*"It has been demonstrated that GOCI data are associated with high accuracy compared to the low-orbiting Moderate Resolution Imaging Spectroradiometer (MODIS) and Visible Infrared Imaging Radiometer Suite 30 (VIIRS) products (Lee et al. (2010); Wang et al. (2013); Xiao et al. (2016); Choi et al. (2018)"*

Line 31. "assimilating AOD derived from MODIS observations (Remer et al., 2005)"
*Should it be like this: "Liu et al. (2011) were the first to implement assimilation of Aerosol Optical Depth (AOD) retrieved from the MODIS sensors (Remer et al., 2005) into the*

*National Centers for Environmental Prediction (NCEP) Gridpoint Statistical Interpolation (GSI; Wu et al. (2002); Kleist et al. (2009)) system."*

Line 34. "forecasts of a dust storm"

Line 35. "..widely used for air quality forecasting. The system has been extended for….."
*Please make two sentences of this relatively long original sentence.*

Line 5. "MOdel….."

Line 8. "….assimilation **of** AOD improved…"

Line 9. "In the present study, the assimilation............system has been extended to be better used in the GOCI AOD retrievals during the current investigation period…"
Line 10. Not clear what is meant by this "careful investigation of data characteristics."
*The change suggested here by me don't hold and I am sorry for that. However, this sentence on line 9, page 3, in the original version of the manuscript need in any case to be improved. Here is a new suggestion: "In the present study, the assimilation capabilities in the GSI 3DVAR system has been further extended to optimize the use of GOCI AOD retrievals during the KORUS-AQ period. (with careful investigation of data characteristics) I suggest to excluding the last part of this sentence, in the brackets, or improve it.*

Lines 10-13. The last part of this sentence is not clear: "……compared to that of other observations."
*I suggest creating two sentences of this long original sentence.*

Line 13. "data and examine"
*Sorry, I referred to the wrong line here, thus, it should be Line 12 instead. Here you should remove the redundant word "then" in the original manuscript.*

Line 18. "conclusions are presented"
*This is one of several examples showing that the manuscript needs an English proof-check.*
*Thus, the phrase "conclusions are made in Section 5" is not grammatical correct.*

---

## Author Response (AR2)

Dear anonymous referee 1,

Thank you for your time and effort trying to improve our manuscript.
In response to your review, we updated Figs. 3, 7 and 8 along with new captions and rewrote the caption for Fig. 11.
Please find our point-by-point response in red below.
(Our original response in the first round of review remains with "=>". The reviewer's new comments come with "R:" while our new response starts with "A2:".)

Response to anonymous referee 1:

I have used the first review of the present original version of the manuscript as a ground here in my second review of the current study. Additional comments by me are denoted with Italic font style. For some of the issues below I have included the answers from the authors to my comments in the first review, which are marked with blue colors and an arrow at the begging of the text. The comments by me and answers by the authors below are pointing to updated page and line numbers that correspond to the revised version of the manuscript. I have only included issues from my first review of this manuscript that are still relevant to include here.

Major comment

2. There are issues with the language, which need to be improved. In the section Technical corrections below suggestions are given in an attempt to improve the language and clearness of the manuscript. However, my review and corrections of the manuscript concerning the language has only been carried out for the abstract and Introduction to show that the clearness of the text need to be improved. Therefore, I recommend that the full text needs an English proof-check.
=> This manuscript has been internally reviewed twice in our lab and proofread by another native English speaker. As we replied to the reviewer's technical corrections at the very bottom, most of the corrections the reviewer suggested are incorrect in English or distorted our points, if not irrelevant, in the manuscript. The reviewer raised most of the comments or questions regarding MODIS retrievals, which is not the focus of our study but included only for completeness, we thus wanted to stay focused on our goal and highlights of our 10 work. However, we appreciate different views and tried our best to accommodate the reviewer's comments and reflect them in our manuscript unless we have a specific reason not to.

R: It is not ok by the authors to ignore many of my suggestions concerning the language, thus, here they have not replied at all (see technical corrections below). My suggestion of an English proof-check remain. However, when now have reading more of particular Sections 4 and 5 I do not think it is much work needed to improve the text in the Sections 3, 4 and 5.

A2: Please note that, in respect of reviewer's comment, we'd already gone through another round of proof-reading and made necessary corrections throughout the manuscript. But because the reviewer's comments have numerous error in grammar, we do not think they can improve our manuscript. For instance, your last comment was *"conclusions are made in Section 5" is not grammatical correct*.
There, your statement itself is wrong in grammar because you should use "grammatically", not "grammatical". In fact, our sentence "conclusions are made ∼" is correct and commonly used in the literature. As we responded in the first round of review, we chose not to take your suggestions when we found your suggestions problematic - either trivial or inappropriate, if not wrong - or when we disagreed with you. But we never ignored any of your comments.

Specific comments

Page 5 Line 17: MODIS is not a satellite and which version is used here, 6.1?
=> We modified "MODIS and GOCI satellites" to "MODIS and GOCI sensors". And yes, version 6.1 was used.

R: Since I do not find it in the revised manuscript, please include text that describe it is Version 6.1 that is used in the present study.

A2: We chose not to specify the versions of retrieval algorithms for both MODIS and GOCI sensors here because it has nothing to do with our observation operators described in the statement.

Line 22. It is not correct to write that AOD measures something. This sentence need to be re-written.

R: By writing "AOD measures the amount. . . ." then the latter word is a verb and it somewhat odd that a parameter measuring something (an active action), although it might be correct in the English language. By using "AOD is a measure of the amount. . . ." Instead then the latter word is a substantive, which is a more suitable phrase. Itt is the same as writing "AOD is an estimate of the amount. . . ..".

A2: Disagree. We do not see any difference between "AOD measures the amount. . . ." and "AOD is a measure of the amount. . . .". No changes.

Page 7 Lines 19-21. "Following Remer et al. (2005), observation errors are specified as the retrieval errors: (0.03 + 0.05 * AOD) over ocean and (0.05 + 0.15 * AOD) over land. They do not include the representativeness error and are slightly smaller than those for GOCI AOD, as described below."

R: The former part of this sentence sounds better now. However, it is very strange that the authors refer to retrieval errors corresponding to the ocean retrievals, estimated by the MODIS aerosol team (including Lorraine Remer), that are not valid anymore. Please update the retrieval error and corresponding references according to my comment in the first review of the original version of the manuscript.

A2: Again, MODIS data was included only for the completeness and we never intended to make the use of MODIS retrievals optimal in this study. For its error specification, we followed previous data assimilation studies like Liu et al. (2011) where it is also referred to Remer et al. (2005). No changes.

Page 9 Lines 7 and 8. The word "validation" can be used when comparing satellite derived AOD against ground-based sunphotometer measurements. However, you cannot validate AOD obtained from passive remote sensing against AOD derived from observations with another satellite sensor used in passive remote sensing.

=> We do not want to argue about how others described their work. The term of "validation" was used in Choi et al. (2018) and we just adopted it here. Also, in a broad sense, the terminology of "validation" is commonly used when one data is evaluated against another independent observation in the data assimilation community, so we do not see it problematic. No changes.

R: You don't need to argue here and instead of just adopt this term and rely on what is commonly used it is better to go to a dictionary. The word validation is a too strong word to describe an inter-comparison of AOD obtained with passive remote sensing from two different satellite platforms (see also comment below corresponding to Figure 2). The word "evaluation", used in the text above by the authors or "comparison" are options that are more suitable.

A2: Disagree. In many data assimilation studies, "validation" has long been used interchangeably with "evaluation". No changes.

Lines 18-21. Concerning the sentence "When these different observation errors were applied to GOCI retrievals in the assimilation, the smallest error ($\epsilon_2$) produced slightly better fits to observations specially for the high values (AOD > 2). . . . . . ." This statement seems not hold, since $\epsilon_2$ is not better than $\epsilon_1$ over land for the situations with lower AOD.

=> We do not understand why our statement doesn't hold due to the case of lower AOD, which we did not even discuss here. We mentioned that the smallest error ($\epsilon_2$) produced slightly better fits to observations for the high (!) AOD values. We also stated that such a result is not statistically significantly different, so we do not understand why the reviewer is arguing over the statement. No changes.

R: Remove then "specially" from the original sentence in the manuscript, then your argumentation and statement hold.

A2: The smaller the observation error gets, the better the model states fit to the observation. We described the trend here and tried to pull our model values closer to high AOD values, in particular. We believe our statement holds, even with "specially". No changes.

Lines 29-31. Suggestion, change the sentence "This is partly because AOD is not directly associated with surface PM2.5 ........." to "This is partly because AOD is not directly related to surface PM2.5 ........ 2, since "related" is a more correct word to use.

A2: Disagree. The expression of "associated with" is not really different from "related to". No changes.

Section 5 and page 14

Line 26 and first sentence in this section. I think it is too strong positive words are used here when describing the GOCI AOD retrievals. I am sorry about this comment since I refer to the sentence on page 15 instead of the correct page 14: "GOCI AOD retrievals provide reliable and consistent aerosol information, monitoring air pollutants flowing over to the Korean peninsula at high resolution every day." Thus, I think it is too strong positive words used in the first part of the sentence when describing the GOCI AOD satellite retrievals on the accuracy of air quality forecasting This is what the authors write in the abstract "During the Korea-United States Air Quality (KORUS-AQ) period (May 2016), the impact of GOCI AOD on the accuracy of air quality forecasting is examined by comparing with other observations including Moderate Resolution Imaging Spectroradiometer (MODIS) sensors and fine 10 particulate matter (PM2:5) observations at the surface." Concerning the latter parameter the result presented in Figure 5 is not impressive. In addition, you cannot rely the statement on an inter-comparison with AOD derived from observations carried out from another satellite platform. The first and second parts of the sentence is not synchronized and this is an odd phrase "air pollution flowing over. . . ." Therefore, here is a suggestion: "GOCI provide daily AODs with high spatial resolution and frequently detect air pollutants over to the Korean peninsula."

A2: Thanks for your suggestion. We agree that "air pollution flowing over to..." is a bit awkward. We now take out "flowing" and change the part as "air pollutants over the Korean peninsula..."

Figures

Figure 2 It is not correct to write that AOD is retrieved at this time, since it is the observations that is carried out at this time and it is a very long way to come up with an estimate of AOD, for example you have to introduce a model that describe radiation transfer in the atmosphere. Change "retrieved" to "corresponding" in the first sentence of the figure caption to Figure 2. Describe in the figure caption the solid black box introduced.

=> This study is not meant for describing the retrieval process, but how the data is used in the assimilation cycle. The data was processed at the time and that's how they are described and presented in Figure 2. Even in-situ measurements such as radiosonde do not report the values at the exact time (depending on the vertical levels as it goes up), but in the data assimilation context, that's how they are all described. Please note that we already illustrated the temporal distribution of the data in the last paragraph of page 7 ("In terms of temporal distribution, ). In response to your last comment, though, we added one paragraph "Domain 2 is marked as a black box in each panel." at the end of the caption.

R: It has nothing to do with reporting the values at the exact time, instead you have to separate "retrieved" and "observed", thus, the TOA radiance was measured/observed at the current time. It is just simply to write it more correctly so you don't hide that AOD derived from satellite measurements is not a direct observation, instead it need to be related to measured radiance scattered only by aerosols. The latter means that the contributions of Rayleigh scattering and surface reflection have to be reduced for. In addition, to relate AOD with TOA radiance associated with purely aerosols you have to introduce radiative transfer calculations. I present here a new suggestion: "Field of GOCI AOD at 550 nm derived/retrieved from observations carried out at 2016-05-01_06:00:00 UTC. . . . . . . . . . . ."

A2: Disagree. We already mentioned GOCI data as retrievals. And we've never seen any literature on data assimilation that described the retrieval data in the way you suggested (e.g. "Field of GOCI AOD at 550 nm derived/retrieved from observations carried out at 2016-05-01_06:00:00"). No changes.

Figure 3. You have to include information that the right y-axis goes down to 2000 (the latter value I picked up in the file with the author's response in the review process), since it is otherwise hard for the readers to understand part of the figure. You need to use different symbols for each cycle or at least each pair of cycles. The half-last part of the last sentence in the figure caption need to be re-written.

A2: Figure 3 is now changed with new caption, as suggested.

Figure 4. In the figure caption you have to refer to the body text about the three different types of observation errors. Take the color blind persons in consideration and use the three colors in combination with solid, dashed and dotted lines and separate land and ocean with heavy and normal lines, respectively.

=> This draft uses a lot of colors throughout the figures, and is not meant for color-blinded readers, unfortunately. But based on your comment, we added "The first two errors ($\epsilon_1$ and $\epsilon_2$)) are described in equations (3) - (6) and the third error ($\epsilon_3$) increases by 20% everywhere." in the caption.

R: I agree on "a lot of colors", but why not make it easier, particularly for the colorblind person, by improving those figures for which it possible to do that for? You do not use colors in Figure 3 and it seems to work relatively well (see comments concerning Figure 3 above).

A2: The reason Figure 3 could go with black and gray is because it only needed two different colors. Now we have total of six cases to present here, and they are very close to each other, especially near the low x values. We believe colors can show these errors more clearly. No changes.

Figure 7. It is a lot of space in the figure and therefore write the names of the species in all figures.

=> We decided to put the species name in the main title because the first panel does not have room for it due to the legend. As this figure has to take up the whole page (height-wise) anyway, we decided to keep the main title. No changes.

R: I guess the journal will not accept the presentation of these figure and then it could be worthwhile to wait with the changes. You could in any case already now try to improve the presentation of the figures by using the free spaces on the right and left sides of the figures. Thus, make three rows, 3, 3 and 4, instead and remove "Model levels" and values on the y-axis corresponding to the figures presented at the middle and right positions. This means that you can move the figures closer to each other. The results of it will be that you get somewhat larger figures, thus, it will be possible to remove the titles and put all species names in the figures. In addition, you have to explain the species name in the figure caption or at least write in the figure caption that this information can be found in Section 4.2.

A2: Figure 7 is now replotted. Thanks for your suggestion.

Figure 8. Write "Model levels" connected to the y-axis. => The caption already stated that it 5 is the same as Figure 7. We tried to reserve more x-axis space to zoom in differences between the experiments here, dropping y-axis title intentionally. No changes made.

R: It is not acceptable to refer to the previous figure concerning text belong to the y-axis. You will neither reduce the size of the figures by including this text. Thus, you have free space available on the left side of the figures and you need only to include "Model levels" (once) at this place. You could also exclude the values on the y-axis corresponding to the right figure.

A2: The y-axis label "Model levels" is now added.

Figure 11. Suggestion "Figure 11. Same as Fig. 10, while here the results of forecast accuracy (%) for categorical forecasts are presented, subdivided according to classification of air quality in Tables 2 and 3." Include "Model level" on the y-axis.

=> Figure 11 shows different statistics, not the model level, as shown in the main title. No changes made.

R: The authors are of course right in the comment above, but the figure caption is once again not clear. For example you have to write "with the exception for" instead of just "except" and "Tables 2 and 3" instead of "Table 2 and 3". However, even with these changes suggested the sentence is not clear and complete.

A2: We now changed the entire caption as "Time series of forecast accuracy (%) of the hourly forecasts from the 00 Z initialization for May 4 - 31, 2016 in domain 2 for categorized events based on hourly surface $PM_{2.5}$ concentrations, as defined in Tables 2 and 3."

Technical corrections

It seems that in the creation of the pdf-file, when my review was uploaded into the reviewing system, the space between two comments disappeared. It is pity that I did not notice that. Even so, all the comments are clearly separated in the way that every new comment start with a new line, for example "Line 2" and "Line 3" corresponding to the abstract. My comments are either incomplete. With an example I explain here how to handle the comments. This text "........ underestimates predicted surface PM2..5........." on Line 11 in the abstract means that it is only these words in the sentence that the authors need to put attention on. The following comments presented below remain for the authors to take in consideration, and for some of these I have included a motivation.

Abstract

Line 2. Suggestion "The Korean Geostationary Ocean Color Imager (GOCI) provides, based on daily high temporal and spatial resolution data, unprecedented information on air pollutants over the upstream region of the Korean peninsula for the last decade." Note that GOCI is not a satellite and the phrase "…..has monitored the East Asian region in high temporal and spatial resolution every day for the last decade......" need to be rewritten.
A2: Reviewer seems to be obsessed with terminology. As we already described it as a sensor, we do not make any changes here.

Line 3. "the GOCI aerosol optical depth (AOD)" instead of "the GOCI Aerosol optical depth (AOD)."
A2: Changed, as suggested. Thank you.

Line 6. "......assimilated with three-dimensional…..technique in the Weather….."
A2: One can use "in" rather than "with". No changes.

Line 9. "Sensors" are not observations and therefore change to "….(MODIS) AOD…….", but this is a better suggestion "…… (MODIS) AOD and in-situ ground-based fine particle matter (PM2,5).", since then you don't need to include "observations" again in the sentence.
A2: We don't see any differences. No changes.

Line 11."....... underestimates predicted surface PM2..5........." You have to be more clear what you mean with the second time you write "surface PM2..5" in the sentence. Is it "predicted" or "estimated" or something else?
A2: We believe that it is obvious to be the forecast value because we clearly mentioned that the impact lasts for about 6 h at the end of sentence.

Line 13. The last part beginning with "with the most…" of this sentence is not clear.
=>We clearly demonstrated the most significant contributions to the prediction of heavy pollution events in Figure 11 where the assimilation of GOCI data produced the biggest improvement in b) high pollution accuracy.
R: Make then a new sentence of it "This resulted in the most significant contributions to the prediction of heavy pollution events over South Korea."
A2: This is basically the same as what we described. No changes.

Introduction
Line 21. "Surface concentrations" of what ?
=> of chemical species. We believe this should be clear as the previous paragraph is immediately followed by this one.
R: You have to make the current sentence clear and not rely on the information in the previous paragraph, particularly not in this case when you also describe aerosols in the previous paragraph.
A2: Disagree. We believe it is clear enough for readers. No changes.

Line 23. "The latter is highly dependent on......" The word "relies" don't suit here.
A2: Disagree. "depend on" is not much different from "rely on". No changes.

Line 3. "....performs chemical simulations according to 3-km horizontal resolution at present day." Please accept this change or make the original text more clear.

A2: We don't see anything unclear. No changes.

Line 4. What is meant by "these fast-varying complex mechanisms" or what is it pointed to?

=> All the mechanisms described in the previous paragraph, particularly the aerosol-meteorology interaction at short time scales. Again, this sentence is also connected with the paragraph right ahead.

R: To improve the text the best solution is to include the information you touch on in the answer above. Another but not the best solution could be to write it like this : "For such a high-resolution application and for situations with very high aerosol concentrations, the fast-varying complex mechanisms described above might be better represented through online coupling between chemical and meteorological components."

A2: So you suggested that we add "described above" in the middle. The previous paragraph that described the fast-varying complex mechanisms is directly followed by this paragraph. Thus, we see "described above" as redundant. No changes.

The authors have not given any respond to the comments below and again this is not ok.

A2: We did not intend to ignore any of them, but as we made it clear in the first round, we found most of your suggestions below either cosmetic or irrelevant, if not wrong. You basically asked us to rewrite the entire page for pages 1 - 3 for something that does not matter to our main points or the quality of this work. But we reflected a few of them in our manuscript for clarity. Thank you for your time and suggestions.

Lines 9-10. This sentence need to be re-written.

Suggestion: "Chemical modeling are associated with large uncertainties, particularly concerning emission data and to simulate meteorology process. One of the most effective ways of utilizing aerosols is instead to assimilate aerosol observations into the forecast model and improve the initialization of aerosol simulations."

A2: Disagree. No changes.

Line 12. Not clear written: "(usually in the optical properties)"

Suggestion: "(usually the optical properties)"

A2: Incorrect. No changes.

Line 12. Change "observed information" to "information" or "results"

A2: There are no such thing as "observed results". No changes.

Line 15. Change "for" to "of"

A2: It is correct to say "prediction for precipitation,...". No changes.

Line 16. "....conducted in Korea between 1 and 12 June 2016.............."

A2: We already mentioned the time period at the end of the sentence. No changes.

Line 17. Remove "a field campaign"

The original sentence "An international cooperative air quality field study conducted in Korea between 1 and 12 June 2016, named as the Korea-United States Air Quality (KORUS-AQ), was a field campaign jointly developed by air quality researchers in the United States and South Korea to improve our understanding of major contributors to poor air quality in Korea for May 1-June 12, 2016." need to be improved. Note that this sentence include the word "field" twice and this is redundant. I think also that it is possible and more suitable to make the sentence shorter.

Suggestion: The Korea-United States Air Quality (KORUS-AQ) field campaign conducted in Korea between 1 and 12 June 2016 was developed by researchers in the United States and South Korea to improve our understanding of the major contributors to the poor air quality in Korea."

A2: Your suggestion doesn't look any better to us. No changes.

Line 21. "....occurred due to long-range......"
A2: "...occurred by" is not incorrect. Keep it.

Lines 25-27. Based on the Korean Geostationary Ocean Color Imager (GOCI) onboard the Communication, Ocean, and Meteorology Satellite (COMS) retrievals of hourly AOD scenes, for multiple spectral bands, are centred with respect to the Korean peninsula during daytime (Kim et al., 2017). AOD scenes with high spatial and temporal resolutions are available since 2010." The sentence in the original manuscript lose connection at "....spectral bands monitoring....". This is a new sugges-
10   tion to uses two sentences instead of the original one: "The Korean Geostationary Ocean Color Imager (GOCI) onboard the Communication, Ocean, and Meteorology Satellite (COMS) provides hourly AOD retrievals at multiple spectral bands. GOCI monitoring the East Asian region centered on the Korean peninsula during daytime (Kim et al., 2017). AOD scenes with high spatial and temporal resolutions are available since 2010."
A2: It is just another way of saying. No differences. No changes.

Line 28. "It has been demonstrated"
The original sentence need to be re-written. Here is a new suggestion: "It has been demonstrated that GOCI data are associated with high accuracy compared to the low-orbiting Moderate Resolution Imaging Spectroradiometer (MODIS) and Visible Infrared Imaging Radiometer Suite 30 (VIIRS) products (Lee et al. (2010); Wang et al. (2013); Xiao et al. (2016); Choi et al.
20   (2018)"
A2: Again, no differences. No changes.

Line 31. "assimilating AOD derived from MODIS observations (Remer et al., 2005)" Should it be like this: "Liu et al. (2011) were the first to implement assimilation of Aerosol Optical Depth (AOD) retrieved from the MODIS sensors (Remer
25   et al., 2005) into the National Centers for Environmental Prediction (NCEP) Gridpoint Statistical Interpolation (GSI; Wu et al. (2002); Kleist et al. (2009)) system."
A2: Disagree. No changes.

Line 34. "forecasts of a dust storm"
30   A2: It is correct to state "forecasts in a dust storm event". No changes.

Line 35. "..widely used for air quality forecasting. The system has been extended for....."
Please make two sentences of this relatively long original sentence.
A2: It is only three-line long. No changes.

Line 5. "MOdel....."
A2: Changed, as suggested.

40   Line 8. "....assimilation of AOD improved..."
A2: AOD means AOD retrievals. Why should we take out "retrievals"? No changes.

Line 9. "In the present study, the assimilation............system has been extended to be better used in the GOCI AOD retrievals during the current investigation period..."
45   A2: Nothing is wrong with our original expression. No changes.

Line 10. Not clear what is meant by this "careful investigation of data characteristics."
R: The change suggested here by me don't hold and I am sorry for that. However, this sentence on line 9, page 3, in the original version of the manuscript need in any case to be improved. Here is a new suggestion: "In the present study, the assimilation

capabilities in the GSI 3DVAR system has been further extended to optimize the use of GOCI AOD retrievals during the KORUS-AQ period. (with careful investigation of data characteristics) I suggest to excluding the last part of this sentence, in the brackets, or improve it.
A2: Disagree. Keep our original statement.

Lines 10-13. The last part of this sentence is not clear: "......compared to that of other observations."
I suggest creating two sentences of this long original sentence.
A2: Disagree. Keep it.

Line 13. "data and examine"
Sorry, I referred to the wrong line here, thus, it should be Line 12 instead. Here you should remove the redundant word "then" in the original manuscript.
A2: Removed, as suggested.

Line 18. "conclusions are presented"
This is one of several examples showing that the manuscript needs an English proof-check. Thus, the phrase "conclusions are made in Section 5" is not grammatical correct.
A2: Disagree. No changes.

20    Dear anonymous referee 2,

      Thank you for your meticulous review of this draft and great suggestions.
      We truly appreciate your help on finding a bug in Fig. 7, which is now replotted with new caption. We also found most of your
      comments very helpful in clarifications, improving our draft significantly.
25    Our line-by-line response to your comments can be found in "A2" below. (Our response in the first round remains as "A:".)

      Response to anonymous referee 2:
      My major comment remains being the evaluation of the modeled/assimilated vertical profiles using data available from the
30    KORUS-AQ campaign. They can address this in different ways. There was an HSRL in Seoul (see sample data in Peterson et
      al., 2019) that can provide full time series of vertically resolved aerosol extinction that can be compared to the simulations.
      Also, there was an HSRL in the DC8 aircraft, so the comparison can be done for some days across the Korean peninsula. Also,
      data from the AMS onboard of the DC-8 can be used to compute PM1 and compare to model estimates, there were 2-3 full
      vertical profiles over Seoul every day the aircraft flew. I'm not asking the authors to do all of these (although this would be
35    nice), but at least show some effort of trying to assess the skill of the model in representing vertical profiles and if the changes
      in vertical distribution generated by the assimilation are somewhat reflected in better agreement to the observations. If discrep-
      ancies arise this is still useful as one can blame model uncertainties (e.g., computation of optical properties) for them. Louisa
      Emmons (NCAR ACOM, same institution as authors) is a modeler that was heavily involved in the KORUS-AQ campaign,
      perhaps consulting with her on this topic might be a good idea.

      A2: Thank you for your suggestion. We may need to start with some clarification.
      In the first round of your review, you misunderstood our main results as the assimilation of GOCI retrievals had degraded,
      rather than improved, the forecasts in surface $PM_{2.5}$ concentration when not assimilated with surface $PM_{2.5}$ observations. You
      made the point as your first and the most important finding, which was completely contradictory to our major conclusion on
45    how useful the assimilation of GOCI retrievals was, particularly for high pollution events.
      Thus, in response to your review in the first round, we added three more figures to prove that the GOCI retrievals actually
      **improved** the forecasts with respect to **independent** $PM_{2.5}$ and AOD observations on the ground, as you suggested.
      But, to our surprise, you did not comment on this at all in the 2nd round of review whether our revised manuscript with the
      objective validation corrected your misunderstanding. As your major comment was all related to that and it is the most critical
      finding of this study, it is mandatory to agree to each other on this particular point, we believe.
      However, skipping the main point, you continue to ask for the model verification on the vertical profiles, now in order to ad-
      dress the issue of model uncertainties, which is off the topic.
      When you had thought that our assimilation of AOD degraded forecasts, it was legitimate to request for examining why and
 5    how the assimilation of AOD could result in poor forecasts. Along the line, it made sense for you to infer that the vertical pro-
      files in the model could have deviated from the observed ones, thus to ask for evaluating the profiles against other observations.
      In return, we provided three new figures to support our conclusions and proved that you had misunderstood our results.
      Now, if you corrected your misinterpretation of our results, you need to come with another rationale behind your request for
      the model evaluation.
10    At this point, we want to remind you that the main goal of this study is to demonstrate the benefit of the GOCI assimilation on
      the prediction of **surface** $PM_{2.5}$ concentration. And it is our belief that we had provided sufficient evidence for the value of the
      GOCI assimilation, including independent verification, as you suggested.
      Of course, the model uncertainty could play a nontrivial role on the analysis quality, but there are many other factors in the
      analysis system that can contribute to making the analysis successful to the same extent, such as the estimation of background
15    error covariance, observation error statistics, and the error in the observation operators. In the limitation of the simple GO-
      CART aerosol scheme and the poor quality of emission data, we do not think that accurate representation of vertical profiles in
      the model is a primary concern of this study on the air pollution at the surface.
      We thus disagree with you that we should include the evaluation of the model profiles using additional data from a field cam-
      paign. Please note that we already have a total of 16 figures and 4 tables to demonstrate the benefit of the GOCI assimilation.

Comments from initial review

R1: Initial reviewer comments, A: Author response, R2: New reviewer comment

R1: Figure 1. Why show observations for a given time? Why not show maybe an average of the period analized?

A: => Figure 1 simply shows the model domain with the observing network. No changes are made.

R2: But you can also use it to show average concentrations over the period analyzed instead of showing a random time. Also, it would be nice to get a second panel with the 2nd domain to see details on the observation distribution over the Korean peninsula

A2: => Figure 1 is now changed, as suggested.

R1: Figures 8 nd 7. You could model vertical distribution and impact after assimilation using airborne data and surface lidars deployed as part of KORUS-AQ

A: => Not clear on your point here. Figures 7 and 8 show how the model responded to the assimilation of observations in use. This analysis is needed to understand how our assimilation worked in the model space. It has nothing to do with verification.

R2: I apologize for the typo, I meant "You could evaluate model vertical distribution . . ." This is related to one of my main comments on using KORUS-AQ observations for evaluating vertical distributions, that the authors have not addressed.

A2: => Now we understood your point. However, we disagree that it is essential to evaluate the vertical distribution of the model variables over South Korea (or in the surrounding region) with two major reasons:

i) The positive impact of the GOCI assimilation is mainly attributed to a wide **horizontal** coverage of the **upstream** region (not the vertical penetration through the troposphere). This point is well illustrated in our figure 9 and summarized in the last section. Unfortunately, this cannot be verified against any other instruments located in Seoul or even with the DC8 aircraft that flew around South Korea (based on the flight tracks depicted in Fig. 1 in Peterson et al. (2019)). As marked in dark red in the GOCI panel in Fig. 9, the biggest impact of the assimilation was found around the Jilin Province in China, northwest of North Korea, where we do not have any measurement even during the KORUS-AQ campaign. That's why the GOCI AOD retrievals are so valuable, as we stressed throughout the manuscript.

ii) More importantly, the assimilation of AOD is not predominantly dependent on the model performance or behavior on the vertical structure of aerosol species at certain times. A successful data assimilation requires a tremendous effort on understanding and harmonizing various components with disparate characteristics, and the model error is just one of them. Even if we can evaluate the vertical profiles of a few model aerosol variables over a tiny portion of our model domain 1, it wouldn't give you a synthetic picture of how the vertical structure of the model affected the analysis quality (both quantitatively and qualitatively) because it is not possible to disentangle it from all the complicated behaviors in the model and the analysis systems in a statistically meaningful way.

This is simply not how you can make a real data assimilation system work. Therefore, we chose not to take the reviewer's suggestion.

Comments by line on revised manuscript

2 1. Provide references for this statement, latest IPCC report should do

A2: Thanks for your suggestion. We checked the latest IPCC report "IPCC, 2014: Climate Change 2014: Synthesis Report. Contribution of Working Groups I, II and III to the Fifth Assessment Report of the Intergovernmental Panel on Climate Change [Core Writing Team, R.K. Pachauri and L.A. Meyer (eds.)]. IPCC, Geneva, Switzerland, 151 pp.",

but could not find anything associated with aerosol-meteorology interaction. It reported a great deal of emission levels or effect on the global scale, but not on the local meteorology, which is not directly related to our study. So we chose to not include it.

20    3 3-8. You can add Park et al (2014) to this paragraph
A2: Again, thanks for the reference. We also recognized several other papers with the assimilation of GOCI using the CMAQ model. But one of the main points of this study is to examine the impact of GOCI assimilation on the online coupled forecasting system, as described in the paragraph. So it is not appropriate to refer to the study using the *offline* coupled system here. Not included.

4 5. You can cite LeGrand e t al. (2019) for the AFWA scheme
A2: Included. Thank you.

8 30-34. If I understood correctly, what you are trying to say here is that there are large values after thinning that were not
30    found in the original data. Please rephrase to make this clearer and to the point.
A2: The statement is now changed in lines 31-32, as below.
"When all the GOCI data thinned in the GSI system were checked for the entire month, there were such extreme values that did not exist in the original dataset for multiple cases." =>"For the month of May 2016, multiple cases with such extreme fake values were found after the thinning process."

9 7-16. I'm assuming subscript 1 and 2 in the error (eqns 3-6) correspond to the different verifying object (AERONET or satellite-based retrievals)? Please specify which is which. AERONET is generally treated as ground truth, so that's probably the one you should be using.
A2: For the clarity, we now added "We assign $\epsilon_1$ following their error specification with respect to AERONET and $\epsilon_2$ based on
40    their expected error against retrieved satellite AOD in GOCI YAER V2.".
But in the data assimilation framework, observation error can be further adjusted based on the model's representativeness. In other words, observation error in the analysis system does not need to be the same as the instrument error. But thanks for your comment.

45    9 27-29. This sentence is not clear, please rephrase.
A2: It is now changed from "it is not guaranteed that such an analysis better fit to AOD retrievals would actually lead to better forecasts in surface PM$_{2.5}$."
to "it is not guaranteed that the analysis in a good agreement with AOD retrievals would actually lead to better forecasts in surface PM$_{2.5}$."

9 30-31. "This is partly because AOD is not directly associated with surface PM2.5 . . . " I think what you are trying to say is that AOD is a column integrated quantity while PM2.5 is measured at the surface?
Might be better to specify it that way, the way it's currently stated is vague and not necessarily true (many approaches exist to
5    compute surface PM2.5 from AOD)
A2: We now added ", a column integrated quantity, " right after AOD.

9 30-31. ". . . and partly because large uncertainties in the forecast model and the emission forcing can dominate over the analysis error during the model integration" This is only applicable for forecasts, not for the analysis. Might want to split the
10    arguments.
A2: It is applicable for the analysis as it is computed based on the forecast error as well.
But in response to your comment, the paragraph in Page 9, 30-31 is now rewritten as below.
"Even though it would be hard to quantify the model error and the emission uncertainty (and their impact on the forecast quality), it might be worth checking the correlation between GOCI AOD retrievals and surface PM$_{2.5}$ observations before
15    evaluating the impact of GOCI AOD on surface PM$_{2.5}$ forecasts."
=> "Even if the efficiency of assimilating AOD on improving surface PM$_{2.5}$ forecasts can be largely affected by the quality of the forecast model and the emission data in use, the effectiveness of the AOD assimilation is based on the relationship between the column-integrated AOD and PM$_{2.5}$ on the ground. Therefore, it might be worth checking the correlation between GOCI

AOD retrievals and surface PM$_{2.5}$ observations for the cycling period."

9 26 – 10 2. Based on the bad correlation, you might want to mention that this is why a model is needed to "translate" AODs into PM2.5. And that this "translation" might depend on the ability of the model of properly represent the aerosols vertically, and the conversion from aerosol mass to optical properties.

A2: Good point. We now added one more sentence at the end of the paragraph.

25  "Such an indirect relationship between the two observations makes the analysis challenging because it can induce a large error in the observation operator and heavily depend on the model's ability of deriving PM$_{2.5}$ from the AODs based on the vertical structure of aerosol variables and the conversion from aerosol mass to optical properties."

Figure 7. How can be sulfate in the background so low (seems to be equal to 0) and different than 0 in the assimilation

30  experiments?

A2: Excellent point. We now fixed a bug in the plotting algorithm which did not properly convert the model variable "sulf" in ppmv to actual "sulfate" in ug/mg in the background fields. Figure 7 is replotted with the correction of the background fields in sulfate. We very much appreciate your careful review on this.

35  11 5-6 . "When all the observations are assimilated together (in "ALL"), it combines the effect of surface PM2.5 and GOCI retrievals, as expected". You might want to explain that this combined effect ends up changing the vertical distribution, pulling the surface levels towards PM2.5 and the upper levels to match the AOD columns.

A2: Thanks for your suggestion. The sentence is now modified to replace ", as expected" with "changing the vertical distribution of aerosol species to match with the AOD column values and pulling the surface states towards surface PM$_{2.5}$ concentrations.".

11 32. I think it's worth mentioning/discussing that although errors are reduced, none of the assimilation experiments are able to reduce the bias compared to the NODA experiment, which is pretty low to start with.

A2: In terms of systematic error, we focused more on the mean absolute error (mae) than the mean error (bias). In that sense, most experiments actually reduced the error from the one in NODA. As we already stated that in the same paragraph (Page 11,

45  lines 25-26), we decided not to add any more discussion related to it.

13 21-29. The data used for evaluation in Fig 14 correspond to 3 urban sites where the 9 km res model with simple aerosol chemistry and emissions that have no hourly variation will have a very hard time representing the observed fluctuations. For instance, see Nault et al., (2018), the model configuration you are running doesn't even consider secondary organic aerosol formation. You can pick sites that you assimilated that are close-by, plot them in the same way, and I would expect you find similar fluctuations. When you average many sites that are both urban and background, you expect some of these fluctuations to be smoothed out. My point is that I wouldn't blame observation quality in this case.

5  A2: Your point is taken. Now we changed

"As raw data, these observations do not seem to be reliable and fluctuate a lot for the entire period," to

"In the assimilation system, raw data are not considered to be reliable,".

14 23-24. I would add "and more detailed aerosol chemistry mechanisms"

10  A2: Thank you. We now added "and more sophisticated aerosol chemistry mechanisms".

14 26. "The best use . . ." I would tone down this statement

A2: Now it is changed to "One of the best ways of utilizing such invaluable observations  ".

15  15 2-4. Similar to my previous comment, might be better to just state that one is a column integrated quantity and the other is surface, and their connection depends on the vertical distribution of aerosols and conversion from mass to optical properties.

A2: These aspects are stated in the following points (ii and iii). We only change "surface PM$_{2.5}$" to "PM$_{2.5}$ on the ground".

22-23. Not correct, see Peterson et al (2019). There are periods of stagnation where local contribution generates pollution
20   episodes that are not negligible.

A2: This is correct and consistent with the findings in Peterson et al (2019).

We defined heavy pollution events as the cases with surface $PM_{2.5} > 50$ $\mu g/m^3$, as specified in page 13, line 32.

In Peterson et al. (2019), for the same $PM_{2.5}$ (in the bottom panel of Fig. 3), they highlighted the heavy pollution events (based on the same reference line of 50 $\mu g/m^3$) in orange and marked them as "Transport" cases.

25   Thanks for making us confirm that our finding is correct here. No changes.

Technical Corrections

30   1 7-10. This sentence reads as you are using MODIS and surface PM to evaluate the assimilation, but are actually assimilating them as well. Please rephrase.

A2: We now added "those of" as "$\sim$ comparing with those of other observations". Thanks for your clarification.

Figure 6. State in the caption which DA experiment is plotted.

35   A2: We now added a sentence at the end of the caption as below.

Here, "DA" refers to the "ALL" experiment.

10 1. "... coefficient of 0.33, the two observation types ..."

A2: "the" is now added in front of "two observation types...", as suggested.

10 17 "... observation types separately, ..."

A2: Corrected.

Figure 7. State concentration units in the caption. Also, use full specie name instead of abbreviation or reference location in
45   the text where species are defined, some of the names are not straightforward to to figure out.

A2: Thanks for your suggestion.

Figure 7 is now completely replotted with new caption "Vertical profile of 10 GOCART aerosol variables composed of $PM_{2.5}$ - unspeciated aerosol contributions to $PM_{2.5}$ (P25), sulfate, OC1 and OC2 (BC1 and BC2) as hydrophobic and hydrophilic organic (black) carbon, respectively, DUST1 and DUST2 (SEAS1 and SEAS2) as dust (sea salt) aerosols in the smallest and 2nd smallest size bins. All the variables shown are mixing ratios in the unit of $\mu g/kg$. Different experiments are depicted in different colors, as averaged over domain 2 for the period of May 4 - 31, 2016. Analysis ("A") is drawn as solid line while background (e.g. 6-h forecast; "B") as dashed line."

5   10 29 "... produces the largest PM2.5 throughout the ..."

A2: Modified, as suggested.

14 1. "As our best experiment "ALL" analyzed," not clear what this means, please rephrase

[revised manuscript text omitted]

---

## Author Response (AR3)

Dear Editor,

Thank you for your careful review of our revised manuscript.
Based on your comments, we made another revision, which we believe improved the clarity of our draft.
Please find our line-by-line response (with the modification in the manuscript marked in blue) below.

Comments to the Author:
First, thank you for your patience with the long handling time of this manuscript. Based on a careful review of the revised manuscript and the reviewer comments, I believe this manuscript can be publishable in ACP after the following minor (mostly editorial to improve readability) points have been addressed:

1. Abstract. Please give the abstract one more thorough read and revise it to be as specific and quantitative as possible to improve it's readability and make it more standalone.
=> Thanks for the comment. We now revised the abstract, as suggested. Details follow below.

Here are some examples on potential points to improvement:
- Line 1: What is "high spatial and temporal resolution"? Why not be more specific?
=> We now changed it to "high temporal (e.g. hourly) and spatial resolution (e.g. 6 km) ∼".

- Line 4: "...the analysis quality". Which analysis? Please specify.
=> By definition, a product of assimilation is called "analysis". As this study is all about our data assimilation and the paragraph already starts with "the GOCI is assimilated", the analysis clearly means our own analysis, as a result of our assimilation of GOCI data.
But to make it clear once again, we now changed it to "...the quality of the aerosol analysis".
Also, for those who are not familiar with data assimilation or the analysis procedure, we now add one more paragraph in section 2.1.1 (lines 22-24, page 4), as below:
∼, which is called the "analysis". (The analysis is then used to initialize aerosol variables in the forecast model (e.g. WRF-Chem) so that the quality of aerosol forecasts can be largely dependent on the quality of the aerosol analysis produced in the 3DVAR system.) "

- Line 5: What do you mean by "data characteristics"? Please specify.
=> Now it is extended as "∼data characteristics such as temporal and spatial distribution".

- Line 9: What do you mean by "...by comparing with those of other observations"? Please specify exactly which parameters were compared.
=> "those" is now changed to "effects".
In terms of the parameter, the following paragraph had specified it as surface $PM_{2.5}$. However, in respect of the editor's comment, we now change "...the accuracy of air quality forecasting" to "...the accuracy of surface $PM_{2.5}$ prediction".

=> For some clarification and smoothness, we also changed the paragraph as below.
Old: "...comparing with those of other observations including Moderate Resolution Imaging Spectroradiometer (MODIS) sensors and fine particulate matter (PM2.5) observations at the surface. Consistent with previous studies, the assimilation of surface $PM_{2.5}$ concentrations alone systematically underestimates surface $PM_{2.5}$ and its positive impact lasts mainly for about 6 h."
New: "...comparing with effects of other observations including Moderate Resolution Imaging Spectroradiometer (MODIS) sensors and surface $PM_{2.5}$ observations. Consistent with previous studies, the assimilation of surface $PM_{2.5}$ measurements alone still underestimates surface $PM_{2.5}$ concentrations in the following forecasts and the forecast improvements last only for about 6 h."

2. p. 7 line 30: Please specify that prepbufr is a data format.
=> It is now described as "prepbufr format", although our point is not on the data format, but on the "prepared" bufr data, as described in the following statement (which ∼).

3. p. 9 line 7: I agree with Reviewer 1 on the use of term "evaluation" instead of "validation". The fact that the term "validation" has been misused in the literature does not justify continuing to use it when "evaluation" would be the more accurate term. This is of course a minor point, but sometimes the terminology does matter.
=> Corrected.

4. Figures 7 and 8. Thank you for an extensive response to the Reviewer 2's request to include an evaluation of the vertical profiles. I do understand your point of the focus of the paper being the surface observations, but it would be good to explicitly comment on the possibility of doing such evaluation in your outlook and perhaps acknowledging the data sets mentioned by the reviewer (and possibly any other relevant ones).
=> In respect of your comment, we now added two more sentences in lines 5-9, page 11:

[revised manuscript text omitted]